# Identification, characteristics, and dynamics of Arctic extreme seasons

Katharina Hartmuth[1], Maxi Boettcher[1], Heini Wernli[1], and Lukas Papritz[1]

[1]Institute for Atmospheric and Climate Science, ETH Zurich, Zurich, Switzerland

**Correspondence:** Katharina Hartmuth (katharina.hartmuth@env.ethz.ch)

**Abstract.**

The Arctic atmosphere is strongly affected by anthropogenic warming leading to long-term trends in, e.g., surface temperature and sea ice extent. In addition, it exhibits a pronounced seasonal cycle and strong variability on time scales from days to seasons. Recent research elucidated processes causing short-term extreme conditions in the Arctic that are typically related to the occurrence of specific weather systems. This study investigates unusual atmospheric conditions in the Arctic on the seasonal time scale, characterized by surface temperature, precipitation, and the atmospheric components of the surface energy balance. Based on a principle component analysis in the phase space spanned by the seasonal-mean values of the considered parameters, individual seasons are objectively identified that deviate strongly from a running-mean climatology, and that we define as extreme seasons. Given the strongly varying surface conditions in the Arctic, this analysis is done separately in Arctic sub-regions that are climatologically characterized by either sea ice, open ocean, or mixed conditions.

Using ERA5 reanalyses for the years 1979-2018, our approach identifies 2-3 extreme seasons for each of winter, spring, summer, and autumn, with strongly differing characteristics and affecting different Arctic sub-regions. While some show strongly anomalous seasonal-mean values mainly in one parameter, others are characterized by a combination of unusual seasonal conditions in terms of temperature, precipitation, and the surface energy balance components. Two extreme winters affecting the Kara and Barents Seas are selected for a detailed investigation of their substructure, the role of synoptic-scale weather systems, and potential preconditioning by anomalous sea ice extent and/or sea surface temperature at the beginning of the season. Winter 2011/12 started with average sea ice coverage and was characterized by constantly above-average temperatures during the season related to a sequence of quasi-stationary cyclones in the Nordic Seas, favoring the frequent advection of warm air to the Barents Sea. An enhanced frequency of blocking anticyclones and a reduced frequency of cold air outbreaks in the Kara and Barents Seas further helped to maintain the warm anomaly. In contrast, winter 2016/17 started with a strongly reduced sea ice coverage and enhanced sea surface temperatures in the Kara and Barents Seas. This preconditioning, together with increased frequencies of cold air outbreaks and cyclones, was responsible for the large upward surface heat flux anomalies and strongly increased precipitation during this extreme season.

In summary, this study shows that extreme seasonal conditions in the Arctic are spatially heterogeneous, related to different near-surface parameters, and caused by different synoptic-scale weather systems, potentially in combination with surface pre-

conditioning due to anomalous ocean and sea ice conditions at the beginning of the season. The framework developed in this study and the insight gained from analyzing the ERA5 period will be beneficial for addressing the effects of global warming on Arctic extreme seasons.

## 1 Introduction

Near-surface atmospheric conditions in the Arctic show a high variability on synoptic to inter-annual temporal scales, which is superimposed on a strong, long-term warming trend (e.g., Serreze and Barry, 2011; Cohen et al., 2014). Key drivers of variability on the synoptic to weekly time scale are interactions with the mid-latitudes for instance via air mass exchanges (e.g., Woods et al., 2013; Laliberté and Kushner, 2014; Graversen and Burtu, 2016; Messori et al., 2018; Papritz and Dunn-Sigouin, 2020) and air mass transformations within the Arctic (Ding et al., 2017; Pithan et al., 2018; Papritz, 2020). Both air mass exchanges and transformations are found to be related to synoptic weather systems. On longer time scales, in contrast, memory effects and feedback mechanisms such as the sea ice albedo feedback (Arrhenius, 1896; Curry et al., 1995), the water vapor and cloud feedbacks (Vavrus, 2004; Graversen and Wang, 2009; Boisvert et al., 2016), as well as the temperature feedback (Pithan and Mauritsen, 2014) play an important role. Given this broad spectrum of processes, this leads to the question how variability on various temporal scales is inter-connected. In this study, we focus on the seasonal scale and it is our goal to analyze the role of intra-seasonal processes, including synoptic-scale weather systems, for the emergence of seasonal extremes in the Arctic. The following paragraphs provide the relevant background on the key near-surface meteorological parameters in the Arctic and how they are interrelated. Furthermore, we discuss the role of different synoptic-scale weather systems for the variability of these parameters and the occurrence of short-term extremes and seasonal anomalies in the Arctic.

Near-surface temperature, the components of the surface energy budget - including radiative and turbulent heat fluxes - as well as surface precipitation are especially important parameters linking the variability of the atmosphere with that of the ocean and the cryosphere. Large fluctuations in the surface energy budget, which themselves are closely linked to air temperature fluctuations, contribute to the variability of sea ice (Stroeve et al., 2008; Olonscheck et al., 2019), the ocean mixed layer as well as open ocean convection (e.g., Marshall and Schott, 1999). Radiative and sensible heat fluxes drive the variability of the surface energy budget components over sea ice (Lindsay, 1998), whereas over open ocean turbulent heat fluxes dominate (Segtnan et al., 2011). Precipitation variability influences snow cover, which is strongly linked to the albedo feedback, and it affects the freshwater balance of the Arctic Ocean and the Nordic Seas (Serreze and Francis, 2006; White et al., 2007), which jointly with turbulent heat fluxes impacts the thermohaline circulation (Dickson et al., 1996; Talley, 2008).

The three parameters - near-surface temperature, surface energy budget, and surface precipitation - do not vary independently from each other but they are interlinked. Thereby, the surface boundary conditions, i.e., sea ice vs. open ocean, strongly affect the type of linkages between parameters as well as feedback processes due to vastly different heat capacities. On synoptic time scales, for instance, warm and cold air advection strongly influence heat fluxes over the open ocean, where the most intense

upward fluxes occur in cold air outbreaks (Harden et al., 2015; Papritz and Spengler, 2017; Pope et al., 2020). On longer time scales, surface air temperature changes are largely influenced by variations in the sea surface temperature via surface sensible heat fluxes (Johannessen et al., 2016). In addition, incoming shortwave radiation is absorbed and can be released to the atmosphere later. Over sea ice, in contrast, temperature is to a large degree determined by the surface energy balance, which includes radiative and turbulent heat fluxes, conductive heat fluxes across the ice and latent energy for freezing and melting (Serreze and Francis, 2006). In winter, when the incoming shortwave radiation is strongly reduced, the surface sensible heat flux and net surface longwave radiation mainly determine the surface energy balance in regions covered by sea ice (Ohmura, 2012). These considerations reveal that a meaningful identification of extreme seasons in terms of the surface temperature, energy budget and precipitation parameters must take their co-variability and the underlying surface boundary conditions into account.

The role of synoptic-scale weather systems for inter-annual variability in the Arctic has been subject of multiple recent studies, which emphasized especially the importance of cyclones (Simmonds and Rudeva, 2012; Messori et al., 2018), blocking anticyclones (Wernli and Papritz, 2018; Papritz, 2020), and Rossby wave breaking (Liu and Barnes, 2015). Air mass exchanges between the mid-latitudes and the Arctic region are often facilitated by cyclones, which, on one hand, transport warm and moist air to higher latitudes (Sorteberg and Walsh, 2008; Messori et al., 2018), causing there an increase in downward heat fluxes as well as the formation of clouds and precipitation. On the other hand, the advection of cold and dry air in the cyclones' cold sector enhances ocean evaporation and heat fluxes into the atmosphere. Additionally, extreme moisture transport into the Arctic is often associated with events of Rossby wave breaking (Liu and Barnes, 2015), which can be strongly linked to the evolution of surface cyclones (Martius and Rivière, 2016). Air mass transformations within the Arctic can similarly result in anomalous conditions. Recent studies emphasized the importance of polar anticyclones and blocking events in the High Arctic for driving subsidence-induced adiabatic warming, leading to anomalies in surface temperature and net surface radiation which cause increased sea ice melting (Wernli and Papritz, 2018; Papritz, 2020). In winter, radiative heat loss under clear-sky conditions can lead to extreme cold conditions, whereas cloud formation favors the trapping of longwave radiation, thus providing a positive warming feedback and causing an increase in surface temperature (Burt et al., 2016; Boisvert et al., 2016; Woods and Caballero, 2016). Similarly, a persistent and strong tropospheric polar vortex over the pole can isolate polar air masses and result in anomalously cold conditions due to enhanced radiative cooling (Messori et al., 2018; Papritz, 2020). Therefore, air mass transport and air mass transformation can significantly influence the Arctic surface energy balance. Whereas the modification of turbulent heat fluxes is of particular importance over the open ocean, the impact on radiative fluxes, for instance due to an increase in the atmospheric moisture content, is highly relevant in regions covered by sea ice.

Several studies have analyzed short-term Arctic extreme events and the involved dynamical processes, for instance the unusual warm event in winter 2015/16, which led to above freezing temperatures close to the North Pole (Cullather et al., 2016) and caused significant sea ice melting in the Kara and Barents Seas (Boisvert et al., 2016). Binder et al. (2017) were able to show that several pathways of exceptional air mass transport caused this warm event. Another example is an extreme melt event

on the Greenland ice shield in July 2012 (Nghiem et al., 2012), which was found to be related to a blocking anticyclone and associated anomalous long-range transport of warm and humid air masses from the South (Hermann et al., 2020). Such extreme weather events can have significant long-term effects, particularly due to their impact on sea surface temperatures and sea ice extent. For instance, Simmonds and Rudeva (2012) have shown that a particularly intense Arctic cyclone in summer caused the dispersion and separation of sea ice, leaving the main sea ice pack more exposed and thus vulnerable to further melting. Similarly, the described extreme warm event in December 2015 caused positive anomalies in sea surface temperature and negative anomalies in sea ice concentration in the Kara and Barents Seas, which persisted throughout the year 2016 (Blunden and Arndt, 2017). Single events of extreme weather, causing episodes of strongly anomalous conditions such as exceptionally high or low surface temperatures, can thus have a major impact on seasonal-mean surface temperature, the formation and melting rates of sea ice, and on minimum and maximum sea ice extent.

Despite these insights, so far only little attention has been given to systematically understanding the characteristics of extreme seasonal-mean conditions in the Arctic, and the role of synoptic weather systems in their formation. Therefore, our study aims to address the following research questions:

1. How spatially (in)homogeneous are the seasonal-mean near-surface atmospheric conditions in the Arctic in winter and summer?

2. How can extreme seasons be defined objectively, based on a combined analysis of different key surface parameters in Arctic sub-regions?

3. In which way do synoptic-scale weather systems such as cyclones, blocks and marine cold air outbreaks determine the sub-structure of extreme seasons?

4. What is the role of surface preconditioning , i.e., of early season anomalies of sea surface temperature and/or sea ice concentration for the formation of extreme seasons?

To address these research questions, a novel method will be introduced to determine the "unusualness" of a season, which we define based on a combination of various surface parameters. Our study is organized as follows: Data and methods are described in Section 2. Section 3 presents an overview of the seasonal variability of surface temperature, surface precipitation, and of the surface energy budget components. In Section 4 we define anomalous and extreme seasons in the Arctic based on seasonal anomalies of these parameters, and analyze their substructure in distinct Arctic sub-regions. Detailed analyses of three Arctic extreme seasons and the involved atmospheric synoptic-scale processes are presented in Section 5, followed by the main conclusions in Section 6.

 ## 2 Data and methods

### 2.1 ERA5 data

To perform a detailed analysis of Arctic extreme seasons, the ERA5 reanalysis dataset of the European Centre for Medium-Range Weather Forecasts (ECMWF) is used (Hersbach et al., 2020). Hourly atmospheric fields and short-range forecasts were spatially interpolated to a $0.5° \times 0.5°$ horizontal grid on model levels. The study period includes winters [December-February (DJF)] from 1979/80 to 2017/18 as well as springs [March-May (MAM)], summers [June-August (JJA)] and autumns [September-November (SON)] from 1980 to 2018. Based on the ERA5 dataset, we additionally consider synoptic features such as extratropical cyclones and blocks identified following the methods presented in Sprenger et al. (2017). Here, cyclones are defined as objects covering the area around a sea level pressure minimum, delimited by the outermost closed sea level pressure contour (Wernli and Schwierz, 2006). Blocks are identified based on the deviation of vertically averaged potential vorticity between 150 and 500 hPa from the monthly climatological mean. Contiguous areas where this value falls below $-1.3$ pvu are tracked in time and tracks that persist for at least 5 days are considered as blocks (Schwierz et al., 2004; Croci-Maspoli et al., 2007).

Further, we define marine cold air outbreaks (CAOs) based on the exceedance of the 900 hPa sea-air potential temperature difference ($\theta_{SST} - \theta_{900}$) by $+4$ K (cf. Papritz and Spengler, 2017), whereby we exclude grid points over land or with a sea ice concentration of more than 50 %. As outlined below, particularly anomalous seasons are identified based on seasonal-mean anomalies of the following six variables in specific regions: 2 m temperature ($T_{2m}$), precipitation ($P$, defined as the sum of convective and large-scale precipitation), surface sensible heat flux ($H_S$), surface latent heat flux ($H_L$), net surface shortwave radiation ($R_S$) and net surface longwave radiation ($R_L$). The last four variables are relevant for the surface energy balance and their sum is denoted by $E_S$. Positive signs denote energy fluxes into the surface, whereas negative signs are indicative of energy fluxes into the atmosphere. We use short-range forecasts for the fluxes $P$, $H_S$, $H_L$, $R_S$ and $R_L$ and analyses for the other fields.

| Abbreviation | Variable name | Unit |
|---|---|---|
| $E_S$ | sum of $H_S$, $H_L$, $R_S$ and $R_L$ | [W m$^{-2}$] |
| $H_L$ | surface latent heat flux | [W m$^{-2}$] |
| $H_S$ | surface sensible heat flux | [W m$^{-2}$] |
| $P$ | precipitation | [mm day$^{-1}$] |
| $R_L$ | net surface longwave radiation | [W m$^{-2}$] |
| $R_S$ | net surface shortwave radiation | [W m$^{-2}$] |
| SIC | sea ice concentration | |
| SST | sea surface temperature | [K] |
| $T_{2m}$ | 2 m temperature | [K] |

**Table 1.** List of variable names used in this study.

To compute anomalies, a transient climatology is calculated at every grid point as follows. First, daily-mean values of the variables are smoothed with a 21-day running mean filter. In a second step, the 9-year running mean is computed for each calendar day. Thus, the seasonal cycle is retained in the climatology, but decadal variations and long-term trends related to the overall warming of the Arctic are removed. The climatology is kept constant at the beginning and end of the study period when no 9-year running mean can be calculated. Examples of this filtering procedure are shown in the supplementary material, where Fig. S1a and b shows the original $T_{2m}$ time series in the Kara and Barents Seas and illustrates that the 9-year running mean can effectively eliminate also non-linear long-term trends ($T_{2m}$ in the Kara and Barents Seas steeply increases in the decade from 2000-2010). Seasonal-mean anomalies are then defined as deviations of the seasonal-mean values from this transient climatology. With this approach (also used by Messori et al. (2018) and Papritz (2020)), the identified extreme seasons appear relatively uniform throughout the study period (see Table 2 and Tables S3-S6 in the supplementary material). Throughout the study, we denote daily anomalies of a variable $\chi$ as $\chi^*$, seasonal-mean anomalies as $\overline{\chi^*}$ and seasonal-mean absolute anomalies as $\overline{|\chi^*|}$. Please note that in the case of absolute anomalies combined with spatial averaging, we first compute the absolute anomalies and then perform the spatial averaging.

## 2.2 Definition of sub-regions

Extreme seasons will be identified in three distinct geographical regions, the Nordic Seas (**NO**), endpoint of the Atlantic storm track (e.g., Wernli and Schwierz, 2006) and area of deep water formation (e.g., Dickson et al., 1996), the Kara and Barents Seas (**KB**), which are strongly affected by recent changes in sea ice concentration (e.g., Cavalieri and Parkinson, 2012), and the remaining Arctic poleward of 60° N (**AR**, containing the Arctic Ocean), which is to some extent dynamically de-coupled from the mid-latitudes. Grid points above land are excluded. It is one goal of this study to analyze the characteristics of Arctic extreme seasons with respect to climatological conditions. As the variables, especially the surface heat fluxes and surface radiation are strongly dependent on the surface conditions (e.g., Pope et al., 2020), the regions are additionally subdivided in each season

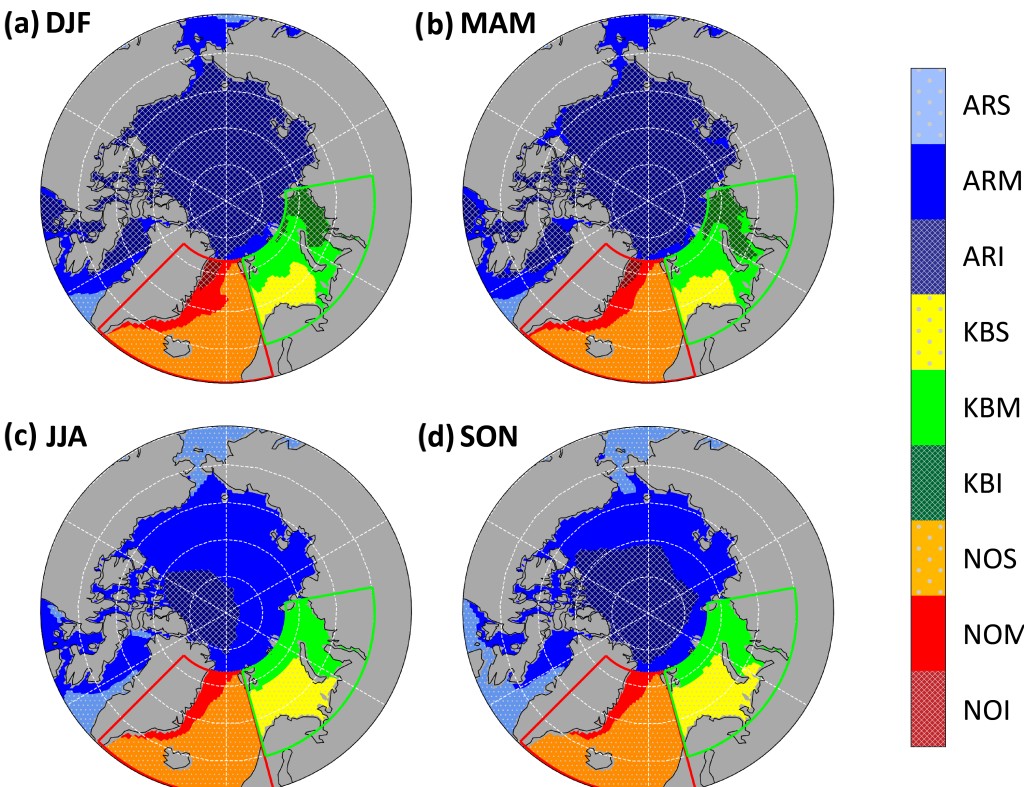

**Figure 1.** Sub-regions defined based on $SIC_{clim}$ in **(a)** DJF, **(b)** MAM, **(c)** JJA, and **(d)** SON. The labels refer to: **NOI**: Nordic Seas Ice, **NOM**: Nordic Seas Mixed, **NOS**: Nordic Seas Sea, **KBI**: Kara and Barents Seas Ice, **KBM**: Kara and Barents Seas Mixed, **KBS**: Kara and Barents Seas Sea, **ARI**: Arctic Residual Ice, **ARM**: Arctic Residual Mixed, **ARS**: Arctic Residual Sea. Green and red boxes denote the areas of the Kara and Barents Seas and Nordic Seas regions, respectively.

according to the climatological seasonal-mean sea ice concentration ($SIC_{clim}$). A distinction is made between areas where, on
all days of the considered season in the time period covered by this study, mainly sea ice is present (**Ice**, $SIC_{clim} > 0.9$), mainly
open ocean is present (**Sea**, $SIC_{clim} < 0.1$), and regions of intermediate $SIC_{clim}$ (**Mixed**, $0.1 \leq SIC_{clim} \leq 0.9$). Furthermore we
require a minimum size of a sub-region of $10^5$ km$^2$. With these criteria, three sub-regions are defined in each region, which
results in overall seven distinct sub-regions in JJA and SON and nine distinct sub-regions in DJF and MAM (Fig. 1). For ex-
ample, **ARM** denotes the sub-region with intermediate sea ice cover in the High Arctic and **NOS** the sub-region with mainly
open ocean in the Nordic Seas. In these sub-regions and based on the surface parameters listed above, anomalous and extreme
Arctic seasons are defined using a method based on principal component analysis (PCA) as detailed in the following.

## 2.3 Definition of anomalous and extreme seasons

To determine in an objective way whether a season is anomalous or extreme, a PCA is performed for each sub-region. For that purpose, the seasonal anomalies of the six variables (referred to as precursors) are standardized with their inter-seasonal standard deviation to ensure comparability and equal weighting of the different parameters. Here, the variables $H_S$, $H_L$, $R_S$ and $R_L$, which all contribute to the surface energy balance ($E_S$), are weighted by the maximum standard deviation of the four $E_S$ components, thus emphasizing variables contributing stronger to $E_S$ variability. We use the PCA to reduce the dimensionality of the six-parameter phase space to two dimensions by focusing on the first and second principal component ($\widetilde{PC1}$ and $\widetilde{PC2}$). $\widetilde{PC1}$ and $\widetilde{PC2}$ maximize the so-called "explained variance", which is the explained proportion of the total inter-seasonal variability in the six-dimensional phase space of the precursors.

To define extreme and anomalous seasons, $\widetilde{PC1}$ and $\widetilde{PC2}$ are first rescaled by their respective standard deviation ($\sigma_1$ and $\sigma_2$), such that outliers in both PCs are treated similarly independent of the variance explained by $\widetilde{PC1}$ and $\widetilde{PC2}$, thus providing a measure for the unusualness of each season with respect to each of the principal components (from now on, we will refer to these rescaled components as $PC1$ and $PC2$). Then, the Euclidian distance in the reduced phase space spanned by the two rescaled components, the so-called "Mahalanobis distance" ($d_M$), is calculated as:

$$d_M = \sqrt{PC1^2 + PC2^2} = \sqrt{\frac{\widetilde{PC1}^2}{\sigma_1^2} + \frac{\widetilde{PC2}^2}{\sigma_2^2}}. \tag{1}$$

This measure $d_M$ can now be used to quantify how strongly a particular season deviates from climatology, representing the combination of the seasonal anomalies of the six variables. We therefore refer to $d_M$ as "anomaly magnitude" of a particular season. Seasons with $d_M \geq 3$ are defined as "extreme seasons", and seasons with $3 > d_M \geq 2$ as "anomalous seasons".

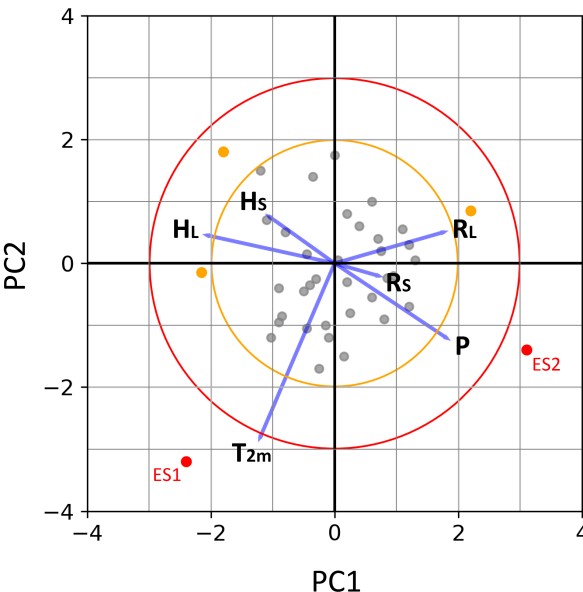

**Figure 2.** Schematic PCA biplot with PC1 along the x-axis and PC2 along the y-axis. Grey dots represent single seasons, red (orange) dots show extreme (anomalous) seasons. Blue lines represent the projections of the original parameters onto the first two principal components. Values of $d_M$=2 and $d_M$=3 are shown by orange and red circles, respectively.

The phase space of the rescaled principal components can be illustrated using a biplot (Fig. 2), similar as in Graf et al. (2017). The axes of such a plot represent PC1 and PC2, respectively, and each dot represents one season in the study period, whereby anomalous and extreme seasons are shown as colored dots. The closer two dots are, the more similar are the anoma-

200 lies of the corresponding seasons. Radial vectors show the relative contribution of the precursor variables to PC1 and PC2, whereby the projected values of a vector on both axes illustrates the weight on the respective PC. In the case shown in Fig. 2, the vector for $T_{2m}$ is mainly aligned along PC2, thus $T_{2m}$ variability is important for the second principal mode of variability in the six-dimensional phase space. Relatively longer (shorter) vectors indicate a larger (smaller) contribution of the precursor to the explained variance. If two vectors are approximately perpendicular, the precursors are uncorrelated. This interpretation

of correlations is more precise, the higher the explained variance by PC1 and PC2 (Gabriel, 1971, 1972). The relative position of each season in the biplot (i.e., the scores) with respect to the precursor vectors indicates the contribution of the different precursor variables to the anomaly magnitude $d_M$ in the considered season. For instance, seasons with a positive $T_{2m}$ anomaly are positioned in the direction of the $T_{2m}$ vector and seasons with a negative $T_{2m}$ anomaly in the opposite direction.

In the example given in Fig. 2, the variables $T_{2m}$ and $P$ show no correlation, whereas $H_S$ and $H_L$ are positively correlated and $H_S$ and $P$ are strongly anti-correlated. Further, $T_{2m}$ shows the largest contribution to the variance explained by PC1 and

PC2 (mainly determining PC2) whereas $H_L$, $R_S$ and $R_L$ mostly contribute to PC1. $R_S$ contributes the least to the explained variance. Two seasons with $d_M \geq 3$ are marked as extreme season 1 (ES1) and extreme season 2 (ES2). Their score vectors are roughly orthogonal to each other, which indicates that a different combination of anomalies and thus different processes are decisive for explaining their large anomaly magnitudes. In this example, ES1 is mainly determined by a positive $T_{2m}$ anomaly, while ES2 is an anomalously wet season with negative surface heat flux anomalies, as the respective precursor vectors are directed more or less directly towards ($P$) respectively away ($H_L$, $H_S$) from ES2.

## 3   Spatial and temporal variability of Arctic seasons

In order to characterize Arctic seasons in general, we first analyze the co-variability of seasonal-mean anomalies of surface temperature ($\overline{T_{2m}}^*$), precipitation ($\overline{P^*}$) and surface energy balance ($\overline{E_S}^*$) in the three regions, considering the varying surface conditions of the different sub-regions (Fig. 3). We are interested in correlations between the seasonal anomalies, how their magnitudes vary between the regions, and in aspects of the seasonal substructure (e.g., is an anomalously warm season constantly warm?). Here, correlations with a p-value below 0.05 are defined as statistically significant.

In winter, warm seasons are generally wetter and cold seasons are drier (Fig. 3a-c), except for sub-regions **NOS** and **KBS** (see also Table S1 in the supplement). In regions with $\mathrm{SIC_{clim}} > 0.9$ (Fig. 3a), $\overline{T_{2m}}^*$ and $\overline{P^*}$ are strongly positively correlated, thus warm winters are almost always (in 79.8 % of the cases) wet and tend to have a positive $E_S$ anomaly (and again vice versa for cold winters). In contrast, regions with $\mathrm{SIC_{clim}} < 0.1$ in the Nordic Seas and Kara and Barents Seas do either show a weak negative or no significant correlation between $\overline{T_{2m}}^*$ and $\overline{P^*}$ (Fig. 3c). Over the open ocean, warm winters show strongly positive, and cold winters negative $\overline{E_S}^*$ values. Regions with intermediate sea ice extent (Fig. 3b) do not show this correlation between $\overline{T_{2m}}^*$ and $\overline{E_S}^*$, but warm winters tend to be wet and cold winters dry, similar to the **ice** sub-regions.

In summer, no correlation between $\overline{T_{2m}}^*$ and $\overline{P^*}$ is found, but 73.3 % of the warm summers show a positive $\overline{E_S}^*$ and 72.5 % of the cold summers a negative $\overline{E_S}^*$, independent of the surface conditions (Fig. 3d-f and supplementary Table S2). Regional differences are much smaller during summer, indicating more homogeneous conditions among the sub-regions.

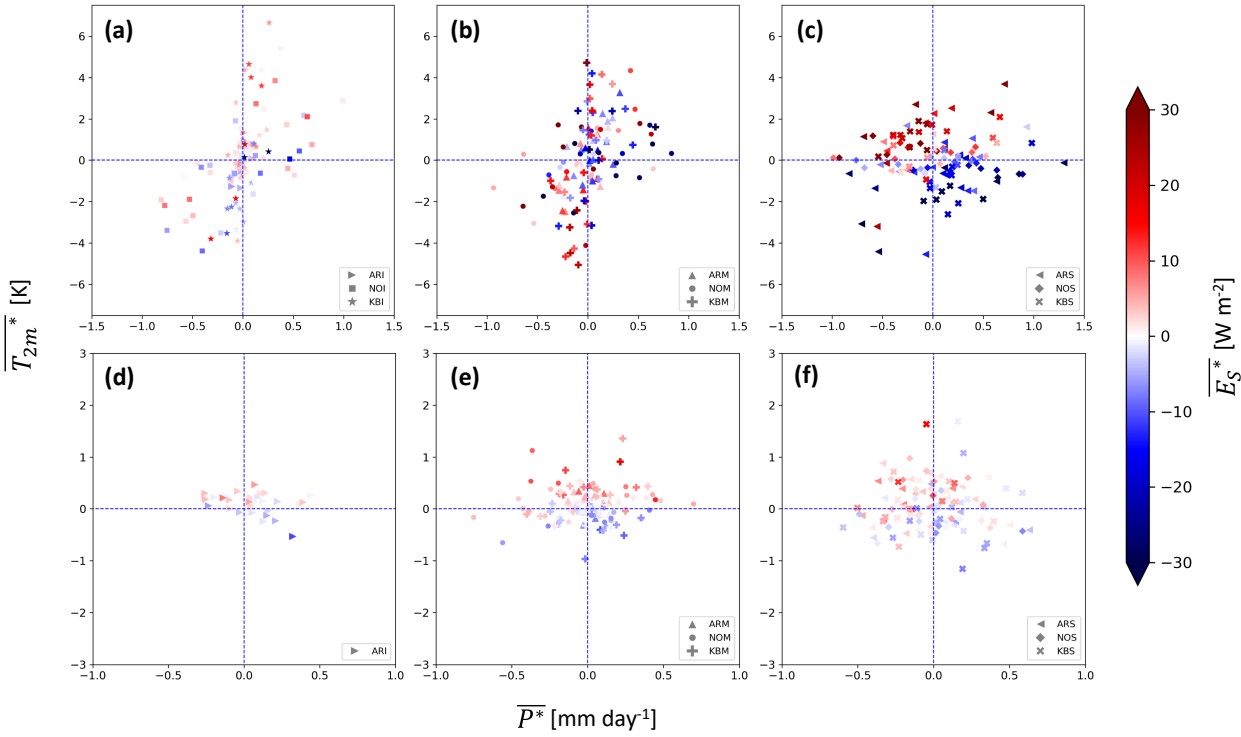

**Figure 3.** Seasonal-mean anomalies of $P$ ($\overline{P^*}$; $\mathrm{mm\,day^{-1}}$, along x-axis), $T_{2m}$ ($\overline{T_{2m}^*}$; K, along y-axis) and $E_S$ ($\overline{E_S^*}$; $\mathrm{W\,m^{-2}}$, color) for 39 seasons in **(a-c)** DJF and **(d-f)** JJA for sub-regions with **(a, d)** $\mathrm{SIC_{clim}} > 0.9$, **(b, e)** $0.1 \leq \mathrm{SIC_{clim}} \leq 0.9$ and **(c, f)** $\mathrm{SIC_{clim}} < 0.1$. Tables S1 and S2 in the supplement show correlations and respective p-values for the described relations between the different parameters in each sub-region.

In addition to the previously discussed seasonal-mean anomalies, the intra-seasonal variability of the individual parameters is an important and complementary characteristic of Arctic seasons. As we will show in the following, the strength of the intra-seasonal variability can depend, in particular, on the surface conditions. To compare the intra-seasonal variability of $T_{2m}^*$, $P^*$ and $E_S^*$, we consider in Fig. 4 seasonal-mean absolute anomalies $\overline{|T_{2m}^*|}$, $\overline{|P^*|}$ and $\overline{|E_S^*|}$, which are defined as the seasonal mean of the spatially-averaged absolute daily anomalies. They are used as a measure for the overall spatio-temporal variability of the individual parameters throughout a season. Distinct clusters occur for the different sub-regions in winter (Fig. 4a). Regions mostly over sea ice show only small variations in $E_S^*$ and $P^*$, except for **NOI**, implying a relatively small amplitude of day-to-day and inter-seasonal fluctuations of these variables. In the Kara and Barents Seas, sub-regions **KBI** and **KBM** show high variability in daily and seasonal $T_{2m}$ anomalies but a similarly small $\overline{|P^*|}$. Sub-regions over the open ocean, where $T_{2m}$ anomalies are typically smaller and less variable, show smaller values of $\overline{|T_{2m}^*|}$. $P$ and especially $E_S$ variability is strongly enhanced over the open ocean due to intensified air-sea interaction. The clear distinction of the seasonal-mean absolute anomalies between the different sub-regions reveals the spatial inhomogeneity of Arctic meteorological conditions in winter, which

is due to varying surface conditions as well as differences in seasonal variability between distinct Arctic Seas. This also serves as an a posteriori confirmation of our approach to separately consider Arctic extreme seasons in these sub-regions.

In summer, the variability of the three analyzed parameters is smaller due to reduced meridional gradients of surface temperature and radiation causing smaller $T_{2m}$ and $E_S$ fluctuations (Fig. 4b). Similar to winter, sub-regions over the open ocean show a relatively large $\overline{|E_S{}^*|}$ and a larger variability of $P$ can be observed in the Nordic Seas compared to the Kara and Barents Seas, probably due to reduced moisture availability in the latter region. However, as the surface conditions between the sub-regions

become more homogeneous, the regions do appear in less distinct clusters as for winter with the exception of the sub-region **ARI** which covers most of the perennial sea ice and shows, as in winter, only a small variability of the three parameters. It is further noteworthy that $\overline{|T_{2m}{}^*|}$ and $\overline{|P^*|}$ are positively correlated, indicating a larger $\overline{|P^*|}$ in summer seasons where $T_{2m}$ fluctuates more.

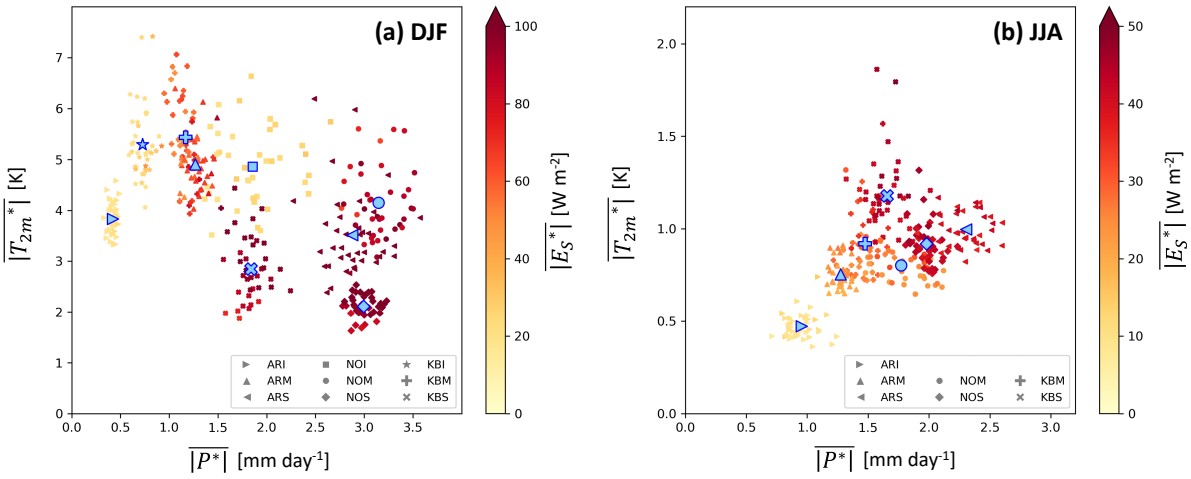

**Figure 4.** Seasonal-mean absolute anomalies of $P$ ($\overline{|P^*|}$; mm day$^{-1}$, along x-axis), $T_{2m}$ ($\overline{|T_{2m}{}^*|}$; K, along y-axis) and $E_S$ ($\overline{|E_S{}^*|}$; W m$^{-2}$, color) for 39 seasons and **(a)** 9 sub-regions in DJF and **(b)** 7 sub-regions in JJA. Blue symbols mark average values for each sub-region.

To better understand the seasonal substructure of Arctic winters and summers, we compare the seasonal-mean anomalies

with the seasonal-mean absolute anomalies for $T_{2m}$, $P$ and $E_S$ in selected sub-regions in DJF (Fig. 5) and JJA (Fig. 6; for remaining sub-regions see supplementary Figs. S2 and S3). The ratio of seasonal-mean and seasonal-mean absolute anomalies, $\frac{\overline{\chi^*}}{\overline{|\chi^*|}}$, is indicative of the temporal persistency of an anomaly throughout a season. Thus, the location of a season in the diagrams provides information about the substructure of the season in terms of the considered parameter. In general, the further to the right, the more positive is the seasonal-mean anomaly of the shown parameter and the further to the left, the more negative. The

closer the seasonal-mean anomaly is to the seasonal-mean absolute anomaly (dots close to the outer stippled grey line representing $\frac{\overline{\chi^*}}{\overline{|\chi^*|}} = \pm 1$), the more persistent the anomaly is throughout the season. Thus, we define seasons with $0.8 \leq \left| \frac{\overline{\chi^*}}{\overline{|\chi^*|}} \right| \leq 1$ as

seasons with a "continuous" anomaly. With a smaller value of $\frac{\overline{\chi^*}}{\overline{|\chi^*|}}$, the seasons are located further away from the outer stippled grey lines, meaning that positive or negative anomalies in the respective parameter occur more episodically throughout the season. The closer a season is positioned towards the blue dashed line where $\overline{\chi^*}=0$ and thus $\frac{\overline{\chi^*}}{\overline{|\chi^*|}}=0$, the more positive and

negative daily anomalies cancel each other, leading to a weak overall seasonal anomaly. The value of $\overline{|\chi^*|}$ is further indicative of the magnitude of the daily anomalies throughout a season. A season located at the top of the plot shows stronger daily anomalies than a season with the same $\frac{\overline{\chi^*}}{\overline{|\chi^*|}}$ ratio but a smaller $\overline{|\chi^*|}$.

     For example, a season can be anomalously warm, because the daily-mean $T_{2m}$ values are larger than the climatology on

almost all days of the season, resulting in $\frac{\overline{T_{2m}^*}}{\overline{|T_{2m}^*|}}\approx1$. With a decreasing ratio of the anomaly metrics, e.g., $\frac{\overline{T_{2m}^*}}{\overline{|T_{2m}^*|}}=0.5$, the season is still anomalously warm, but it results from several warm episodes alternating with weaker and/or shorter periods with negative $T_{2m}^*$ values. If $\frac{\overline{T_{2m}^*}}{\overline{|T_{2m}^*|}}\approx0$, cold and warm episodes cancel each other leading to a weak overall seasonal anomaly. Comparing seasons with the same $\frac{\overline{T_{2m}^*}}{\overline{|T_{2m}^*|}}$, the ones positioned further along the y-axis (showing larger values of $\overline{T_{2m}^*}$ and $\overline{|T_{2m}^*|}$) show a larger variability in $T_{2m}$ with more intense warm and/or cold episodes.

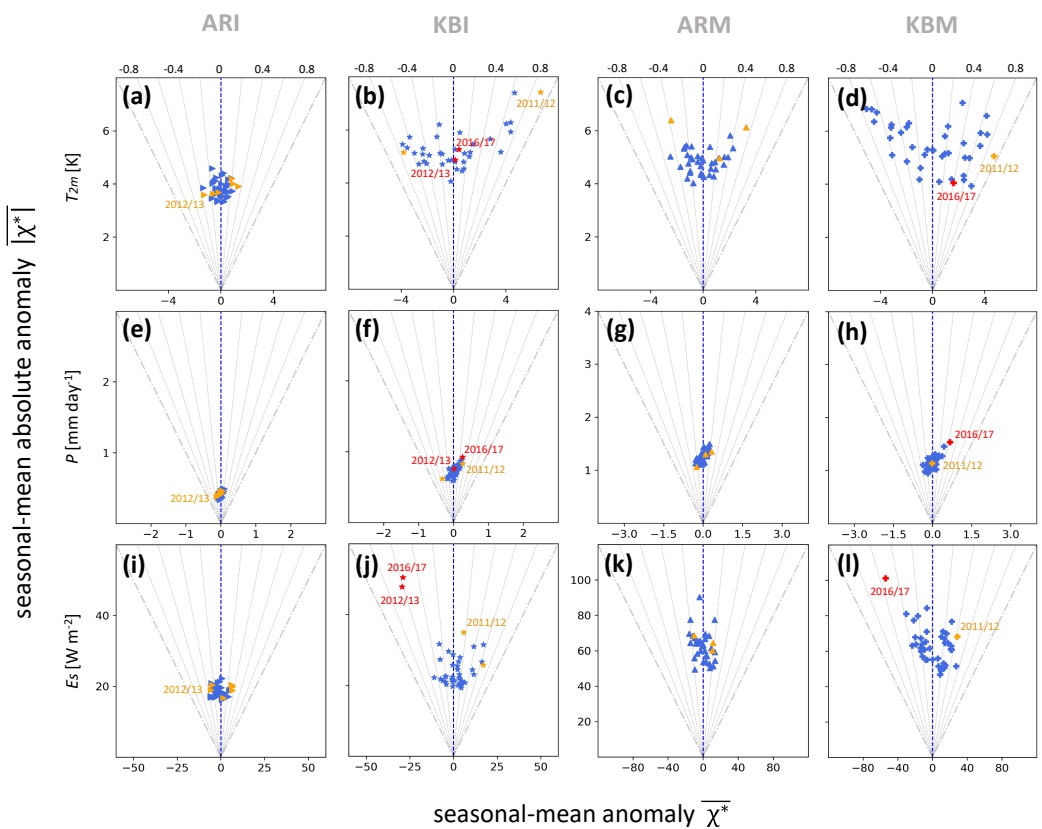

**Figure 5.** Seasonal-mean anomalies ($\overline{\chi^*}$, along x-axis) vs. seasonal-mean absolute anomalies ($\overline{|\chi^*|}$, along y-axis) in DJF for **(a-d)** $T_{2m}$ (K), **(e-h)** $P$ (mm day$^{-1}$), and **(i-l)** $E_S$ (W m$^{-2}$) in sub-regions **ARI**, **KBI**, **ARM** and **KBM**. The ratio of both measures, $\frac{\overline{\chi^*}}{\overline{|\chi^*|}}$, is additionally visualized by grey dashed lines. $\frac{\overline{\chi^*}}{\overline{|\chi^*|}} = \pm 1$ is shown by stippled grey lines and $\frac{\overline{\chi^*}}{\overline{|\chi^*|}} = 0$ is shown by a dashed blue line. Red (orange) markers represent extreme (anomalous) seasons (see Sect. 2.3) and selected case study seasons are labeled. Remaining sub-regions are shown in supplementary Fig. S2.

The seasonal substructures of the three parameters differ. In particular during summer, several seasons show continuous $T_{2m}{}^*$ (Figs. 6a-d and S3a-c), including several clear outlier seasons (Figs. 6c, d and S3a). In winter, the overall $T_{2m}$ variability is much larger and only very few seasons show distinct $\overline{T_{2m}{}^*}$ outliers (Figs. 5a-d and S2a-e). Further, no continuous $P^*$ can be observed (Figs. 5e-h and 6e-h, resp. Figs. S2f-j and S3d-f), indicating that even in very wet seasons precipitation is episodic and includes dry periods. In addition, and maybe less obvious, also the driest seasons feature some precipitation events. The tilted shape of the scatter plots for $P$ indicates that wet seasons tend to show a larger $\overline{|P^*|}$ than dry seasons, resulting from the skewness of the parameter $P$. For $E_S$, in the sub-regions **NOM**, **KBM** and **KBS**, different distributions of the anomalies occur in DJF and JJA. Winters with a negative $\overline{E_S{}^*}$, which is often caused by several episodes of cold air outbreaks (Papritz and Spengler, 2017), tend to show enhanced $E_S$ variability throughout the season (Figs. 5l and S2l, o) compared to winters with a

positive $\overline{E_S}^*$, where CAOs are less frequent. The opposite occurs in summer, when periods of increased net surface radiation
can cause a positive $\overline{E_S}^*$ and enhanced $\overline{|E_S}^*|$ compared to seasons with a negative $\overline{E_S}^*$ (Fig. 6i, k and S3g).

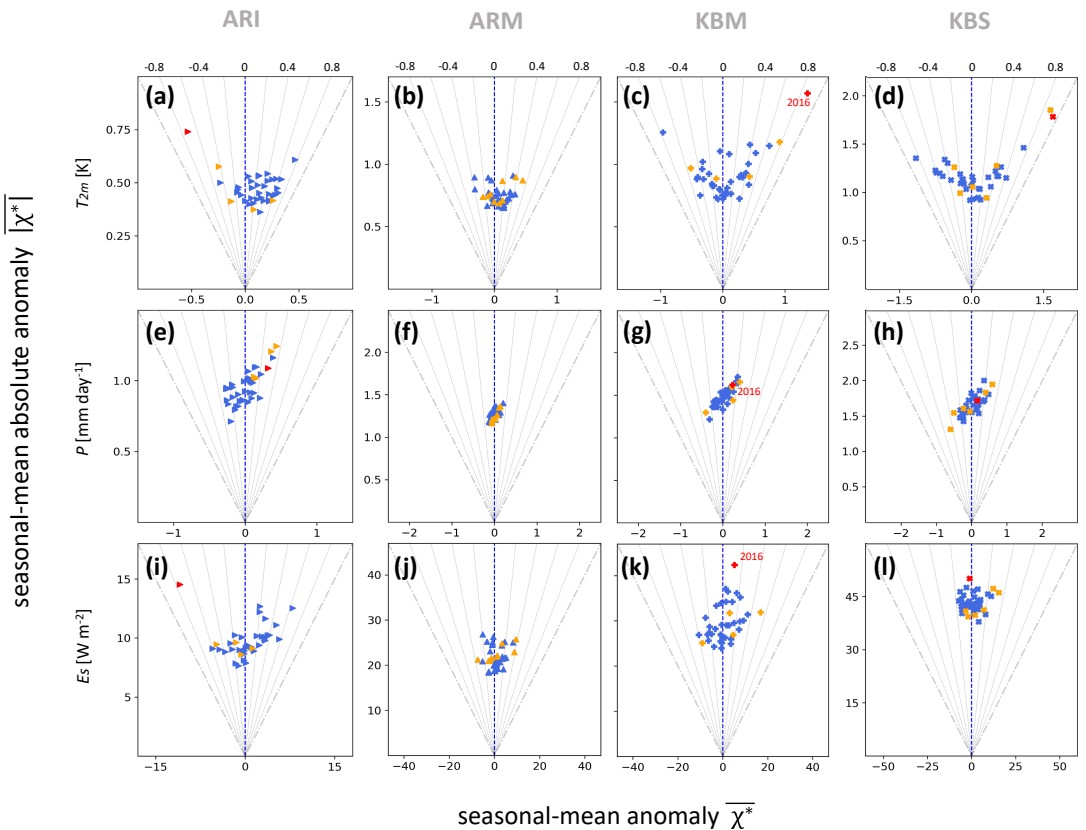

**Figure 6.** Same as Fig. 5 but for JJA in sub-regions **ARI**, **ARM**, **KBM** and **KBS**. The extreme summer 2016 is labeled in **(c, g, k)** due to its role in the preconditioning of the extreme winter 2016/17 (see subsection 5.2). Remaining sub-regions are shown in supplementary Fig. S3.

## 4 PCA results

In the previous section, we discussed the co-variability of $T_{2m}$, $P$ and $E_S$ and regional differences for Arctic winter and summer seasons as well as the seasonal substructure of these parameters. In a next step, we identify and then characterize anomalous Arctic seasons, based on the combination of the seasonal-mean anomalies of the three surface parameters. To this end, a PCA is performed for each season (DJF, MAM, JJA, SON) and sub-region, as explained in Sect. 2.3. Figures 7 and 8 show the resulting biplots for DJF and JJA (for MAM and SON see Figs. S5 and S6 in the supplementary material). Depending on the region and sub-region, the contributions of the precursor variables to the principal components PC1 and PC2 vary, which usually explain about $80\% - 90\%$ of the total variance in the combined seasonal anomalies. Further we can show that,

following the method introduced in North et al. (1982), the first two PCs in DJF and JJA are, with exception of sub-region

**ARM** in JJA, always statistically distinguishable from the others (see supplementary Fig. S4). This implies that by considering PC1 and PC2, we capture most of the variance.

In winter, sub-regions over ice show a positive correlation between $\overline{T_{2m}}^*$ and $\overline{P^*}$ (Fig. 7a, d, g). This correlation is particularly strong in the High Arctic, where precipitation events during winter are predominantly caused by synoptic weather systems

that transport warm and moist air masses into the region (e.g., Webster et al., 2019; Papritz and Dunn-Sigouin, 2020). $\overline{T_{2m}}^*$, $\overline{P^*}$ and $\overline{R_L}^*$ mainly determine PC1 and thus the direction of maximum variance in the phase space spanned by all precursor variables in **ice** sub-regions. Surface sensible and latent heat flux anomalies are positively correlated and mostly uncorrelated with $\overline{T_{2m}}^*$ and $\overline{P^*}$ as they contribute mostly to PC2.

Similarly, sub-regions with intermediate sea ice concentration show a positive correlation of $\overline{T_{2m}}^*$ and $\overline{P^*}$ (Fig. 7b, e, h), although slightly weaker than over ice for regions **KB** and **NO**. Again, the heat fluxes are mostly uncorrelated with $\overline{T_{2m}}^*$ and slightly negatively correlated with $\overline{P^*}$, particularly $\overline{H_L}^*$. $\overline{R_L}^*$ is contributing less to the variance in **mixed** regions, which indicates a comparatively lower importance of radiation compared to heat fluxes for determining the seasonal variability.

Over the open ocean (Fig. 7c, f, i), a positive correlation between the heat flux anomalies and $\overline{T_{2m}}^*$ can be observed, indicating increased surface fluxes from the ocean into the atmosphere during periods with anomalously cold temperatures. Unlike over ice, the maximum variance over open water is mainly determined by the surface heat fluxes. $\overline{P^*}$ is mostly uncorrelated to the other variables and strongly related to PC2, reflecting that precipitation can occur in warm conditions (e.g., warm sector of a cyclone) and in cold conditions (CAO).


Arctic summer seasonal variability is mainly determined by $\overline{T_{2m}}^*$, $\overline{P^*}$ and $\overline{R_S}^*$, whereby $\overline{T_{2m}}^*$ and $\overline{P^*}$ are mostly uncorrelated in all regions (Fig. 8). Whereas $\overline{T_{2m}}^*$ shows only weak correlations with other parameters in general, $\overline{P^*}$ is strongly anti-correlated with $\overline{R_S}^*$ in sub-regions **NOS** and **ARS** (Fig. 8f and i), most likely due to the presence of clouds during precipitation events. In sub-regions **ARI** and **ARM** (Fig. 8g and h), $\overline{R_L}^*$ additionally influences the seasonal variability and strongly

correlates with $\overline{T_{2m}}^*$, again emphasizing the importance of clouds in this region.

| Season | Sub-regions | $d_M$ | $\overline{T_{2m}}^*$ [K] | $\overline{P}^*$ [mm day$^{-1}$] | $\overline{E_S}^*$ [W m$^{-2}$] | Area [$10^5$ km$^2$] |
|---|---|---|---|---|---|---|
| **DJF 2004/05** | ARS | 3.2 | −0.13 *(−0.07)* [21−] | +1.30 *(+2.73)* **[1+]** | −47.10 *(−2.00)* **[1−]** | 3.5 |
| **DJF 2012/13** | KBI | 3.0 | +0.13 *(+0.05)* [18+] | +0.02 *(+0.12)* [19+] | −29.48 *(−3.07)* **[1−]** | 5.4 |
| **DJF 2016/17** | KBI | 3.4 | +0.41 *(+0.16)* [16+] | +0.25 *(+1.96)* [2+] | −29.00 *(−3.02)* [2−] | 5.4 |
| | KBM | 3.3 | +1.61 *(+0.59)* [13+] | +0.67 *(+3.44)* **[1+]** | −54.20 *(−3.03)* **[1−]** | 10.7 |
| | KBS | 3.1 | +0.83 *(+0.65)* [14+] | +0.98 *(+3.03)* **[1+]** | −19.14 *(−0.81)* [8−] | 6.8 |
| **JJA 2013** | ARI | 3.2 | −0.53 *(−2.90)* **[1−]** | +0.32 *(+1.75)* [4+] | −10.9 *(−3.21)* **[1−]** | 14.3 |
| **JJA 2016** | KBM | 3.1 | +1.36 *(+3.27)* **[1+]** | +0.23 *(+1.18)* [7+] | +0.52 *(+0.91)* [8+] | 11.6 |
| | NOM | 3.3 | +1.13 *(+3.47)* **[1+]** | −0.37 *(−1.10)* [8−] | +9.27 *(+1.65)* [4+] | 5.1 |
| **MAM 1990**[1] | NOI | 3.7 | +0.37 *(+0.27)* [17+] | −0.16 *(−0.37)* [16−] | −1.51 *(−0.47)* [12−] | 1.3 |
| | ARI | 4.1 | +3.08 *(+3.45)* **[1+]** | +0.25 *(+3.81)* **[1+]** | +2.36 *(+1.23)* [4+] | 80.9 |
| **MAM 1996** | NOM | 3.3 | +1.32 *(+1.20)* [6+] | −0.54 *(−1.25)* [4−] | +33.96 *(+2.69)* **[1+]** | 5.8 |
| **SON 1995** | KBM | 3.0 | +0.68 *(+0.42)* [16+] | −0.06 *(−0.27)* [15−] | −27.11 *(−2.99)* **[1−]** | 10.7 |
| **SON 2007** | ARM | 3.3 | +1.33 *(+1.40)* [8+] | +0.06 *(+0.78)* [11+] | −15.2 *(−3.47)* **[1−]** | 52.7 |
| **SON 2018** | ARI | 3.2 | +1.63 *(+1.36)* [4+] | +0.31 *(+3.07)* **[1+]** | −2.76 *(−1.60)* [4−] | 32.3 |

**Table 2.** Extreme seasons in DJF, JJA, MAM and SON, including the affected sub-regions and respective Mahalanobis distance ($d_M$, see Sect. 2.3), the seasonal-mean anomalies of $T_{2m}$, $P$ and $E_S$ (standardized seasonal-mean anomalies in parentheses) and affected area per sub-region. The rank of each seasonal-mean anomaly with respect to all seasons is given in brackets, with "1+" denoting rank 1 in terms of a positive anomaly (e.g., wettest season) and "1−" denoting rank 1 in terms of a negative anomaly (e.g., driest season).

## 4.1 Arctic extreme and anomalous seasons

As explained in Sect. 2.3, by using a threshold for the anomaly magnitude ($d_M \geq 3$), seasons that appear as clear outliers in their respective PCA biplot are defined as extreme seasons, whereas seasons located at the edges of the point cloud formed by all seasons are characterized as anomalous seasons ($3 > d_M \geq 2$). The two thresholds are chosen pragmatically to distinguish seasons with different anomaly magnitudes and to classify the season with the largest anomaly magnitude as "extreme season". Using these thresholds we find 2 extreme seasons in DJF, JJA and MAM, respectively, and 3 extreme seasons in SON (Table 2).

---

[1]Extreme season MAM 1990 in sub-region **NOI** shows rank **[1+]** for $R_S$, rank [7+] for $H_S$, rank [9−] for $H_L$ and rank **[1−]** for $R_L$. Although single components of $E_S$ show rank 1, but in opposite directions, this leads to an overall medium rank [12−] for $E_S$.

With this total number of extreme seasons, the return period of such a season corresponds to approximately 40 years, which has been used as an adequate measure for defining extreme seasons by several studies, e.g., Röthlisberger et al. (2021). The

number of sub-regions where one particular season is identified as extreme varies between one and three, however the varying size of the sub-regions and thus significant differences in the extent of the affected area have to be considered. Further we identify on average 3.3 anomalous seasons per sub-region in DJF, 5 anomalous seasons per sub-region in JJA, 4.7 anomalous seasons per sub-region in MAM and 4.4 anomalous seasons per sub-region in SON (see supplementary Tables S3-S6).

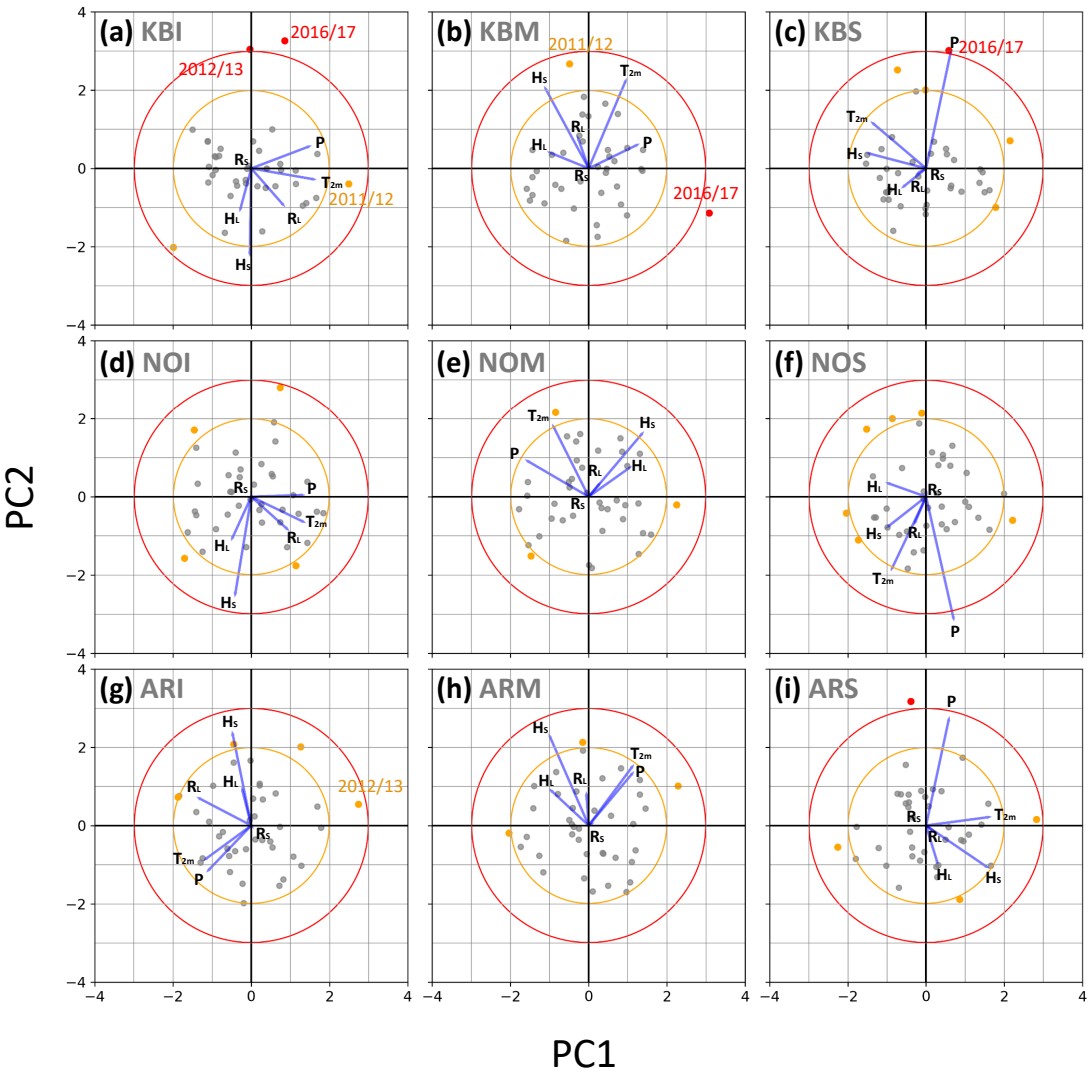

**Figure 7.** PCA biplot for all sub-regions in DJF with PC1 and PC2 along the x- and y-axis, respectively. Every season is represented by a grey dot, red and orange dots show extreme and anomalous seasons, respectively. Blue lines represent the coefficients of the precursor variables. Red and orange circles represent $d_M$=3 and $d_M$=2, the thresholds used for extreme and anomalous seasons, respectively. Selected case study seasons are labeled.

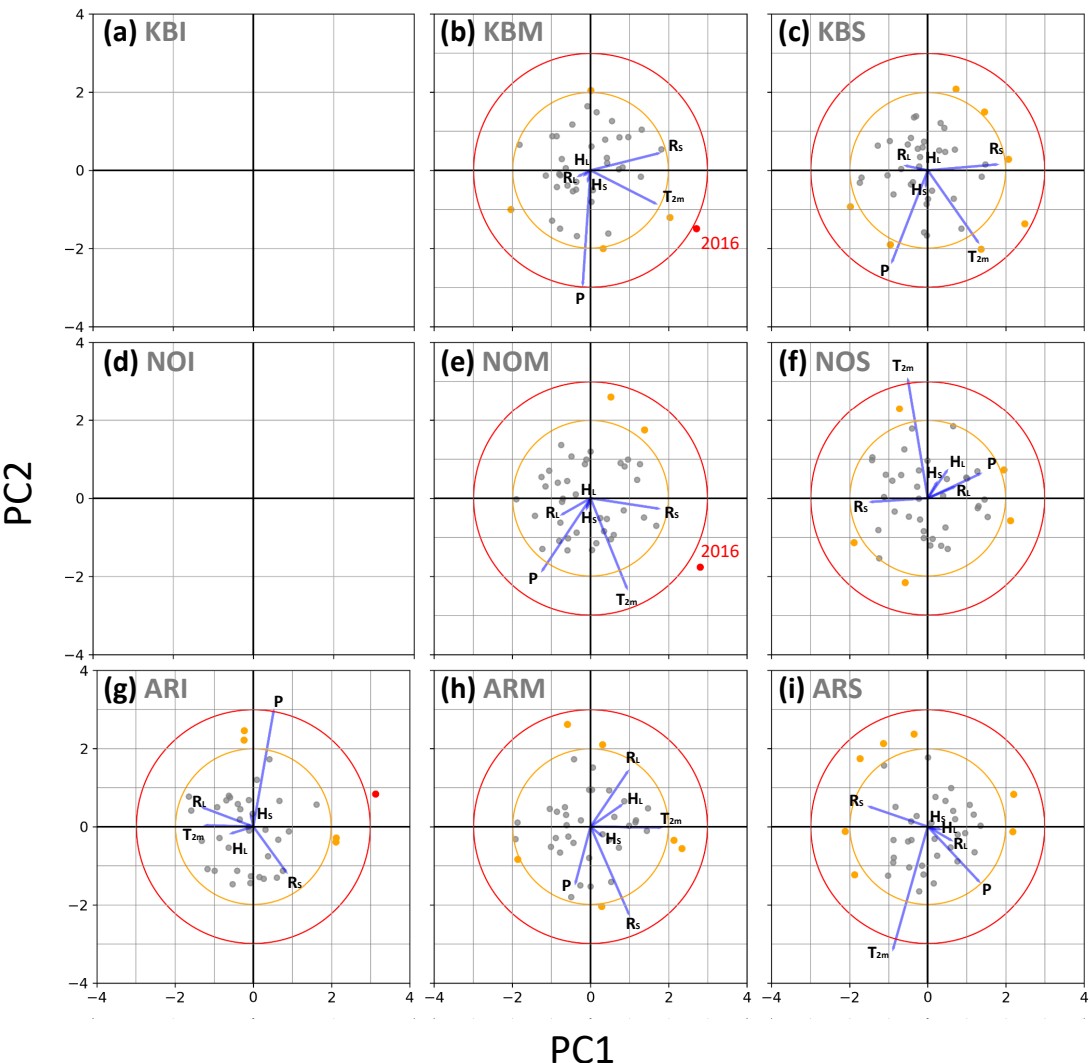

**Figure 8.** As Fig. 7 but for JJA. No biplots are shown for the sub-regions **KBI** and **NOI**, because they fall below the minimum size threshold of $10^5$ km$^2$ in summer. The extreme summer 2016 is labeled in **(b, e)** due to its role in the preconditioning of the extreme winter 2016/17 (see subsection 5.2).

After identifying Arctic extreme and anomalous seasons as well as the surface parameters determining their variability in the different sub-regions, we are now interested in the substructure of such seasons with respect to $T_{2m}$, $P$ and $E_S$. Therefore, we briefly reconsider Figs. 5 and 6 and focus on the extreme and anomalous seasons, shown by red and orange dots, respectively, in comparison to all seasons in the study period. By design, extreme seasons with $d_M \geq 3$ have very large anomalies for at least one parameter (see ranks for seasonal-mean anomalies in Table 2), for example the strong positive $\overline{P^*}$ in the extreme winter

in **KBM** (Fig. 5h) or the negative $\overline{E_S}^*$ in both extreme winters in **KBI** (Fig. 5j). In summer, all extreme seasons are characterized by a strong $\overline{T_{2m}}^*$ outlier (Fig. 6a, c, d), which coincides with $\overline{E_S}^*$ outliers in **ARI** and **KBM** (Fig. 6i, k). Similarly, most anomalous seasons also show outliers or anomalies near the edge of the point cloud for at least one parameter. However, some anomalous seasons do not show very strong anomalies in one particular parameter, which implies that for these seasons it is the combination of several parameters that makes them anomalous. In a given region, several extreme or anomalous seasons can have similar seasonal anomalies, for instance both extreme winters in **KBI** (Fig. 5b, f, j) and two anomalous and one extreme summers in **ARI** (Fig. 6a, e, i), indicating similar characteristics and most likely also underlying processes causing the anomalous nature of these seasons. However, in other regions with multiple anomalous seasons, they show a similar behavior in one but a contrasting behavior in another variable. For example, the anomalous and extreme winters in **KBM** both have a positive $\overline{T_{2m}}^*$ but different signs in their respective $\overline{E_S}^*$ (Fig. 5d and l). We thus expect different processes to be responsible for these seasons to be anomalous.

Based on the results of the PCA analysis and Fig. 5, the following winter seasons are chosen for detailed case studies to better understand their seasonal substructure as well as the underlying processes: The winters 2011/12 and 2016/17 in the Kara and Barents Seas and the winter 2012/13 in **ARI**. We do not consider the extreme winter in **ARS** in 2004/05, as **ARS** is only a very small region that consists of two remote fragments and thus the meaningful analysis of the involved processes would be less straightforward. Furthermore this selection allows to, on one hand, contrast two seasons in the same geographical region, and on the other hand also point out differences in terms of the underlying processes in a region at the edge of the Arctic and in the High Arctic. This choice of case study seasons is subjective and motivated by the intention to reveal the diversity and complexity of the involved processes. It is further strongly limited by the available amount of suitable seasons for in-depth investigation. Choosing two winter seasons in the same region allows us to emphasize inter-annual variability, while avoiding additional effects of seasonal variations.

## 5  Case Studies

### 5.1  DJF 2011/12

The winter of 2011/12 is classified as an anomalous season in **KBI** and **KBM**. In both sub-regions, this winter shows the largest positive $\overline{T_{2m}}^*$ during the 39-year study period (Fig. 5b and d). The time series in Fig. 9a shows that the daily-mean surface temperature is continuously above the climatology in the **KB** region (consistent with the fact that the dots in Fig. 5b and d are close to the diagonal). In **KBI**, $\overline{T_{2m}}^*$ is the main contributor to this season's anomaly magnitude, supported by positive $\overline{P^*}$ and $\overline{R_L}^*$ (Figs. 5b and f, and 7a). In **KBM**, positive $\overline{T_{2m}}^*$ and $\overline{H_S}^*$ mainly determine the exceptional character of this winter (Figs. 5d and 7b), which also leads to one of the most positive $\overline{E_S}^*$ compared to all winters in the study period (Fig. 5l).

In DJF 2011/12, $\overline{T_{2m}}^*$ is $+6.6\,\text{K}$ in **KBI** and $+4.7\,\text{K}$ in **KBM**. In the whole region, during December, $T_{2m}$ values are continuously around $+6\,\text{K}$ above climatology, before approaching more average levels at the beginning of January (Fig. 9a).

The largest $T_{2m}^*$ values are reached in February. The SIC anomaly shows an opposite behavior and is continuously negative, reaching values close to climatology only at the beginning of the season and during the period with reduced $T_{2m}^*$ in January

(Fig. 9c). Similarly to the other variables, here we calculate the SIC anomaly using a transient climatology, as this effectively removes non-linear SIC trends in the Kara and Barents Seas (see Fig. S1c and d in the supplement). Daily-mean $E_S$ values are strongly correlated with daily-mean $T_{2m}$, resulting in mostly positive $E_S^*$ during the particularly warm episodes and shorter periods of negative $E_S^*$ when $T_{2m}^*$ is reduced (Fig. 9b). The positive $\overline{E_S^*}$ is mainly due to a strongly positive $\overline{H_S^*}$, i.e., strongly reduced heat fluxes into the atmosphere, favored by the warm surface temperatures and comparatively few CAOs (see

next paragraph). During the period with the largest $T_{2m}^*$ in February, when the surface air temperatures exceed $0\,°C$ at several grid points on multiple days, even positive $H_S$ values occur over the open ocean (not shown). Daily $P$ values show only small deviations from climatology, except for the first five days of the season and in the beginning of February (Fig. 9d).

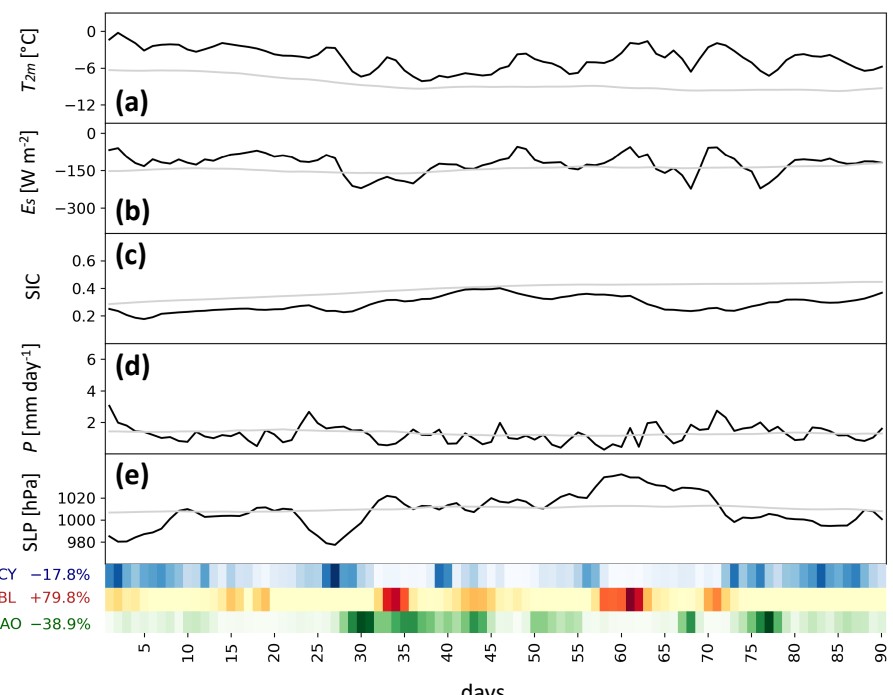

**Figure 9.** Time series of daily-mean **(a)** $T_{2m}$ (in $°C$), **(b)** $E_S$ (in $W\,m^{-2}$), **(c)** SIC, **(d)** $P$ (in $mm\,day^{-1}$) and **(e)** sea-level pressure (SLP, in hPa) averaged in the region of the Kara and Barents Seas (**KBI**, **KBM**, and **KBS**) in DJF 2011/12 (black lines). The transient climatology is shown by grey lines. Blue, orange, and green heatmaps at the bottom of the figure show the daily-mean coverage of the region by cyclones, blocks, and CAOs, respectively (the darker the color the higher the coverage). Relative frequency anomalies of the three weather systems are given in percentages. The horizontal axis indicates days since the start of the season with day 1 corresponding to 01 December.

It is now interesting to compare the time series of the basic variables with the occurrence of specific weather systems. The colored heatmaps at the bottom of Fig. 9 provide information about the occurrence of cyclones, blocks and CAOs in the Kara and Barents Seas. As each weather system is identified as an object described by a two-dimensional binary field (grid points that belong to a system have a value of 1 and other grid points have a value of 0), the weather system frequency field is calculated by time averaging of these binary fields. For example, if a cyclone mask covers a grid point at 25 % of all times, then time averaging of the binary fields yields 0.25, corresponding to a cyclone frequency of 25 %. Here, the color intensity is representative for the daily mean weather system frequency averaged over the area of the sub-region, thus it indicates the percentage of the sub-regions' area that overlaps with a cyclone, blocking or CAO mask on a daily basis. The repeated passage of cyclones (Fig. 9, blue heatmap) originating from the Nordic Seas (not shown) ensures the continuous transport of warm and moist air masses into the Kara and Barents Seas throughout several periods, mostly during December and February. Yet, in the wintertime average, cyclone frequency in this region was slightly below climatology (as further discussed in section 5.3). In contrast, CAO frequency (Fig. 9, green heatmap) was strongly reduced while blocking frequency was substantially increased (Fig. 9, red heatmap) in this season. CAOs, which often occur after the passage of a cyclone in the cyclones' cold sector, as can be seen for example around days 30, 43 and 77, usually lead to a strong decrease in $T_{2m}$ and $E_S$ (associated with intense surface fluxes). Therefore, the relative lack of CAOs in this winter favors the persistence of above average $T_{2m}$. Several blocking episodes around days 34, 61 and 71 are associated with notable peaks of $T_{2m}$ and $E_S$. Animations S1-3 in the supplementary material show daily synoptic plots for each of the discussed case studies and further illustrate the interplay of the synoptic systems and the occurrence of the anomalies in the considered surface parameters.

This season's large anomaly magnitude in sub-regions **KBI** and **KBM** was mainly determined by its exceptionally positive $\overline{T_{2m}}^*$ and the resulting positive $\overline{H_S}^*$, favored by unusually frequent blocking events and the reduced frequency of CAOs throughout the season.

## 5.2 DJF 2016/17

The winter 2016/17 is classified as extreme in all sub-regions of the Kara and Barents Seas. The PCA biplot shows that in **KBI** and **KBM** the anomaly magnitude of this winter is mainly determined by negative surface flux anomalies, especially of $H_S$ (Fig. 7a and b). In **KBS**, a positive $\overline{P^*}$ is the strongest contributor to the anomaly magnitude (Fig. 7c) and in **KBM** this winter occurs with a strong positive $\overline{P^*}$ outlier (Fig. 5h). In fact, it is the winter with the most precipitation in the Kara and Barents Seas during the study period. Further, in **KBI** and **KBM**, a strongly negative $\overline{E_S}^*$ occurs as a clear outlier with respect to other winters (Fig. 5j and l). Finally, $\overline{T_{2m}}^*$ shows a positive anomaly in **KBM** and **KBS**, which, however, is not exceptional (Fig. 5d).

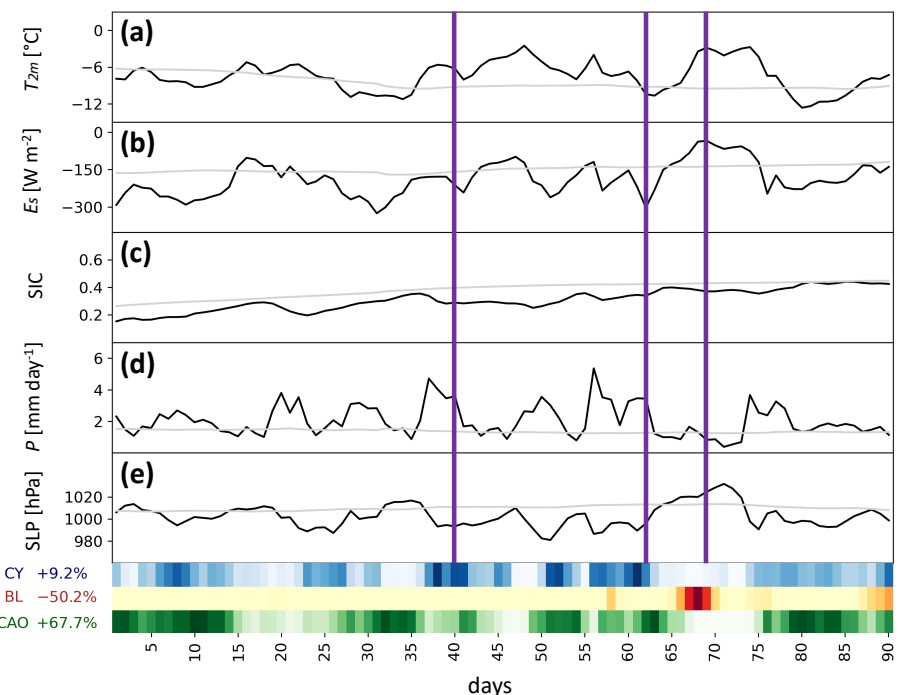

**Figure 10.** Time series of daily-mean **(a)** $T_{2m}$ (in °C), **(b)** $E_S$ (in W m$^{-2}$), **(c)** SIC, **(d)** $P$ (in mm day$^{-1}$) and **(e)** sea-level pressure (SLP, in hPa) averaged in the region of the Kara and Barents Seas (**KBI**, **KBM**, and **KBS**) in DJF 2016/17 (black lines). The transient climatology is shown by grey lines. Purple vertical lines indicate the three time steps shown in Fig. 11, 09 January 2017 (day 40), 31 January 2017 (day 62) and 07 February 2017 (day 69), respectively. Blue, orange, and green heatmaps at the bottom of the figure show the daily-mean coverage of the region by cyclones, blocks, and CAOs, respectively (the darker the color the higher the coverage). Relative frequency anomalies of the three weather systems are given in percentages. The horizontal axis indicates days since the start of the season with day 1 corresponding to 01 December.

Several precipitation events result in the strongly positive $\overline{P^*}$ which often can be linked to the passage of a cyclone (Fig. 10d,
blue heatmap). Only very few episodes show $P^*$ values below climatology, e.g. at the beginning of February when the occurrence of a block causes dry conditions (Fig. 10d, red heatmap). The positive $\overline{T_{2m}^*}$ results from several episodic warm events with a duration of $\sim$ 5–10 days (Fig. 10a), each deviating more than $+5$ K from climatology. There are, however, also several periods that are notably colder than climatology, thus implying a small seasonal-mean anomaly. These periods typically are characterized by a CAO (Fig. 10, green heatmap). A negative SIC anomaly occurs throughout the season (Fig. 10c), which is
especially pronounced in **KBM** (not shown), with strong decreases in SIC following warm and wet episodes linked to the passage of cyclones (Fig. 10, blue heatmap). During these episodes, which occur for example around days 21, 37 and 57, the wind field associated with the cyclone affects the sea ice transport and pushes the sea ice edge further north, momentarily reducing the sea ice coverage mainly in the sub-region **KBM**. In the supplementary Fig. S7 we show an example, using PIOMAS sea

ice data (Schweiger et al., 2011), of how the passage of several cyclones between days 17 and 24 affects the sea ice transport
in the Kara and Barents Seas.

In general, daily-mean $E_S$ values correlate well with daily-mean $T_{2m}$ values, showing the most negative $E_S$ values on the colder days. The frequent occurrence of CAOs, resulting from the advection of cold and dry air in the cold sectors of cyclones, favors strong upward surface heat fluxes, further enhanced by the increase in open water area due to the preceding
sea ice retreat. Most cyclones pass the Kara and Barents Seas from west to east and, thus, rather zonally during this winter (supplementary animation S2), often causing the consecutive passage of the cyclones' warm and cold sector in the considered sub-region. The resulting positive relative frequency anomaly of CAOs in combination with the negative $\overline{\mathrm{SIC}^*}$ causes a strongly negative $\overline{E_S}^*$ (Fig. 10b), which is particularly pronounced in **KBI** and **KBM**. The $E_S$ anomaly results mainly from negative $\overline{H_L}^*$ and $\overline{H_S}^*$.


As pointed out previously, the anomalous conditions during this winter are related to different synoptic weather systems. Figure 11 exemplifies three characteristic but different synoptic circulation patterns associated with anomalously warm conditions (Fig. 11a, c) and anomalous surface fluxes (Fig. 11b, d). In January, a sequence of multiple cyclones continuously transport warm air from the southwest towards the Kara and Barents Seas (Fig. 10, blue heat map, and supplementary animation S2).
Figure 11a shows a typical situation where a cyclone from the Nordic Seas propagated into the Kara and Barents Seas region, leading to anomalously warm conditions in its warm sector and precipitation along its cold front. Since the cyclones become nearly stationary and a large part of their cold sector is often located outside of the region in the Greenland Sea or towards the High Arctic, they cause a net warming in the region of the Kara and Barents Seas as well as persistent precipitation during their passage (Fig. 10a and d). Figure 11b shows the case of a strong CAO in the wake of a cyclone. It causes enhanced upward
surface heat fluxes over the open ocean, resulting in a strongly negative $E_S$ anomaly as discussed in the previous paragraph (Fig. 11d). The persistent large-scale situation during a warming episode in February 2017, when a stationary block over northern Scandinavia in combination with a strong cyclone to the South of Greenland leads to anomalously warm conditions in its northern periphery (Fig. 10, red heatmap, and supplementary animation S2) is shown in Fig. 11c. Next to the enhanced poleward transport of mid-latitude air masses which is favored by this pattern, subsidence-induced adiabatic warming additionally
causes high surface temperatures for the duration of the block (cf. Papritz, 2020). At the same time, the presence of the block suppresses precipitation in the region, resulting in one of the driest periods of the season (Fig. 10d). In supplementary Fig. S8, we show the differences in the air mass origin for both warm events by using air parcel trajectories.

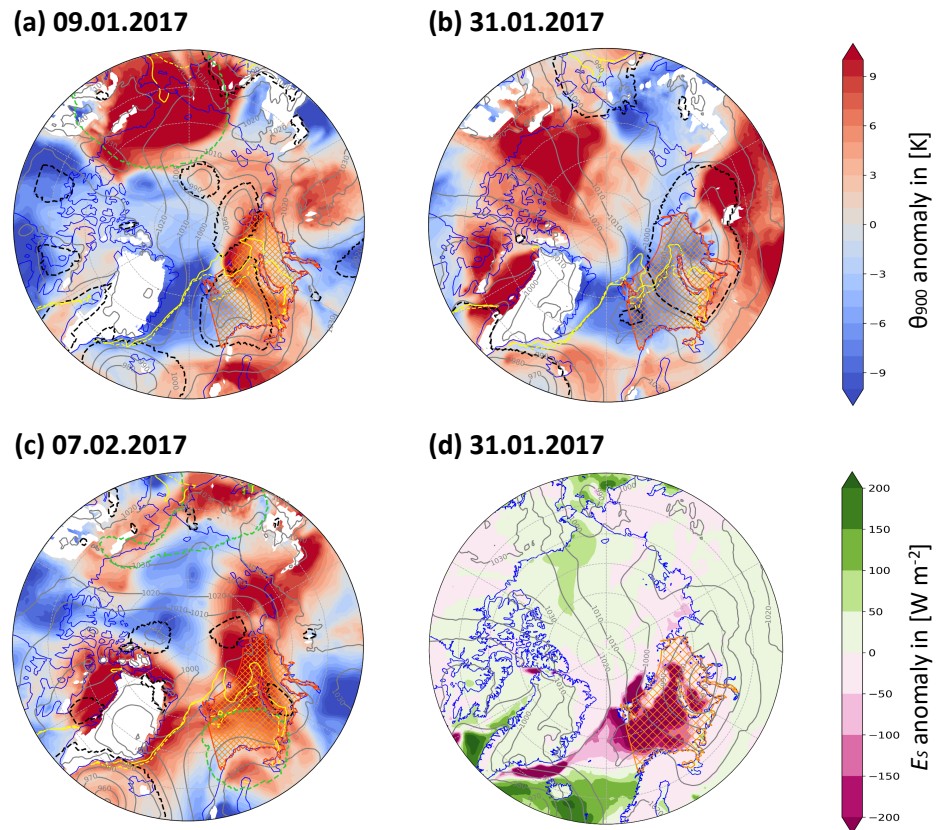

**Figure 11.** Synoptic situation on **(a)** 09 January 2017 (day 40), **(b)** 31 January 2017 (day 62), and **(c)** 07 February 2017 (day 69). Daily anomaly of potential temperature at 900 hPa ($\theta_{900}$; K, color). Sea-level pressure (SLP, grey contours, in intervals of 10 hPa), sea ice edge (SIC = 0.5, solid yellow line), climatological sea ice edge (SIC$_{\text{clim}}$ = 0.5, dashed yellow line), cyclone mask (dashed black contour) and blocking mask (dashed green contour) at 00 UTC on the considered days. Daily $E_S$ anomaly ($E_S^*$; W m$^{-2}$, color) on day 62 is shown in **(d)**. The region of the Kara and Barents Seas is marked by orange hatching.

Besides synoptic processes, preconditioning potentially plays an important role for the occurrence of an extreme season, as we aim to discuss now. From Fig. 10c, it can be seen that SIC in the Kara and Barents Seas was already exceptionally low at the start of the winter season, in fact, the sea ice extent on 01 December was the lowest on this date for the entire study period. At the same time, the sea surface temperature (SST) shows a significantly positive anomaly of about + 1 K on average, which favors a delayed freeze-up in the region and at the same time also more intense upward sensible and latent heat fluxes. These initial surface conditions provide an important precondition for the strongly negative $\overline{E_S^*}$, which itself is decisive for the anomaly magnitude of this winter. Analysing SIC and SST anomalies in the Kara and Barents Seas during the previous seasons in 2016 shows that they developed since the previous winter (SIC) or spring 2016 (SST, see Fig. 13b, which will be discussed in section 5.3). At the end of 2015, an extreme warm event (e.g., Boisvert et al., 2016; Binder et al., 2017) led to a

significant thinning of the sea ice in the Kara and Barents Seas, causing an early start of the melt season in 2016 and subsequently increased SST values in MAM, coinciding with a positive $\overline{T_{2m}}^*$ in the same region. The summer of 2016 does occur as an extreme season in sub-regions **KBM** and **NOM** (Fig. 8b and e) and as an anomalous season in **KBS** (Fig. 8c), mainly due to a strong $\overline{T_{2m}}^*$ of on average $+1.4\,\text{K}$ in the Kara and Barents Seas, which was facilitated by a reduction in total cloud cover and thus strongly enhanced $R_S$. Together with the already existing positive SST anomaly this extremely warm summer led to record low SIC and ice-free conditions in the Barents Sea from July to September (Petty et al., 2018). Strong blocking over large parts of the Arctic during October and November 2016 caused positive surface temperature anomalies across the whole Arctic region (Tyrlis et al., 2019) as well as strong positive $E_S$ anomalies, favoring the persistence of the negative SIC and positive SST anomalies (Blunden and Arndt, 2017) until the beginning of DJF 2016/17.

In summary, the winter 2016/17 was extreme in the Kara and Barents Seas due to a combination of preconditioning and favorable synoptic conditions. Specifically, a combination of strongly positive $\overline{\text{SST}^*}$ and negative $\overline{\text{SIC}^*}$ at the beginning of the season, and a relatively large number of CAO events throughout the season, resulted in strongly negative surface heat flux anomalies. Furthermore, an enhanced frequency of cyclones transporting warm and humid air masses into the region lead to a strongly enhanced $\overline{P^*}$.

### 5.3 Comparison of DJF 2011/12 and DJF 2016/17

Comparing both anomalous winters in the Kara and Barents Seas, it becomes already evident from the PCA biplots (Fig. 7a and b) that the processes leading to their respective anomaly magnitude are fundamentally different, as the vectors pointing to the two seasons in the biplot are nearly orthogonal. The winter of 2011/12 is dominated by a continuous positive $T_{2m}$ anomaly favored by a reduced frequency of CAO events, whereas in DJF 2016/17 the negative heat flux anomalies and exceptionally positive $\overline{P^*}$, enhanced by strongly reduced sea ice cover are most important. We have further seen in the previous paragraphs that both seasons feature large variability in the substructure of the respective parameters. To better understand the underlying processes leading to these differences, we will now analyze the synoptic situation in both seasons in more detail.

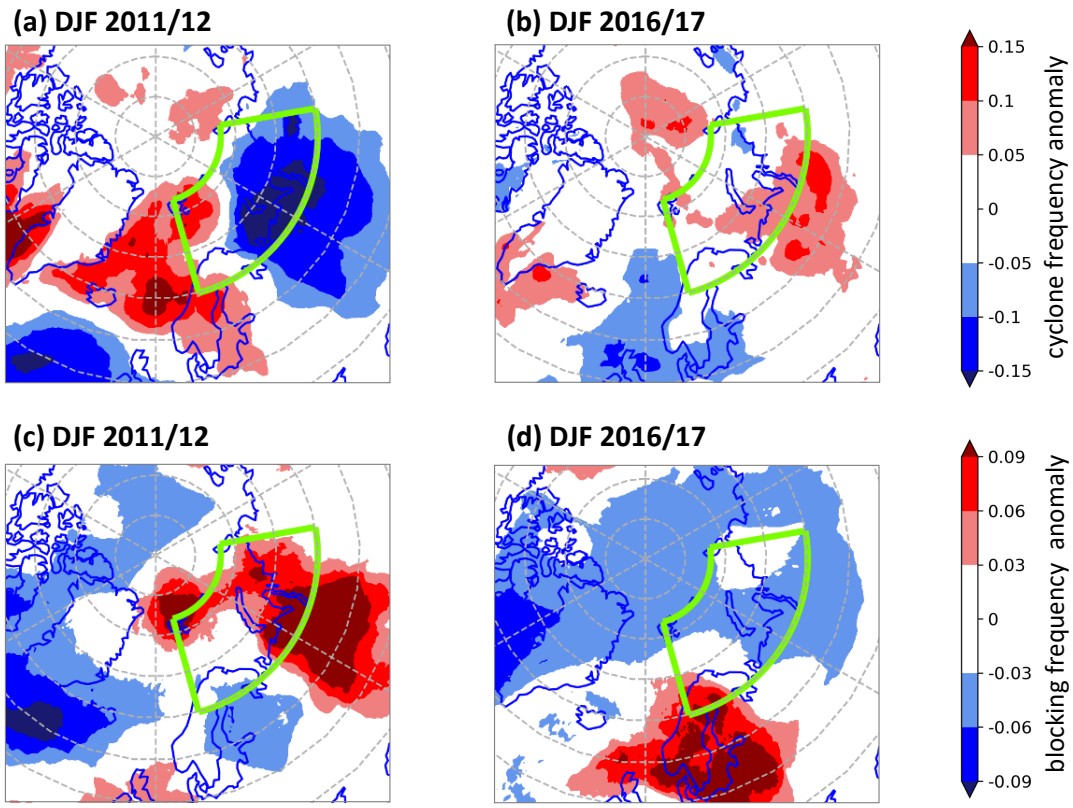

**Figure 12.** Seasonal-mean **(a, b)** cyclone frequency and **(c, d)** blocking frequency anomalies for **(a, c)** DJF 2011/12 and **(b, d)** DJF 2016/17. Region of the Kara and Barents Seas is marked with green contour.

The synoptic activity differs between these seasons. In DJF 2011/12, cyclone frequency was strongly enhanced over the Nordic Seas concomitant with a reduction in the Kara and eastern Barents Seas (Fig. 12a), which favored the frequent advection of warm air masses into the Barents Sea. Enhanced cyclone activity was restricted to the Nordic Seas and the western Barents Sea where several cyclones slowed down and became stationary (see supplementary animation S1). As a result, during several days of this winter, the warm sector of a cyclone was located in the Barents Seas, causing an increase in surface temperatures, whereas its cold sector was positioned in the Nordic Seas. Thus, the frequency of cold air outbreaks, which preferentially occur in the cyclone's cold sectors, was reduced in the region of the Kara and Barents Seas, favoring the formation of a positive $T_{2m}^{*}$. In addition, recurrent blocks over the Ural mountains (Fig. 12c) contributed to above average surface temperatures. In DJF 2016/17, in contrast, cyclone activity was close to climatology (Fig. 12b) as cyclones crossed the region (see supplementary animation S2), but instead blocking frequency over Scandinavia was strongly enhanced (Fig. 12d). Subsidence-induced warming and long-range transport of warm air masses contributed to several warm episodes (see Fig. S8 in the supplement).

However, an enhanced frequency of CAOs, facilitated by the frequent passage of cyclones combined with reduced SIC and warm ocean temperatures, limited $\overline{T_{2m}}^*$ but contributed to a strongly negative $\overline{E_S}^*$. Thus, the patterns of synoptic activity were partly reversed between the two seasons, yet they contributed substantially to their anomalous nature. Further, it becomes evident that the impact of cyclones on surface anomalies depends critically on their track relative to the region of interest.

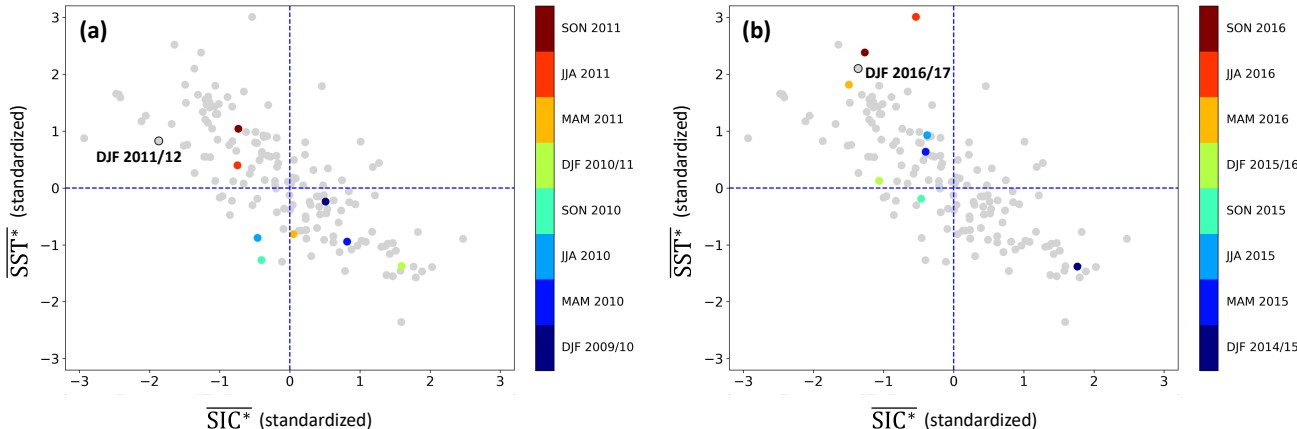

**Figure 13.** Standardized seasonal-mean anomalies of SIC ($\overline{\text{SIC}^*}$; along x-axis) and SST ($\overline{\text{SST}^*}$; in K, along y-axis) in the Kara and Barents Seas for the entire study period (grey dots) including all seasons. Colored dots highlight the eight seasons preceding **(a)** the anomalous winter 2011/12 and **(b)** the extreme winter 2016/17.

In addition to the synoptic activity, we found the influence of preconditioning in SIC and SST values to be of great importance for DJF 2016/17. Figure 13b shows persistent negative $\overline{\text{SIC}^*}$ and positive $\overline{\text{SST}^*}$ throughout the preceding 1.5 years. Comparing the initial conditions for the winter of 2011/12, the influence of the previous seasons seems to be minor, as SIC and SST values are close to normal at the beginning of the winter and seasonal-mean anomalies in spring and summer 2011 show no significant negative and positive anomalies, respectively (Fig. 13a).

## 5.4 DJF 2012/13

After analysing two anomalous winters in the Kara and Barents Seas, we now discuss another anomalous Arctic winter in the High Arctic to better understand the different dynamical processes leading to such seasons in Arctic regions with distinct surface conditions. In the region of the High Arctic, the winter of 2012/13 is classified as strongly anomalous in **ARI** mainly due to its negative $\overline{T_{2m}}^*$ and $\overline{P}^*$ (Fig. 7g), making it one of the coldest and driest winters in this sub-region (Fig. 5a and e). A negative $\overline{R_L}^*$, i.e., less net longwave radiation, resulting in an overall strongly negative $\overline{E_S}^*$ contributes additionally to the anomaly magnitude of this winter (Fig. 5i). Figure 14a shows that the $T_{2m}$ anomaly mainly results from deviations up to $-8\,\text{K}$ from the climatology during the second half of the season, which is quite a substantial anomaly considering the size of the

spatially averaged area, whereas the first half of the season is close to climatology. From mid-January on, $E_S$ values are also consistently below average and little to no precipitation occurs until the end of the winter (Fig. 14b and e). It is evident that only the second half of the season features exceptional conditions, indicating that anomalies do not have to persist throughout a whole season to make it anomalous.


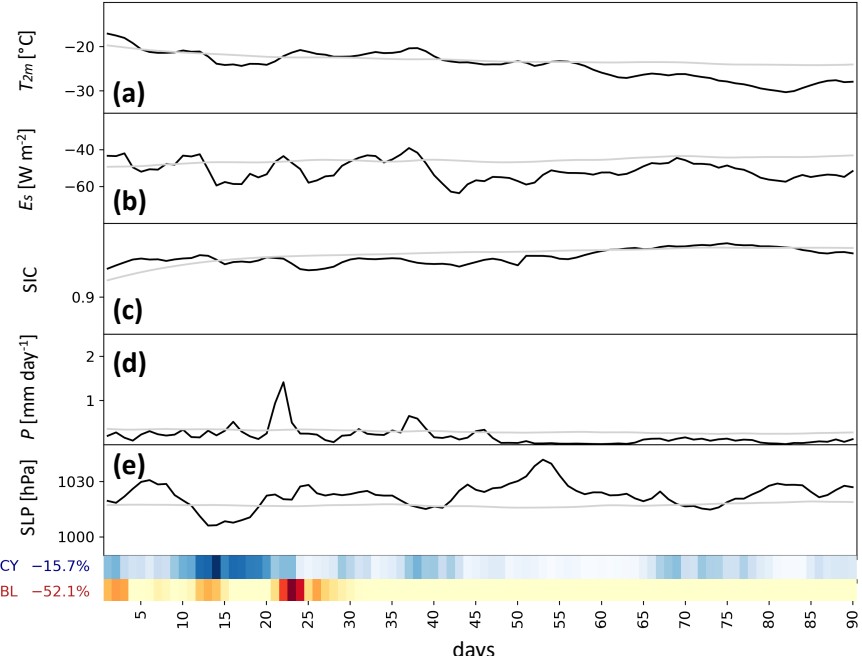

**Figure 14.** Time series of daily-mean **(a)** $T_{2m}$ (in °C), **(b)** $E_S$ (in W m$^{-2}$), **(c)** SIC, **(d)** $P$ (in mm day$^{-1}$) and **(e)** sea-level pressure (SLP, in hPa) averaged in the sub-region **ARI** in DJF 2012/13 (black lines). The transient climatology is shown by grey lines. Blue and orange heatmaps at the bottom of the figure show the daily-mean coverage of the region by cyclones and blocks, respectively (the darker the color the higher the coverage). Relative frequency anomalies of the two weather systems are given in percentages. The horizontal axis indicates days since the start of the season with day 1 corresponding to 01 December.

The anomalies during the second half of the season coincide with a decrease of synoptic activity over the High Arctic. Specifically, the relative cyclone and blocking frequency anomalies in **ARI** are slightly and strongly reduced, respectively, especially in the second half of the season. In December, several cyclones and blocks affect the prevalent conditions in the High Arctic (Fig. 14, blue and red heatmaps and supplementary animation S3). Between days 20 and 25, a strong intrusion of warm and moist air facilitated by adjacent blocking and cyclone systems in the Bering Sea causes a strong precipitation event, coinciding with increasing surface mean temperatures and a local decline in sea ice coverage (Fig. 14a, c, d). At the same time, a displacement of the polar vortex occurs, which subsequently leads to a splitting of the polar vortex and a sudden stratospheric


warming event at the beginning of January 2013 (Coy and Pawson, 2015; Nath et al., 2016). As the region of the High Arctic is positioned beneath the saddle point of the resulting two cyclonic vortices in the stratosphere, relatively calm conditions lead

to the development of a high-pressure system in the Laptev Sea, which evolves into a strong and persistent polar high during January (Fig. 14e). Figure 15 shows that there is no upper-level anticyclone or block present in the High Arctic during that period. This suggests that the strong high-pressure system at the surface is most likely of a thermodynamic origin caused by cold and dense air below an inversion layer (as can be seen in the skew$T$-log$p$ diagram in Fig. 16), resulting from persistent radiative cooling and inducing a first drop in $T_{2m}{}^{*}$ at the end of January (Fig. 14a).

In February, the calm conditions in the High Arctic remain and prolong the isolation of the cold and dry air in this region. Again, a lack of notable upper-level forcing can be observed (see supplementary Fig. S9). With the increasing dryness of the air, persistent longwave radiative cooling of the surface results in a dome of very cold air, causing the formation of another surface high-pressure system during the second half of February and one of the strongest negative monthly $T_{2m}$ anomalies in this region. The formation of the dome of cold air is evident as a strong inversion in the skew$T$-log$p$ diagram (Fig. S10).

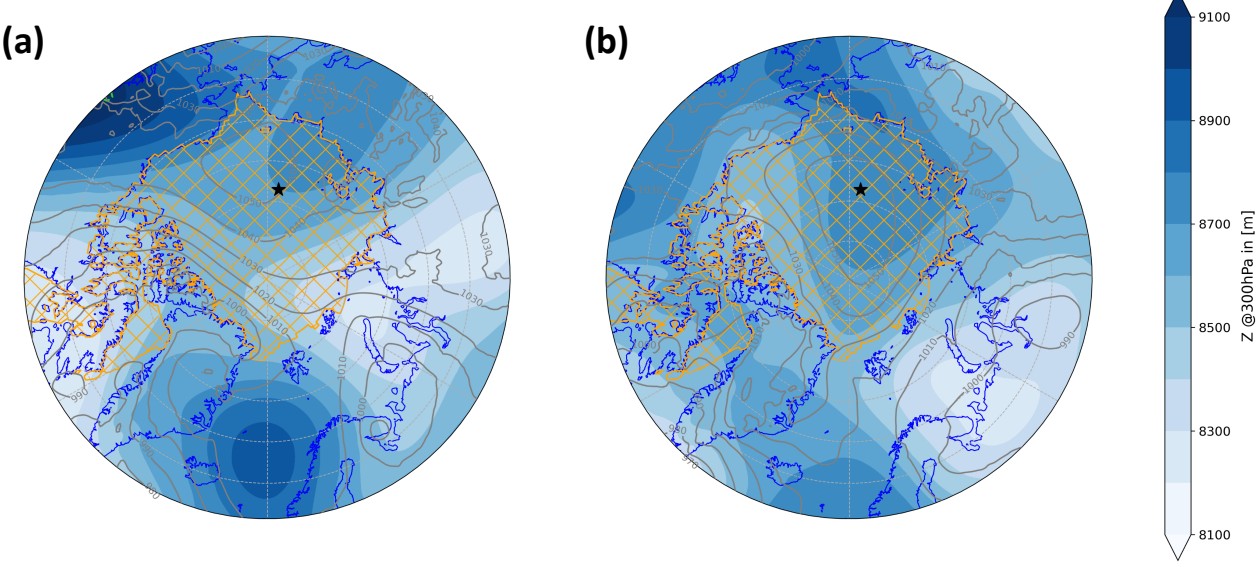

**Figure 15.** Synoptic situation on **(a)** 20 January 2013 (day 51) and **(b)** 24 January 2013 (day 55). Daily mean geopotential height at 300 hPa (in m, color). Sea level pressure (grey contours, in intervals of 10 hPa) and blocking mask (dashed green contour) at 00 UTC on the considered days. Black star at 173 °E, 78.5 °N shows location of skew$T$-log$p$ profile in Fig. 16. Sub-region **ARI** is marked by orange hatching.

Comparing winter 2012/13 in **ARI** with the two anomalous winters in the Kara and Barents Seas reveals fundamentally different characteristics, resulting mainly from the regionally varying synoptic activity but also the prevalent surface conditions. While preconditioning does not play an important role in the High Arctic, which is mainly covered by sea ice, the long-term development of SIC and SST anomalies in areas with varying SIC can significantly influence the initial conditions of winters in the Kara and Barents Seas. Each of the three seasons has its own substructure and different combination of anomalies resulting

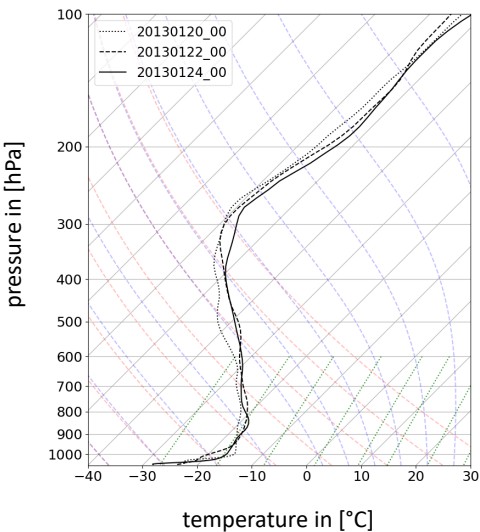

**Figure 16.** Skew*T*-log*p* diagram at 173 °E, 78.5 °N (black star in Fig. 15). Temperature is shown along the x-axis (in °C) and pressure along the y-axis (in hPa). Black lines show the ambient temperature profile for 20 January 2013 (day 51; dotted line), 22 January 2013 (day 53; dashed line), and 24 January 2013 (day 55; solid line) at 00 UTC. Grey lines show isobars (horizontal) and isotherms (skewed), respectively. Colored dashed lines denote dry (red) and moist (blue) adiabats, respectively. Green dotted lines denote constant saturation mixing ratios.

in the respective anomaly magnitude. Besides the rather straightforward "continuous anomaly in one parameter" as is the case for DJF 2011/12, we show that, with our approach to define extreme and anomalous seasons based on a multi-variate anomaly magnitude, there are many different pathways for such a season to develop. In DJF 2012/13, several weeks of consistent extreme conditions resulted in a whole anomalous season, although the first half of the winter was relatively normal. And in 2016/17, it was not only extraordinary atmospheric conditions but also the preconditioning by an anomalous evolution of the

surface conditions during the previous months that led to an extreme Arctic winter.

## 6 Discussion and conclusions

In this study, Arctic winters (DJF) and summers (JJA) have been characterized based on seasonal anomalies of surface parameters including temperature, radiation, heat fluxes and precipitation for distinct regions considering varying surface conditions. In winter, strong spatial differences can be observed dependent on the prevailing surface conditions (i.e., open ocean vs. sea

ice), especially in terms of the surface energy balance components ($E_S$), whereas in summer there is less spatial variability due to reduced surface temperature and radiation gradients. Regions with a climatological sea ice concentration of above 90% show only small $E_S$ variability mainly determined by changes in the net surface thermal radiation, as solar radiation and air-sea interactions are strongly reduced, particularly in the high latitudes. In contrast, areas with predominantly open water surface

show a large seasonal variability in the surface energy balance primarily driven by fluctuations in the surface heat fluxes. Temperature anomalies do show a distinct spatial variability as well, featuring relatively large fluctuations in sea ice covered areas in the Kara-Barents and Nordic Seas and reduced variability over the open ocean. The Nordic Seas are further characterized by an increased precipitation variability compared to the Kara and Barents Seas and the High Arctic, whereby the latter shows smaller variability for all analyzed parameters.

We further characterized Arctic seasons based on the seasonal substructure of surface temperature ($T_{2m}$), precipitation ($P$) and $E_S$. Continuous seasonal anomalies, indicating constantly anomalous conditions of the same sign throughout a whole season, can be observed for $T_{2m}$ and $E_S$ except for the open ocean, where strong surface heat flux variability prevents continuous $E_S$ anomalies. Distinct outlier seasons can be observed featuring exceptional seasonal-mean anomalies in one or several parameter(s).

To define and identify anomalous and extreme seasons objectively, we introduce a multi-variate method. Using PCA, we define anomalous and extreme seasons by means of an anomaly magnitude based on the combination of seasonal anomalies of $T_{2m}$, $P$, surface heat fluxes and surface net radiation. Unlike conventional, univariate approaches, we do not pre-define and thus prioritize one particular parameter by simply choosing, e.g., the warmest or wettest seasons. Instead, our multi-variate approach has the advantage that it also allows to identify seasons that arise from an unusual combination of seasonal anomalies that taken alone are not particularly unusual. This consequently leads to different types of extreme seasons in terms of their individual anomalies which, however, share a similar "unusualness" as expressed by the anomaly magnitude $d_M$. In order to reach a significant $d_M$ value, at least one of the considered variables or a combination thereof must be clearly exceptional compared to other seasons. All of our extreme seasons have very large anomalies for at least one parameter and thus would most probably be found to be extreme with a more conventional approach as well. We show that our identified anomalous seasons often result from various combinations of unusual seasonal anomalies, which allows us to analyze a broader spectrum of unusual seasons with regard to their underlying processes and atmospheric dynamics. Further, using a multivariate approach allows us to compare extreme and anomalous Arctic seasons considering the heterogeneity of the Arctic surface. We analyze sub-regions with climatologically high, mixed or low sea ice cover separately, thus accounting for regional differences in the surface conditions, which have a strong impact on the variability of these parameters.

Based on this definition of extreme seasons, we analyzed the atmospheric processes leading to three selected extreme and anomalous winter seasons by evaluating the relative importance of different synoptic features, namely cyclones, blocks and cold air outbreaks (CAO). This helps improving the understanding of the formation of such seasons and underlines the manifold processes that can cause a season to become particularly unusual. The results of our analysis for three different case studies can be summarized as follows:

1. Seasonal substructure: Extreme and anomalous Arctic winter seasons show a high variability in their substructure and the synoptic processes determining their anomaly magnitude. This magnitude can be due to a continuous seasonal anomaly

in one parameter such as it is the case for the constantly positive temperature anomaly during the exceptionally warm winter 2011/12 in the Kara and Barents Seas. However, also the combination of several noticeable but not exceptional seasonal anomalies can result in a similarly large anomaly magnitude. Furthermore, extreme conditions do not need to persist during a whole season as we can see for the winter of 2012/13 in the High Arctic, where several weeks of persistent cold and dry conditions caused seasonal anomalies that are sufficiently large for the season to be identified as anomalous.

2. Atmospheric processes: Various synoptic processes can cause Arctic winters to become anomalous or extreme. An increase in cyclone frequency often leads to enhanced transport of warm and moist air into the respective region, which is particularly important for the formation of precipitation in the higher latitudes. Episodes of prevailing atmospheric blocking usually favor the persistence of positive surface temperature anomalies due to subsidence-induced adiabatic warming. Recurrent synoptic events such as cyclones, blocking and CAO episodes can strongly influence the entire season, depending on their location relative to the considered region. Similarly, the absence of synoptic activity can be important for the development of extreme conditions as can be seen in the case of the High Arctic extreme winter 2012/13. Contrasting synoptic conditions can lead to extreme seasons in the Kara and Barents Seas, which, however, show very different characteristics. Further, the frequency of CAOs strongly influences surface temperature anomalies and changes in $E_S$ mainly due to the impact on air-sea interaction.

3. Surface preconditioning: Regions with varying sea ice coverage can experience preconditioning due to long-term anomalies in sea ice concentration (SIC) and sea surface temperature (SST), leading to anomalous initial conditions at the beginning of the season and thus influencing the sea ice formation and $E_S$ throughout the following winter. Large SIC and SST anomalies, which developed and persisted throughout the preceding 1.5 years, led to record-low SIC and above average SST in the Kara and Barents Seas at the beginning of the winter of 2016/17. Due to the increased amount of open water area, predominantly negative surface heat flux anomalies prevailed throughout the season, resulting in an exceptionally negative seasonal $E_S$ anomaly. This suggests that extreme and anomalous seasons in regions with a climatological sea ice concentration between 10 % and 90 % can be caused by such a preconditioning, whereas extreme and anomalous seasons in regions with continuous sea ice extent are mainly driven by atmospheric processes.

One of the main limitations of this study is the short time-period for which the ERA5 data is currently available. As our goal is to study anomalous seasons, the number of suitable cases is strongly limited. Future analysis of large ensemble simulations of the CESM climate model will allow us to further statistically quantify and confirm the results of this study. The importance of long-term components such as the near-surface ocean processes leading to possible preconditioning of anomalous seasons have only been briefly considered in this study. Further analysis of anomalies in surface oceanic heat transport and its influence on sea ice formation and melt and sea surface temperatures will allow us to quantify the relative importance of short-term atmospheric and long-term oceanic forcing in driving the processes leading to Arctic extreme seasons.

*Code and data availability.* ERA5 data can be downloaded from the Copernicus Climate Data Store (https://cds.climate.copernicus.eu/). The PIOMAS data set can be obtained from the Polar Science Center web page (http://psc.apl.uw.edu/research/projects/arctic-sea-ice-volume-anomaly/data/). Scripts used to produce the analyses and figures in this study are available on request from the authors.

*Author contributions.* KH performed most of the analyses, produced all figures and wrote the initial draft of the manuscript. All authors
contributed to the design of the study, the understanding and interpretation of the results and the writing of the paper.

*Competing interests.* The authors declare that they have no competing interests.

*Acknowledgements.* KH and MB acknowledge funding by the European Research Council 485 (ERC) under the European Union's Horizon 2020 research and innovation programme (project INTEXseas, grant agreement no. 787652). We thank Mauro Hermann and Matthias Röthlisberger for input and helpful discussions, and Michael Sprenger (all ETH Zurich) for his help with preparing the ERA5 data. KH thanks
Katharina Heitmann for feedback on the first draft of the manuscript. The authors acknowledge MeteoSwiss and ECWMF for providing access to the ERA5 reanalyses. We thank Irina Rudeva, two anonymous reviewers, and the Editor Camille Li for their constructive feedback that helped to clarify and strengthen the presentation of our results.

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
