# Peer review of "Identification, characteristics, and dynamics of Arctic extreme seasons"

_Weather and Climate Dynamics, 2021_

## Author Response (AR1)

**WCD-2021-18**

**Identification, characteristics, and dynamics of Arctic extreme seasons**

Response to the Reviewers' comments by Katharina Hartmuth, Maxi Boettcher, Heini Wernli, and Lukas Papritz

We thank the editor and all three reviewers for their insightful and helpful comments. We address each comment point by point below. The editor's and reviewers' comments are given in blue and our responses in black. The most important aspects of our replies and revisions are:

1) As suggested by reviewer 1, we used the North et al. approach to show that the first two PCs are statistically distinguishable from the others.
2) We now better explain the several subjective choices that were necessary for our analyses.
3) We clarify our choice to use a multivariate approach to explore different types of extreme seasons.

Please note, that we always refer to the lines in the **revised manuscript** (document without track changes). Figures in the reply document are referred to as "Fig. R1", etc., and figures in the revised manuscript are referred to as "Fig. 1", etc. We further supplement this document with a pdf containing track changes (latexdiff-pdf showing changes from first manuscript version to revised manuscript).

**Editor**

All three reviewers appreciated the study, and I agree with them that the work represents an interesting and important contribution. Some valid concerns are raised in their thorough and insightful reviews, and the authors have provided good indications of how they will address these concerns. In preparing a revised manuscript, I would encourage the authors to focus particularly on a few points which many of the reviewers' comments group around. Also, there remain small English usage errors throughout (some but not all mentioned by the reviewers).

1) Interpretation of identified extreme seasons, mentioned in detail by R2 and also touched on by R1 (anomalous vs extreme) and R3 (rescaling). I believe it would be worth spending some more time to put results from this method in context of results from conventional approaches (R2 comment #1), which would also go towards highlighting the impact/novelty of this study.

Thank you for this comment. We extended the discussion of our method and did our best to better highlight the novelty and differences of our approach compared to a more conventional method (such as, e.g., simply choosing the warmest or wettest seasons).

We further updated Table 2, which now includes the rank for each seasonal-mean anomaly (with respect to all seasons) of our identified extreme seasons. We can show that each extreme season has an extraordinary seasonal-mean anomaly (rank 1 or 2) for at least one parameter.

Thus, the extreme seasons identified with our method would also be identified with a more classical approach (e.g., the extreme summers 2013 and 2016, which show the most negative and most positive $\overline{T_{2m}}^*$, respectively, in the whole study period). However, with our method we also identify anomalous seasons that are characterized by an unusual combination of seasonal mean anomalies, each of which is not particularly noteworthy (in terms of their rank). We have included a discussion of the benefits of our approach over a more classical one in the discussion (Section 6).

For further details see responses to reviewer questions (R1: general comment (1) and specific comments (2,15), R2: general comment (1) and R3: specific comment (8)).

2) Clarification of methodology (all reviewers), including the choice of winter case studies only.

We included all suggestions from the reviewers to clarify and improve the discussion of our method throughout the manuscript. At the end of chapter 4 (L363ff.), we included a more detailed justification for the choice of our three case studies.

3) Sharpening of the presentation (e.g., many nice suggestions from R2, R3 regarding framing questions). I agree with the authors that the length is fine, especially given the amount of work that has been done, but the text could be edited to better guide the reader to the main messages. It could help also in the abstract and conclusions to put more weight on what we learn about point #1 and less weight on details of the case studies.

Thank you for these remarks. We added some metadiscourse at the beginning/end of a few sections to ensure a better guidance of the reader. We further sharpened our main messages by integrating the reviewers' suggestions and, e.g., adding further discussion of our method in the discussion chapter (L584ff.).

**Reviewer 1**

**General comments**

The authors evaluate the atmospheric conditions during anomalously extreme seasons in the Arctic. This is performed using a regional principal component (PC) analysis (PCA) from ERA5 data of the first two PCs of all seasons from 1979-2018. Furthermore, the PCA uses six key surface variables and divided spatially into 9 Arctic sub-regions. The sub-regions are subjectively chosen, but based on climatological sea ice conditions in either the Nordic Seas, Kara-Barents Seas, and the rest of the Arctic. Results identify 2-3 extreme seasons for each season (DJF, MAM, JJA, SON) in each sub-region. The PCA applied here provides a quantification of which variables contribute most to the extreme conditions of the respective season, and how consistent those conditions are during those particular seasons. The authors then choose two extreme seasons in the Kara-Barents sea during winter (DJF) to further investigate the synoptic weather conditions that were occurring and how they might have lead

up to the resulting seasonal extremes. The chosen seasons are picked based on their orthogonal, yet extreme, projections onto the PCs.

This research nicely demonstrates how PCs can be used to identify seasonal anomalies and extremes in certain regions of the Arctic. It furthermore demonstrates how to use that information to provide an expectation of how an extreme season was characterized with regard to one of the six variables and how consistent those conditions were. It is certainly a nice way to be able to identify extreme seasons that might be worth analyzing in further detail at shorter time and space scales if desired. Overall, I think these results can make a contribution and be published once some remaining issues are addressed. In particular:

1) Picking the first two principal components is subjective and does not necessarily isolate most of the variance. It first needs to be established that the first two principal components are the only significant ones. I do not doubt that this is the case given that on line 282, it is stated that they usually explain 80-90% of the variance.  However, it should be shown that they are indeed statistically distinguishable from the others. North et al. (1982) provide a well-established method of statistically distinguishing the first few eigenvalues from the others.

Thank you for this comment and for pointing us to the method introduced by North et al. We applied this method and the results reveal that the first two PCs in DJF and JJA are, with the exception of sub-region **ARM** in JJA, always statistically distinguishable from the others. We added this information to the revised manuscript (L305ff.). Here we provide further details about our results from applying the North et al. (1982) method, which we also show in the supplement.

Figures R1 and R2 show the standard errors for each eigenvalue in our PCA as introduced by North et al. (1982) for each sub-region in DJF (left-hand side) and JJA (right-hand side), respectively. The estimate for the standard error is given by

$$\delta\lambda_\alpha \approx \lambda_\alpha (2/N)^{1/2},$$

where $\lambda_\alpha$ denotes the respective eigenvalue and N the sample size, which in our case corresponds to the 39 realizations of the four seasons of the study period. Along the y-axis, the eigenvalue for each Principal Component (PC) is given and the error bars represent the estimated standard error. For both seasons, the first two eigenvalues, $\lambda_1$ and $\lambda_2$, are either clearly distinguishable or their error bars show only a very small overlap, except for sub-regions **KBI** and **NOM** in winter. Further, $\lambda_1$ and $\lambda_2$ are always clearly distinguishable from the third eigenvalue $\lambda_3$. The only exception is the sub-region **ARM** in JJA for which the error bars of $\lambda_2$ and $\lambda_3$ have a significant overlap. Thus, we can show that the first two PCs, which we use for the definition of our extreme seasons and which explain between 80%-90% of the variance in the respective sub-regions, are almost always statistically distinguishable from the remaining eigenvalues. We conclude that PC1 and PC2 isolate most of the variance and the corresponding eigenvalues are statistically distinct.

[Figure]

Figures R1 and R2: Standard errors for each PCA eigenvalue for all sub-regions in DJF (Fig. R1) and JJA (Fig. R2). The number of the Principal Component is given along the x-axis and the eigenvalue of each Principal Component along the y-axis. Error bars denote the estimated standard error following North et al. (1982).

2) Section 3 and generally throughout: The values of all correlations and their p-values that are described should be listed in a table.

We agree that it would be helpful to add a list containing all correlations and respective p-values for the described relations between the different parameters. We thus added such a table to the supplementary material and refer to it in the paper.

3) Figures 5 and 6 are a very nice way to illustrate the seasonal anomalies and the variability that may have also been occurring within those seasons. Having never seen these diagrams before, it at first takes a little bit of time to understand. It would be very helpful if there were a schematic showing the "phase space" of the interpretation that illustrates what is said in words on lines 251-258 (i.e., regions on the graph where there would be anomalies that tend to be continuous, where there would be warm episodes alternating with weak cold episodes, where there would be several intense warm and cold episodes that nearly cancel, where they would be near the climatology, etc.).

Thank you very much for pointing this out. To better understand and interpret the figures, we added lines of a constant ratio of the seasonal-mean anomaly and the seasonal-mean absolute anomaly ($\frac{\overline{X^*}}{\overline{|X^*|}}$) to the diagrams, such as you can see in the schematic figure below. We further adapt lines 258-278 in the revised manuscript as follows:

[Figure]

Figure R3: Schematic figure showing seasonal-mean anomalies ($\overline{\chi^*}$, along x-axis) vs. seasonal-mean absolute anomalies ($\overline{|\chi^*|}$, along y-axis). The dots in this schematic figure are colored by the ratio of both measures, $\frac{\overline{\chi^*}}{\overline{|\chi^*|}}$, which is additionally visualized by grey dashed lines. $\frac{\overline{\chi^*}}{\overline{|\chi^*|}} = \pm 1$ is shown by stippled grey lines and $\frac{\overline{\chi^*}}{\overline{|\chi^*|}} = 0$ is shown by a dashed blue line.

'To better understand the seasonal substructure of Arctic winters and summers, we compare the seasonal-mean anomalies ($\overline{\chi^*}$) with the seasonal-mean absolute anomalies ($\overline{|\chi^*|}$) for $T_{2m}$, $P$ and $E_S$ in selected sub-regions in DJF (Fig. 5) and JJA (Fig. 6). The ratio of seasonal-mean and seasonal-mean absolute anomalies, $\frac{\overline{\chi^*}}{\overline{|\chi^*|}}$, is indicative of the temporal persistence of an anomaly throughout a season. Thus, the location of a season in the diagrams provides information about the substructure of the season in terms of the considered parameter. In general, the further to the right, the more positive is the seasonal-mean anomaly of the shown parameter and the further to the left, the more negative. The closer the seasonal-mean anomaly is to the seasonal-mean absolute anomaly (dots close to the outer stippled grey line representing $\frac{\overline{\chi^*}}{\overline{|\chi^*|}} = \pm 1$), the more persistent the anomaly is throughout the season. Thus, we define seasons with $0.8 \leq \frac{\overline{\chi^*}}{\overline{|\chi^*|}} \leq 1$ as seasons with a "continuous" anomaly. With a decreasing absolute value of $\frac{\overline{\chi^*}}{\overline{|\chi^*|}}$, the seasons are located further away from the outer stippled grey lines, meaning that positive or negative anomalies in the respective parameter occur more episodically throughout the season. The closer a season is positioned towards the blue dashed line where $\overline{\chi^*} = 0$ and thus $\frac{\overline{\chi^*}}{\overline{|\chi^*|}} = 0$, the more positive and negative daily anomalies cancel each other, leading to a weak overall seasonal anomaly. The value of $\overline{|\chi^*|}$ is further indicative of the magnitude of the daily anomalies throughout a season. A season located at the top of the plot shows stronger daily anomalies than a season with the same $\frac{\overline{\chi^*}}{\overline{|\chi^*|}}$ ratio but a smaller $\overline{|\chi^*|}$.

For example, a season can be anomalously warm because the daily-mean $T_{2m}$ values are larger than the climatology on almost all days of the season , resulting in $\frac{\overline{T_{2m}^*}}{\overline{|T_{2m}^*|}} \approx 1$. With a decreasing ratio of both anomaly metrics, e.g. $\frac{\overline{T_{2m}^*}}{\overline{|T_{2m}^*|}} = 0.5$, the season is still anomalously warm, but it

results from several warm episodes alternating with weaker and/or shorter periods with negative $T_{2m}{}^*$ values. If $\dfrac{\overline{T_{2m}{}^*}}{\left|\overline{T_{2m}{}^*}\right|} \approx 0$, cold and warm episodes cancel each other leading to a weak overall seasonal anomaly. Comparing two seasons with the same $\dfrac{\overline{T_{2m}{}^*}}{\left|\overline{T_{2m}{}^*}\right|}$, the season which is positioned further up in the plot (showing larger values of $\overline{\left|T_{2m}{}^*\right|}$ and $\overline{T_{2m}{}^*}$) shows a larger variability in $T_{2m}$ with more intense warm and/or cold episodes compared to the season which is located further down.'

4) The justification for choosing two winter cases is weak. Perhaps this is because the anomaly values are smaller in the summer. But is it not dM and the standardized anomalies that determine how extreme a season is with these methods? These are just as strong in the summer (Table 2). I can see that the results shown in Figure 5 are used to pick the cases, but again, this seems contrary to the main setup of this paper of using the PCs to determine the extremes. Also, why 2011/12 in the Kara Barents Sea when this categorizes as "anomalous" rather than "extreme" in these methods? Regardless, it is hard to justify the title "Identification, characteristics, and dynamics of Arctic extreme seasons" when only the dynamics of winter extreme seasons were discussed.

We agree that our choice of the case study seasons (three winter seasons, two in the Kara and Barents Seas, and one in the High Arctic) is subjective. They are also to a certain extent a compromise between "many more seasons would be interesting to study in greater detail" and "the paper should remain readable and have a reasonable length". We think that choosing three winter seasons makes their comparison easier and their differences more revealing. We would like to show that even in the same region (Kara and Barents Seas) two anomalous/extreme winter seasons can have a completely different substructure and can be associated with different weather systems, emphasizing the inter-annual variability. Showing this for a winter and a summer season would be less surprising and interesting as this would mix seasonal and inter-annual variations. It is also important to us to evaluate the case studies in some depth, which limits the number of cases fitting into the paper to about three. Of course, we also looked at other seasons but then decided that the selected cases nicely illustrate the diversity and complexity of the involved processes, which is one of the key aims of our study. In the revised version we now better explain that the choice of the case studies is subjective and motivated by our intention to reveal the diversity and complexity of the involved processes (L363ff.).

Regarding the title, it's true that we identify and characterize extreme seasons in summer and winter, but then only discuss the dynamics in winter. However, as we now explicitly mention this limitation and better explain the key aim of the case studies, we think that the title is justified.

**Specific comments:**

1) Line 135: Why are only marine cold air outbreaks (CAOs) considered? There are also significant CAOs over land, described in Biernat et al. (2021).

We are only considering ocean and ice grid points and thus only marine cold air outbreaks, which are identified on grid points with less than 50% sea ice.

2) Line 186: Choosing a dM threshold of 3 seems quite subjective. How is this threshold picked? If each principal component has a significant anomaly of two standard deviations, this could provide an expectation for what would be significant when considering the PCs in combination.

The thresholds for anomalous resp. extreme seasons are indeed a rather subjective choice. However we find that with these thresholds we obtain on average 0-1 extreme seasons per sub-region (which equals 0-2.5% of all seasons) and 4-5 anomalous seasons per sub-region (equalling 15-17% of all seasons). Assuming a normal distribution, these values correspond to the range of 2-3$\sigma$ for our extreme seasons and 1-2$\sigma$ for our anomalous seasons. Further, with this number of extreme seasons, the return period of such a season corresponds to approximately 40 years. Several studies, e.g., Röthlisberger et al. (2021) used this as an adequate measure for defining extreme seasons.

As a side note, we would like to mention that preliminary analyses of 1000 years of (present-day) CESM large-ensemble data show that our chosen threshold of $d_M$=3 results on average in a return period of around 70-90 years. We are, thus, confident that classifying the seasons with $d_M$>3 as "extreme" is well justified.

3) Line 219: Be more specific about "almost always." What percentage of the time is it true? Same thing for line 225... what percentage of the cases translates to `usually'?

Thank you for pointing this out. We adjusted the manuscript in the indicated section (L225ff.) to clarify the mentioned relationships between the different variables.

4) Line 262: How close to the |P*| = P* line does a season need to be in order to be called "continuous?" For example, the 2016/2017 winter season was pretty close, but not exactly on it. On the other hand, there are very few cases of |T2m*| = T2m* being exactly equal in the summer while it is described as "continuous" on line 260.

There are indeed only very few cases where the seasonal-mean and seasonal-mean absolute anomalies of a season are equal. Thus we changed our definition of a "continuous anomalous season" from $\overline{|\chi^*|} = \overline{\chi^*}$ to $\overline{|\chi^*|} \approx \overline{\chi^*}$, including seasons with a ratio of $0.8 \leq \frac{\overline{\chi^*}}{\overline{|\chi^*|}} \leq 1$ (see previous comment). We added these changes to the revised manuscript (L265ff.).

5) Line 307: Would also be useful to point out that there is very little 2-m temperature variability over the Arctic sea ice in the summer. This could imply that temperature variability may not play a major role in sea ice loss, which has very large interannual variability in the summer.

Figure 8 does indeed suggest that $T_{2m}$ has only little variability in regions with $SIC_{clim}>0.9$ in summer compared to other sub-regions. However, $T_{2m}$ is capped above sea ice as the air is cold and the excess energy goes into the melting of the ice if $T_{2m}$ is above the freezing point, which essentially limits (near-surface) temperature variability. We further assume that the sea ice loss in summer is equally strong in the other sub-regions (especially the mixed sub-regions with very variable SIC), which show a larger variability in SIC. As there exists only one sub-region with $SIC_{clim}>0.9$ in summer, we think that additional analyses would be needed to make such a statement.

6) The justification of how an extreme season is chosen on Lines 310-314 should be moved up to Section 2.3.

We already explain this in Sect. 2.3. The text in lines 334ff in the revised manuscript is meant as a reminder. We now clarify this by writing "As explained in Sect. 2.3, …".

7) Line 315: Which season does Figure 2 show? This could also be referenced here along with Table 2.

Figure 2 in the manuscript is only a schematic biplot which does not refer to a specific season nor region. We slightly changed the figure caption to clarify that this is only an idealised plot.

8) Figures 9, 10, 14: Would be helpful to label the x-axis with the month/date instead of the day of the season, esp. to be consistent with the text.

We mostly use "on day 12,15, 20…" throughout the text and only rarely real dates. Thus, we adapted the manuscript such that we don't use specific dates anymore, as we think that this ensures better readability.

9) Line 367: How are blocking, cyclone, and CAO frequencies computed exactly? Need references and a short description.

A common feature of our weather system identification schemes is that they produce a two-dimensional binary field, often referred to as the "mask" of the weather systems, where grid points that belong to a system have a value of 1 and the others have a value of 0. Simple time averaging of these binary fields then automatically delivers the weather system frequency field. For example, if a cyclone mask covers a grid point at 25% of all times, then averaging 25% times a value of 1 and 75% a value of 0 leads to a frequency of 0.25 (we added this information in lines 397ff.). For the specifics of the identification scheme, we added a few sentences for each weather system and now give the relevant references to the papers that introduced these schemes in lines 132ff.

10) Line 389: "Several episodic precipitation events..." But wouldn't Fig. 5h suggest consistent precipitation events?

Thank you for pointing this out. We deleted "episodic" to clarify the constant occurrence of precipitation events throughout the season.

We changed "it is obvious that" to "it can be seen that".

To make it clear that JJA 2016 is somehow connected to our case study DJF 2016/17, we additionally labeled it in Figs. 6 and 8 in the manuscript.

13) Lines 441-445: It is misleading to say that there were positive temperature anomalies over large parts of the Arctic in JJA 2016. This and the blocking was more centered over the Kara-Barents Sea region, while much of the central Arctic was not exceptionally warm and had frequent cyclones.

We are not sure if we understood your remark correctly, as we do not state that the positive surface temperature anomalies in JJA 2016 occurred over large parts of the Arctic, but only in the Kara and Barents Seas. We then state that in autumn 2016 (mainly during October and November), positive temperature anomalies occurred across the whole Arctic region as already shown by Tyrlis et al. (2019). We now clarify this further in the text by replacing "during autumn 2016" by "during October and November 2016" (L482).

14) Section 5.3: If the blocking frequency was greatest over Scandinavia, why were the warmest temperature anomalies over the Kara-Barents (KB) region and not co-located with the blocking? Seems like there should have been northerly flow over much of the KB region from air flowing over sea ice. Is it surprising that the air mass was not modified by the time it reached KB?

Thank you for pointing this out. First of all we want to emphasize that DJF 2016/17 was not a particularly warm season, but experienced several episodic warm events. Blocking over Scandinavia influenced the surface temperatures in the Kara and Barents Seas, especially during the warm episode in February 2017. Trajectories show that a majority of the air causing this warm episode originated over Scandinavia and was undergoing subsidence (we will add a short evaluation of some air mass trajectories to the supplement; see answer to comment (15) of reviewer 2 and Fig. R6). However, the pattern of blocking and cyclone anomaly patterns as shown in Fig. 12 in the manuscript does also support northerly flow into the region as you correctly assume, causing for example the period with a strong CAO in mid-February 2017, when cold air is transported from the High Arctic towards the South, facilitated by a block over Scandinavia and a cyclone in the eastern part of the Kara and Barents Seas. Please have a look at the supplementary animation S2 where we show the synoptic evolution for each day throughout the season. We also tried to further shape section 5.3 to better highlight in which way the synoptic patterns influenced the surface temperatures in our case study sub-regions.

15) Lines 120, 541: Is this approach really novel given that (Graf et al. 2017) first introduced it in a similar application?

Using a PCA for finding dominant variability modes has been done in several studies such as, e.g., Graf et al. (2017). However it has never been used to define anomalous or extreme

seasons based on the combination of several parameters. Thus, in terms of defining extreme seasons, this approach is novel. However, we deleted the word "novel" in L584 to clarify that the use of a multivariate approach per se is not novel.

**Technical corrections:**

1) Table 1: 2 m temperature --> 2-m temperature

We followed the WCD submission guidelines (see "House standards" for hyphen usage: "It is our house standard not to hyphenate modifiers containing abbreviated units (e.g. "3-m stick" should be "3 m stick")).

2) Table S1: Caption states standardized values are in brackets, but they are instead in parentheses.

Thank you for spotting this. We replaced "brackets" with "parentheses" in the caption of Tables 2 and S3-S6.

3) Section 2 should be "Data and methods" given that there is more than one method used to complete the analysis.

Changed as suggested.

4) Figure 1 caption: State what the green and red boxes denote.

We added the following sentence to the caption of Figure 1: "Green and red boxes denote the areas of the Kara and Barents Seas and Nordic Seas regions, respectively."

5) Line 135: There does not need to be a space between the number and the "%" symbol

Again, we followed the WCD submission guidelines (see Figure content guidelines: "Spaces must be included between number and unit (e.g. 1 %, 1 m).").

6) Lines 140-141: What is the sign convention for the surface energy balance?

Thanks for hinting at this. We added the following sentence in L144ff.: "Positive signs denote energy fluxes into the surface, whereas negative signs are indicative for energy fluxes into the atmosphere."

7) Line 183: There should be a period at the end of the equation.

A period has been added at the end of the equation.

8) Lines 352, 465: normal --> average

Changed as suggested.

9) Line 393: This --> These

Thanks for spotting this, we changed "this" to "these".

10) Line 424: Remove "of"

"Despite of this" has been changed to "despite this".

11) Line 453: Insert "of" after "Comparison"

"Comparison DJF 2011/12 and DJF 2016/17" → "Comparison of DJF 2011/12 and DJF 2016/17".

12) Line 463: got --> became

Changed as suggested.


**Reviewer 2**

The paper presents an analysis of variability in three Arctic regions using 6 metrics. An input of those metrics into the dominant modes of variability and links between those metric are discussed. Overall, I am impressed by the amount and quality of work done in this study.

Here is what I like about the paper:

- Fig 5 and 6, which show that while strong anomalies may be observed in one or two metrics, other metrics may remain close to their climatological values;
- assessment of the input of the six metrics into the main modes of variability and relationships between them;
- case studies (particularly fig. 10,11, 14) and the discussion around them. An attempt to establish a connection between the weather and seasonal anomalies is valuable;

- a wide range of metrics used in the study - not only T2m/SIC/P, but also energy fluxes, cyclone frequency, CAO and a blocking index.

However, there is a couple of major concerns that need to be addressed before the paper can be accepted for publication:

1. I am not convinced that the approach, introduced in the paper, is a good way to select extreme seasons. Despite the use of a multivariate approach, it often comes to just one metric showing a strong seasonal anomaly, which was enough to identify the season as extreme or anomalous. Thus, without applying this approach, one may simply go through all 6 metrics and select the most extreme season(s) in each of them. I don't think I saw a proof that the seasons selected with the PCA analysis were more anomalous than those that showed a strong anomaly but were not picked up by the PCA approach. The latter may be even more anomalous than those, that were selected using the PCA.

On the other hand, there are seasons that were identified as anomalous though none of the variables showed a strong anomaly. Could it be proved that they are 'true' anomalous seasons and not artefacts of the method?

I am not asking for a change of the approach here, but I think more discussion around potential (dis)advantages of the proposed method is needed. In my opinion, this method identifies the dominant modes of variability and allows for assessment of the contribution of each of 6 metrics into those modes and a link between them. Section 5 explores a few seasons when one of the first two modes of variability was among the strongest.

We appreciate and fully understand this remark. It is a priori not clear how extreme seasons should be defined. An obvious choice, which we also use in other studies, is to simply choose the warmest or wettest seasons. This would prioritize one parameter (e.g. temperature or precipitation) and a justification would be given why this parameter is particularly relevant. Here we tried something else, something more "objective" in the sense that we did not want to pre-specify the most relevant parameter. Instead, we allow for the possibility that besides individual parameters, also their combination can be unusual. Thus, we were led by the hypothesis that our multivariate approach will lead to different types of extreme seasons (different in terms of their individual anomalies of $T_{2m}$, $P$ and $E_S$), which, however, share a similar "anomalousness" as expressed by the parameter $d_M$. We don't think that this method produces artefacts; in order to reach a value of $d_M \geq 2$ (or even $d_M \geq 3$), at least one of the considered variables or a combination thereof must be clearly exceptional compared to the other seasons in the ERA5 time period. In the revised version we will make sure that this line of thought becomes obvious to the reader. At the same time, we cannot (and don't want to) prove that this approach is "better" than a more conventional one. If all that matters in a specific study is for instance the seasonal snow accumulation, then there is no need to work with our approach.

We adapted the discussion (L584ff.) to clarify the novelty and characteristics of our approach in the context of other, more conventional, methods.

2. My other concern is the length of the manuscript. Considering the amount of work, it is hard to make it shorter, but I think the paper will benefit from it. Some plots (especially, Fig. 3) are too busy and are difficult to interpret. Section 3 and 4, while interesting, are hard to read, particularly when plots discussed in the text are a couple of pages away (which is inevitable). Please select the most robust and/or important relationships and focus on them. I understand that each plot provides a lot of information, but, unfortunately, human beings can only keep a few facts in mind at a time.

We did our best to further streamline the text and make it as readable as possible. With regard to the length of the paper, we think that it is still fairly OK.

Other comments:

1) Abstract:   1.The abstract is a bit long, even if there is no word limit, a page-long abstract is not ideal.

Thank you, we shortened the abstract by about 20%.

2) I think it is worth mentioning that 2016/17 winter was mostly anomalous in terms of precipitation and maybe in some other variables, otherwise, until you read the paper, it remains unclear why it was anomalous.

Thank you for this remark. In the revised version of the abstract we write "In contrast, winter 2016/17 started with a strongly reduced sea ice coverage and enhanced sea surface temperatures in the Kara and Barents Seas. This preconditioning, together with increased frequencies of cold air outbreaks and cyclones, was responsible for the large upward surface heat flux anomalies and strongly increased precipitation during this extreme season." This makes it clear that DJF 2016/17 was mainly anomalous in terms of precipitation and surface heat fluxes.

3) Sect. 2.3: For the PCA analysis, was each metric first averaged over the corresponding region? Meaning that the special structure of those anomalies was not accounted for.

Yes, we average over the region and therefore lose information about the spatial structure.

4) Fig. 3: As I already mentioned above, it is a very busy plot, which is hard to read. The only thing that is obvious to me is that in JJA the red/blue markers can be linked to positive/negative temperature anomalies. For DJF, what is obvious is a link between T2m and P anomalies and that the low right corner has predominantly negative Es anomalies. However, regional differences, discussed in the text, are very hard to see. If you decide to keep this plot, maybe splitting into different geographical locations or the sea ice concentrations helps.

Thank you very much for the suggestion. We adapted the figure (Fig. R4 below shows the revised Fig. 3 of the paper), and now show the correlations for each SIC_clim range in separate panels.

[Figure]

Figure R4: Seasonal-mean anomalies of $P$ ($\overline{P}^*$; mm day$^{-1}$, along x-axis), $T_{2m}$ ($\overline{T_{2m}}^*$; K, along y-axis) and $E_S$ ($\overline{E_S}^*$; W m$^{-2}$, color) for 39 seasons in DJF (a,b,c) and JJA (d,e,f) for sub-regions with SIC_clim > 0.9 (a,d), 0.1 ≤ SIC_clim ≤ 0.9 (b,e) and SIC_clim < 0.1 (c,f).

5) l.230: Despite good clustering in Fig. 4, this plot is again very busy. Maybe you can show the average location for each of the nine sub-regions on top of the existing plot.

Thanks for this suggestion. We added averages for each sub-region to Fig. 4 in the revised manuscript (see Fig. R5 below).

[Figure]

Figure R5: Seasonal-mean absolute anomalies for $P$ ($\overline{|P^*|}$; mm day$^{-1}$, along x-axis), $T_{2m}$ ($\overline{|T_{2m}^*|}$; K, along y-axis) and $E_S$ ($\overline{|E_S^*|}$; W m$^{-2}$, color) for 39 seasons and (a) 9 sub-regions in DJF and (b) 7 sub-regions in JJA. Blue symbols mark average values for each sub-region.

6) Fig.5, 'the seasonal-mean absolute anomalies': are these the seasonal-mean absolute *daily* anomalies, as in Fig. 4?

Yes. We define the seasonal-mean absolute anomalies as the seasonal mean of the absolute daily anomalies (L238ff.), which is valid for all figures in Sect. 3.

7) Fig.5,6: why Nordic seas are not shown?

In order to limit the length of the paper (see also your remark above), we selected some sub-regions for Figs. 5 and 6 (those that are most relevant for the case studies). The other sub-regions are now shown in the Supplement.

8) l.285-287: The statement on correlation between T2m and P comes from the fact that the corresponding blue lines are close to each other?

Yes. The more parallel two precursor vectors are, the stronger is the correlation of the precursors, provided that PC1 and PC2 explain a large part of the variance (Gabriel, 1971; 1972).

9) Regarding the comment on the weather systems creating extremes in the high Arctic, I would like to agree, though none of the AR seasons across all regions in fig. 5 look particularly extreme. How about other regions that have stronger extreme seasons often just in one parameter - can they be explained by anomalous weather patterns?

Yes, we agree, variability is distinctively smaller in the High Arctic in winter. But we think that one strength of our method is that it is still able to objectively quantify the "anomaly magnitude"

of one season compared to another in a specific sub-region. A method using absolute thresholds would find most extreme seasons most likely at the poleward end of the storm tracks.

With respect to your 2nd question, we unfortunately don't understand what you mean by "other regions". For all seasons, which we investigated in detail, we found an important role of anomalous weather patterns.

10) l.303-307: Why the described connection between P and Rs over the sea, as well as between T2m and RL in KBM does not hold in the Kara-Barents Sea?

In **NOS** and **ARS** we can see an anti-correlation between $P$ and $R_S$ (corr($P,R_S$)=-0.91 in **NOS** and corr($P,R_S$)=-0.96 in **ARS**. Our argument for this anti-correlation is the presence of clouds during rainfall. As you correctly point out, there is no such anti-correlation in sub-region **KBS** (corr($P,R_S$)=0.05). This would indicate that there is a large variation in cloud cover (and thus possible reduction in $Rs$) also during periods without precipitation. We did, however, not investigate this relationship in further detail.

We also want to point out that the mentioned correlations are only an approximation (see previous comment), which is more accurate the more of the total variance is explained by the first two Principal Components. As in the mentioned sub-regions this explained variance ranges between 85% and 88%, we assume that the correlations are good enough to use them for our interpretation.

11) Section 4: The relationship between 6 metrics during cold and warm seasons, gained from the PCA analysis, is interesting. Could correlations found in this section be confirmed by using the raw data?

We are not sure if we understand this question correctly. Yes, we can confirm some of the correlations found with the PCA. For example it is shown in Fig. 3 that in DJF $T_{2m}$ and $P$ are mostly positively correlated in **ice** and **mixed** sub-regions, whereas there is no such correlation in regions over the open ocean. This correlation can as well be seen in the PCA biplots for DJF in Fig. 7. However, as we use six different variables for the PCA analysis, and there seems to be no conventional method to illustrate the correlation of six variables, we only show $T_{2m}$, $P$ and $E_S$ in Fig. 3. It is further important to mention that we use detrended seasonal-mean anomalies for the PCA and thus remove the seasonality and a potential trend compared to the raw data. Therefore, it is not straightforward to compare the correlation of certain parameters for both data sets.

12) l. 322: "By design, extreme seasons have very large anomalies for at least one parameter… However, some anomalous seasons don't show very strong anomalies in one particular parameter, which implies that for these seasons it is the combination of several parameters that makes them anomalous" I am not sure that the first sentence is true. Moderate anomalies in a few variables may also give an anomalous season and this is what happens in some cases.

Thank you for this remark. We rephrased the mentioned section (L349ff.) in order to clarify that indeed our **extreme** seasons have at least one large anomaly in one parameter (see previous

comment about the approach as well as answer to comment 1 by the Editor) to reach a $d_M$ value which is larger than 3. However, it is correct that this is not necessarily the case for **anomalous** seasons, where it is often the combination of several moderate anomalies resulting in $d_M$>2 (but smaller than 3).

13) l.367: I could not find a description of how cyclones, CAO and blocking events were defined.

For the specifics of the identification scheme, we added a few sentences for each weather system and now give the relevant references to the papers that introduced these schemes (L132ff., see answer to specific comment (9) by reviewer 1).

14) l.372: Even during CAOs the temperature remained above the climatological mean, hence, I doubt that 38%-deficit in CAO can be responsible for the season being anomalous. During the first month (days1-27), there were no significant blocking events and CAOs, but T2m was well above average. To me it looks like there was a strong preconditioning. Furthermore, in the next case, shown in Fig. 10, there is a high number of CAOs but they have relatively small effect on T2m, especially during the first half of the season, brings the temperature down by only, perhaps, 2-3 deg.

Thank you for this remark, it is certainly important to discuss this more thoroughly. As we state in section 5.3, where we discuss the synoptics throughout the winter 2011/12, one important feature is the pathway of the cyclones entering the Arctic from the North Atlantic, as they tend to slow down and get stationary in the region of the Nordic Seas, and their position relative to the Kara and Barents Seas. As a result, during several days of this winter, the warm sector of a cyclone is located in the Kara and Barents Seas whereas its cold sector is positioned in the Nordic Seas. This does not only explain partially the relative lack of CAOs, but also the overall increase in the surface temperature anomaly. If the cyclones were located further east, both the warm and the cold sectors would have been located in the region, likely resulting in no notable $T_{2m}$ anomaly. Comparing the timeseries in Fig. 9 with the supplementary animation S1 shows that this synoptic situation especially occurs in December and in the second half of February, when the $T_{2m}$ anomaly is very strong. For further studies it could thus be very useful to have a metric for the coverage of a region by a cyclones' warm sector as opposed to its cold sector (and thus the position of a cyclone with respect to that region). This would simplify the interpretation of a cyclones' influence on surface parameter anomalies in a distinct region. In the revised manuscript we now emphasize more that the impact of cyclones depends critically on their track relative to the region (L500ff.).

With regard to your comment on preconditioning in this season, we can say that this is most probably only a minor reason for the anomalous surface temperatures. Indeed, SON 2011 shows already slightly positive $\overline{T_{2m}}^*$ values and a slightly reduced SIC, but not to an extent that could explain the strong seasonal-mean $T_{2m}$ anomaly during DJF 2011/12. The sea surface temperature reaches values of about +1-1.5 K above normal in September 2011, however returns to climatological values in October and shows no significant anomalies throughout November.

15) l.465: A seasonal blocking anomaly over Scandinavia is probably not enough to support the statement that 'Subsidence-induced warming [over Scandinavia] and long-range transport of warm air masses contributed to several warm episodes.'

This is indeed correct. To confirm this statement, we added a short evaluation of some air parcel trajectories to the supplement, which show the importance of subsidence-induced warming and long-range transport during episodic warm events in DJF 2016/17.

Figure R6 shows air parcel trajectories for two warm episodes in DJF 2016/17 from 16-19 January 2017 (Fig. R6a) and from 11-14 February 2017 (Fig. R6b). In January, the influence of long-range transport of air parcels at lower levels, mainly from eastern Europe, can be observed. In February, subsiding air masses, favored by the presence of a blocking system over Scandinavia, additionally contribute to the warm event.

[Figure]

Figure R6: 10-day kinematic backward trajectories associated with positive daily mean $T_{2m}$ anomalies in the region of the Kara and Barents Seas for the period (a) 16-19 January 2017 and (b) 11-14 February 2017 colored according to pressure. Trajectories are initialized every 6 hours at 25, 50, 75 and 100 hPa above ground for grid points with $T_{2m}^* \geq 2\,K$. Every 100th trajectory is shown with black dots denoting the starting point of each trajectory.

16) l.498: why a persistent high does not cause subsidence warming? and why there are no blocking events during Jan 2013 at the time of a persistent high? I can also see a number of cyclones in Feb, despite the text says that Feb was also calm. I agree that probably the main reason for decreasing t2m and low P is that the High Arctic remained isolated from the lower latitudes, however, none of the metrics in this study reflect an exchange between latitudes. I am not suggesting adding such metric at this stage, but it might be something to add in the future.

Thank you for these remarks. It would certainly be useful to have a measure which is indicative for latitudinal air mass exchange to better understand the processes leading to extreme seasons in the High Arctic.

Regarding your questions about the non-co-occurrence of the persistent high-pressure system as well as the lack of subsiding air, we analysed the geopotential height as well as the potential vorticity (PV) anomaly at upper levels throughout this winter. Figure R7 shows the geopotential height at 300 hPa (Z300) during the episode of the strong high-pressure system between 15 January and 25 January 2013 in the region of the Chukchi Sea and the High Arctic. Z300 does not show significantly enhanced values above the surface high, indicating that there is no strong upper-level forcing in the form of a persistent ridge which could have caused the formation of a block and the strong subsidence of air. The analysis of the vertically averaged potential vorticity anomaly (VAPVA) between 500 and 150 hPa does further support these results, as it reaches only small negative or even positive values in the same region (for the identification of a block following Sprenger et al. (2017), an area with VAPVA < -1.3pvu which persists for at least 5 days would be needed). Thus we assume that the strong high-pressure system at the surface is caused by very cold air below an inversion layer, decoupled from the synoptics in the upper troposphere. We can show that there exists a strong inversion layer very close to the surface in the center of the high pressure system by using a skew$T$-log$P$ diagram (see Fig. R8), which supports our assumption that the air in this area experiences radiative cooling opposed to subsidence-induced adiabatic warming which one might expect in the presence of an upper-level block.

[Figure]

Figure R7: Synoptic situation on (a) 20 January 2013 and (b) 24 January 2013. Daily mean geopotential height at 300 hPa (in hPa, color). Sea level pressure (grey contours, in intervals of 10 hPa) and blocking mask (dashed green contour) at 00 UTC on the considered days. Black star at 173°E, 78.5°N shows location of skew$T$-log$P$ profile in Fig. R8. Sub-region **ARI** is marked by orange hatching.

[Figure]

Figure R8: Skew$T$-log$P$ diagram 173°E, 78.5°N (black star in Fig. R7). Temperature is shown along the x-axis (in °C) and pressure along the y-axis (in hPa). Black lines show the ambient

temperature profile for 20 January 2013 (dotted line), 22 January 2013 (dashed line), and 24 January 2013 (solid line) at 00 UTC. Grey lines show isobars (horizontal) and isotherms (skewed), respectively. Colored dashed lines denote dry (red) and moist (blue) adiabats, respectively. Green dotted lines denote constant saturation mixing ratios.

We replaced Fig. 15 from the first manuscript with Figs. R7 (Fig. 15) and R8 (Fig. 16) and adapted lines 544ff. in the revised manuscript to clarify the synoptics during DJF 2012/13.

17) l.529-534: the   paragraph first describes obvious seasonal differences (higher variability in winter due to stronger gradients) and then concludes 'hence, it is reasonable to subdivide the Arctic into several regions considering these spatial differences to study anomalous Arctic winter seasons.' But during summer the regions were also subdivided. I am not sure if this paragraph is needed at all.

Thank you very much for this remark. The mentioned paragraph is indeed a bit misleading and possibly not needed at all, which is why we deleted it. However, we still want to mention the difference in spatial variability between winter and summer and therefore added a sentence in this regard to the previous paragraph (L568).

18) l. 541: see my major comment on the PCA approach

See our response on p.12 of this reply document.

Minor comments:

1) l.61 'and of the feedback': remove 'of'

Changed "strongly affect the type of linkages between parameters and of the feedback processes" to "strongly affect the type of linkages between parameters as well as feedback processes".

2) Table 1:  Es should be added

Thank you for pointing this out, we added the variable "$E_s$" to Table 1.

3) Table 2   is first mentioned in section 2.3 but is only shown in section 4. Replace 'brackets' with 'parentheses'

Thank you, we changed "brackets" to "parentheses" in the Table caption.

Indeed we refer to Table 2 already in the method part to justify the detrending of our data set. However we prefer to show Table 2 only in the results part and not yet in the methods part as it basically shows the results of our analysis, based on the PCA biplots in Figs. 7 and 8.

4) l.160: it is not the entire ERA5 period, but the entire period covered by this paper

Thank you for pointing this out. We changed the regarding sentence to "A distinction is made between areas where, on all days of the considered season in the time period covered by this study, mainly sea ice is present …".

5) Please use either the Kara-Barents sea or the Kara and Barents seas

We now only use "Kara and Barents Seas".

6) 406: I'd replace 'single' with 'individual'

We replaced "single" by "individual".

7) 432: on this date

Thank you, changed as requested.

8) Fig. 13 is mentioned earlier than fig. 12.

We rephrased the mentioned part in line 474, to clarify that we only refer to section 5.3 here and not yet want to discuss Fig. 13.

9) Fig. 10,11, 14: I suggest showing months and days of months' along Axis X, instead of days of season, as specific dates are often mentioned in the text (e.g, 9 Jan or 17 Feb). Can SLP be added to fig. 10,11? In fig. 14 the legend mentions CAOs, but they are not shown - could they be added?

Regarding days/dates: see our answer to specific comment (8) of Reviewer 1.

We added SLP to Figs. 10 and 11. Further we removed the CAO heatmap description from the caption. It does not make sense to show the marine air outbreak frequency for sub-region **ARI**, as this region is mainly ice-covered and as mentioned in the method section (L139ff.), we define CAOs only for grid points with a sea ice concentration of less than 50%.

**Reviewer 3**

The authors have investigated seasonal extremes in the Arctic using PCA of six climate variables and analysis of some key dynamical elements – cyclones, blockings, and marine cold air outbreaks – to further investigate particular extreme seasons. This is an interesting and valuable framework for understanding the various causes of seasonal extremes, and it is very well presented. I recommend the manuscript for publication with some minor adjustments. My principal concerns relate to the justification of the many choices which needed to be made in this analysis, these are detailed below.

Specific comments:

1) L13: "respectively" – this doesn't quite follow when you say 2-3 extreme seasons for four seasons.

Thank you, we changed the wording to "...our approach identifies 2-3 extreme seasons for each of winter, spring, summer and autumn, with strongly differing characteristics…".

2) L15: I think a justification of why 2 winter seasons were chosen for the in-depth case studies is needed here.

See answer to general remark (4) by reviewer 1.

3) L117: It is very nice to have these questions in the Introduction to frame the paper, but as far as I could see the synoptic systems of interest are pre-defined in the study (cyclones, blockings, and marine CAOs), so perhaps this question should be reframed to reflect this.

This is indeed a good point. We rephrased question 3: "In which way do synoptic-scale weather systems such as cyclones, blocks and marine cold air outbreaks determine the sub-structure of extreme seasons?"

4) L131: What was the method(s) of interpolation?

This interpolation is done by the ECMWF software when downloading the ERA5 fields from the MARS archive.

5) L155: What is the justification for choosing these regions?

As stated, a distinction between areas with differing sea-ice concentration is made, as surface heat fluxes and surface radiation are strongly dependent on the surface conditions. Further, we defined three different geographical regions, namely the Nordic Seas (**NO**), the Kara and Barents Seas (**KB**) and the remaining Arctic (**AR**). These are chosen based on the following main features: The **NO** region is the endpoint of the Atlantic storm track and important for deep water formation. The **KB** region has been strongly affected by changes in sea ice concentration and reacts very sensitively to atmospheric forcing. It is also a preferred region for atmospheric blocking and has its "own" storm track. Region **AR** is largely uncoupled from the mid-latitudes. Due to these different characteristics, it is useful to look at these regions separately when analysing the dynamical processes leading to Arctic extreme seasons. We added a sentence in this regard to the manuscript (L160ff.) to further justify the choice of our regions.

6) L161: Are results sensitive to the choice of definition of ice, mixed, and sea? Why were these thresholds chosen?

The results are sensitive to the choice of the SIC thresholds when defining ice, mixed and sea, because obviously the resulting regions get larger or smaller depending on how the thresholds are changed. For instance, if for ice, the threshold $SIC_{clim}$ was lowered from 0.9 to 0.8 then this

would increase the size of the ice regions (and decrease the size of the mixed regions) and therefore the results for ice and mixed would be slightly less distinct. We decided to use relatively strict thresholds for ice and sea to ensure that these regions are indeed almost completely ice-covered and ice-free, respectively.

7) L174: Why choose just the first 2 PCs? This seems arbitrary, although I see later you mention that these explain a very large part of the overall variance.

See answer to first general comment of Reviewer 1. And yes, indeed the first two Principal Components explain usually 80-90% of the overall variance (in more detail: in 88% of the cases its >80% explained variance, in 53% of the cases they explain even >85% of the overall variance) and they are - for almost all regions and seasons - statistically distinct.

8) L178: Why are these rescaled by their respective SDs to give equal weight to each PC? Do you not wish to identify the extremeness of a season rather than the extremeness of a season with respect to these two PCs? (ref L114) If you don't do this rescaling do you still identify the same seasons as being extreme seasons?

We decided to use the scaled Euclidean distance (= Mahalanobis distance) in the PCA phase space to define our extreme seasons as with this approach, outliers in both, PC1 and PC2, are considered equally (without the rescaling, there would be more weight on the PC1 outliers). Thus, outliers in both PCs are treated similarly, independent of the individual variance explained by each PC.

We haven't tested the identification of extreme seasons without rescaling. Without rescaling, different, subjective thresholds for the definition of anomalous and extreme seasons would have to be chosen, which would hamper a direct comparison of the two methods.

9) Fig 8 and elsewhere: why was 10^5 km^2 chosen as the size threshold for a region?

This is a very pragmatic and subjective choice. Results from a PCA might be less reliable for very small regions. With this threshold, each region comprises at least 40 model grid points.

10) L393: grammar – "This periods typically are…"

Thank you, changed as suggested.

11) L408: "exemplarily shows" -> "exemplifies"

Thank you, changed as suggested.

12) L510: "as is the case"

Thank you, changed as suggested.

[revised manuscript text omitted]

---

## Referee Report (RR1)

**Review of WCD-2021-18, Revision 2**
**WCD-2021-18**

**Title**: Identification, characteristics, and dynamics of Arctic extreme seasons

**Author(s)**: Katharina Hartmuth et al.

**General comments:**
The authors evaluate the atmospheric conditions during anomalously extreme seasons in the Arctic. This is performed using a regional principal component (PC) analysis (PCA) from ERA5 data of the first two PCs of all seasons from 1979-2018. The PCA uses six key surface variables and divided spatially into 9 Arctic sub-regions subjectively chosen based on climatological sea ice conditions in either the Nordic Seas, Kara and Barents Seas, and the rest of the Arctic. Results identify 2-3 extreme seasons for each season and in each sub-region. The PCA applied here provides a quantification of how anomalous a season is relative to another season, which variables contribute most to the extreme conditions of the respective season, and how consistent those conditions are during those particular seasons. The authors then choose two extreme or anomalous seasons in the Kara-Barents sea during winter (DJF) and one extreme DJF season over the "High Arctic" to further investigate the synoptic weather conditions that were occurring. The chosen seasons are picked based on their orthogonal, yet anomalous or extreme, projections onto the PCs, as well as their diverse processes.

This research nicely demonstrates how PCs can be used to identify seasonal anomalies and extremes in certain regions of the Arctic. It furthermore demonstrates how to use that information to provide an expectation of how an extreme season was characterized with regard to of one of the six variables and how consistent those conditions were. It is certainly a notable method to identify extreme seasons that might be worth analyzing in further detail at shorter time and space scales if desired. I thank the authors for considering and addressing my major concerns from the previous version. I note that in the previous version of the manuscript, the sign convention for the surface energy budget was not stated (and I thank the authors for adding that in the latest revision), which has now allowed me to comment on the surface energy budget in this version with the new insight. Overall the arguments are much clearer and this manuscript will make a positive contribution and be published once a few remaining issues are addressed. Specifically:

1) I appreciate that the authors added correlation coefficient values and respective p-values in Tables S1 and S2. The corresponding text still needs some refinement, however. Since they are performing a statistical significance test, they can choose a p-value (say 0.05 or 0.01) to be a threshold as a "statistically significant correlation" a priori. This would clarify some of the statements, as sometimes I am not sure whether they mean that the correlation coefficient value is high (subjective) or whether they mean that the p value is low (objective since the correlation in that case is statistically significant). For example, on line 243, do they mean there is no *statistically significant* correlation in the summer? That would make most sense to me, given the high p-values. The magnitudes of the coefficients are also relatively low, but it is subjective to make conclusions based off that, alone. Generally, the lower p-values correspond to the higher coefficients, so it shouldn't change most of the conclusions to refine the wording whenever the word "correlation" is used.

2) The story for the DJF 2011/12 case (Section 5.1) is very interesting but does not quite seem to make complete sense to me as described here. It is argued that the consistently warm temperatures are because of the repeated passage of cyclones from the Nordic Seas.

Yet, the authors point out that overall, cyclone frequency was below climatology, and there were frequent blocking episodes. However, precipitation was one of the relatively lowest seasons over KBM with perhaps a few episodic precipitation events (i.e., Fig. 5h). Of course, blocking episodes favor warmer temperatures, and SLP was higher than average for much of the time when temperatures were above average and when there was blocking (Figure 9a,e, and colormaps). But then how does the consistently lower SIC come into the story? Note that the September 2011 sea ice extent was the second lowest since 1979 up to this point, and much of this was in the Kara and Barents Seas, so there could very well have been a new surface forcing in that region. Indeed it seems at a glance like this may not have an impact in DJF, because of the strongly positive and consistent $\overline{E_s^*}$ (Fig. 5l) and $\overline{H_S^*}$ (Figure 7b). Presumably cloud cover was anomalously low, though this could easily be checked by looking at whether $\overline{R_L^*}$ has the same sign as $\overline{E_s^*}$ and $\overline{H_S^*}$. Perhaps though, there was preconditioning in the autumn (October - November), which if there was anomalously more open water instead of ice, then there could be anomalously high upward sensible heat fluxes, which lead to warmer temperatures and higher tropospheric thicknesses by the time DJF started leading to forcings that favored patterns for blocking. Can this be ruled out? Figure 1 shows that in the two months before DJF 2011/2012, surface skin temperature, 2-m air temperature, and 850 hPa air temperature anomalies all had similar patterns to DJF in the lower troposphere before the blocking pattern arose, supporting a preconditioning by the surface conditions.

3) In the DJF 2016/17 case, is it that an extreme $E_s^*$ anomaly occurs when there is reduced SIC **and** a marine CAO? The cyclones reduce the SIC in KBM, and the passage of the cyclone is then followed by a CAO. For example, the extreme negative anomaly in $E_s^*$ between days 30-35 (Figure 10b). There was a cyclone around day 25 (Figure 10e, Figure S7) that reduced SIC. After the cyclone passed, there was a strong CAO at nearly the same time as the negative $E_s^*$ (Compare Figure 10b with Figure 10 CAO heatmap) and negative $T_{2m}^*$ (Figure 10a). Right now, the wording of the text in lines 449-462 makes it seem like the CAO is just an additional factor, but it seems like it may be the critical factor in order to lead to the overall magnitude/rank in $\overline{E_s^*}$ since lower sea ice alone would not necessarily do so. Also, the cyclone paths may be important such that there are frontal passages that promote CAOs in their wake in the right regions. Furthermore, if this is the story, I might consider choosing one of these marine CAO cases to highlight in Figure 11 instead of the current warm temperature anomaly case, esp. since overall $\overline{T_{2m}^*}$ was not ranked as highly (Table 2). I think this then flows better for the text on lines 481-486 where SIC, $T_{2m}^*$, marine CAOs, and $E_s^*$ are readdressed and furthermore when preconditioning is discussed thereafter on lines 488-506.

**Other specific comments:**

1) Line 19: normal $\rightarrow$ either "average" or "typical"

2) Line 88: Recent studies emphasized the *importance* of polar anticyclones and blocking events in the High Arctic **for what**? Insert into sentence.

3) Lines 90-93: Burt et al. (2016) also discuss this positive warming feedback, and additionally show a possible regional implication.

4) Line 145: Since the CAOs here can only be identified over water, insert "marine" in front of "cold air outbreaks."

5) Lines 299-300: How exactly can we see from Figures 5 and 6 that the driest years have non-zero precipitation amounts? What does it mean that it is less evident? Is this just a result that is not shown and therefore not evident from the plots?

6) Lines 346-347: In sub-regions ARI and ARM, I do not understand why the positive correlation between $\overline{R_L^*}$ and $\overline{T_{2m}^*}$ emphasizes the importance of clouds. For example, in the summer, when it is cloudier, it is cooler due to less shortwave radiation reaching the surface, so $\overline{T_{2m}^*}$ is more negative. Clouds increase the downwelling longwave radiation, thus making $\overline{R_L^*}$ more positive according to the sign convention.

7) Line 369: continuous → consistent

8) Line 371: What does "equally strong" mean when comparing two different variables?

9) Line 395: Can't we see that the largest $\overline{T_{2m}^*}$ over the 39-year period from Figure 5b,d instead of Figure 7b,d, where we can only see the principal component values, which are a combination of variables, correct?

10) Line 401: Insert $T_{2m}^*$ in front of "values" to make it less ambiguous exactly what values are being referred to, esp. since $\overline{T_{2m}^*}$ was just referred to beforehand.

11) Line 407: What is meant by "strongly correlated" exactly? Is there a correlation coefficient being referred to or is this meant to be a visual comparison between panels a and b in Figure 9?

12) Line 532: How much of the warming was caused by subsidence vs. horizontal advection? The trajectories show descending air parcels, but I can not see how much warming would be caused by that in this case.

13) Lines 539-543: Up until this point, preconditioning referred to the state of the sea ice at the beginning of the season in question (i.e., Question 4 on line 117). The discussion here and Figure 13 suggests a different (yearly) time scale, i.e., that the sea ice state from previous winters could be a preconditioning for the current winter. The shorter-term preconditioning may be more important in the 2011/2012 case, given that the September 2011 sea ice extent minimum was the second lowest extent on record up to that point.

14) Lines 666-667: Is the CESM large ensemble capable of representing the synoptic processes described in this paper?

**Technical corrections:**

1) Line 372: don't → do not

2) Line 401: Remove "about"

3) Line 405: we ↔ here

4) Line 488: Remove "also" before "preconditioning"

5) Line 545: Remove "want to"

**References**

Burt, M. A., D. A. Randall, and M. D. Branson, 2016: Dark warming. *J. Climate*, **29 (2)**, 705–719.

[Figure]

FIG. 1. Anomalies in monthly mean (a), (c), (e) October-November 2011 and (b), (d), (f) December - February 2011/2012 (a)-(b) surface skin temperature, (c)-(d) 2-m air temperature, and (e)-(f) 850 hPa air temperature. NCEP/NCAR reanalysis data are freely available from https://psl.noaa.gov/cgi-bin/data/composites/printpage.pl.

---

## Referee Report (RR2)

**Review of WCD-2021-18, Revision 3**
**WCD-2021-18**

**Title**: Identification, characteristics, and dynamics of Arctic extreme seasons

**Author(s)**: Katharina Hartmuth et al.

**General comments:**

The authors evaluate the atmospheric conditions during anomalously extreme seasons in the Arctic. This is performed using a regional principal component (PC) analysis (PCA) from ERA5 data of the first two PCs of all seasons from 1979-2018. The PCA uses six key surface variables and divided spatially into 9 Arctic sub-regions subjectively chosen based on climatological sea ice conditions in either the Nordic Seas, Kara and Barents Seas, and the rest of the Arctic. Results identify 2-3 extreme seasons for each season and in each sub-region. The PCA applied here provides a quantification of how anomalous a season is relative to another season, which variables contribute most to the extreme conditions of the respective season, and how consistent those conditions are during those particular seasons. The authors then choose two extreme or anomalous seasons in the Kara-Barents sea during winter (DJF) and one extreme DJF season over the "High Arctic" to further investigate the synoptic weather conditions that were occurring. The chosen seasons are picked based on their orthogonal, yet anomalous or extreme, projections onto the PCs, as well as their diverse processes.

This research nicely demonstrates how PCs can be used to identify seasonal anomalies and extremes in certain regions of the Arctic. It furthermore demonstrates how to use that information to provide an expectation of how an extreme season was characterized with regard to of one of the six variables and how consistent those conditions were. It is certainly a notable method to identify extreme seasons that might be worth analyzing in further detail at shorter time and space scales if desired. I again thank the authors for their consideration of my comments from the previous version, and now there is a more clear description of the two winter cases (DJF 2011/12 and 2016/17). However, given this extra clarity, I am not sure I agree that the two seasons are "fundamentally" different as stated on line 502. Otherwise, I don't see any issues and think this should be published once this remaining issue is addressed as described below:

1) Thank you for elaborating on the preconditioning of DJF 2011/12 in the review response. I have re-included my figure from the last revision (Fig. 1), which I think clearly shows the atmospheric response directly over the region of anomalously low SIC would be confined to mainly the KBI region and to some degree KBM during SON. I also included an additional figure showing the SIC anomaly and that the temperature anomalies persist and extend deep in the troposphere almost directly over the region of anomalous SIC through February (Fig. 2). Perhaps due to combining the 3 regions, figure R1 does not capture the negative $E_s^*$ due to the more limited area of the surface fluxes while the temperature anomalies more broadly surrounded the region because of a lack of other dynamics moving the air masses elsewhere. Thus, combined with all of the other information, I do not necessarily think the two winters are fundamentally different and share many similarities.

I do think these extreme cases are an interesting story and should be in this paper. My suggestion would be that since the stories do not end up being very different in my view, that sections 5.1-5.3 could be condensed. I think this is interesting and is a great demonstration of how sensitive seasonal extremes are to blocking (and how there is still a lot to learn about the onset of blocks). It seems that there was similar preconditioning (positive SST anomalies and

negative SIC anomalies) present at the beginning of the season in both cases. In 2011/12, this pattern set up in early autumn following the second lowest September sea ice extent (up to that time) and in 2016/17 it started in the previous late winter or spring. The primary difference appears to be in the atmospheric response. For whatever reason, the synoptic patterns were such that they did not favor CAOs in DJF 2011/12 while they did in DJF 2016/17, resulting in different surface fluxes and strong but non-consistently signed temperature anomalies in 2016/17 (i.e., surface cyclone tracks were different because the larger-scale flow pattern was different). Persistent blocking in 2011/12 did not provide a way for heat flux introduced into the atmosphere to be advected elsewhere, while 2016/17 was much more of a transient pattern with less frequent blocks. It is interesting that the larger-scale pattern and block more resembled the SIC anomalies in 2011/12 while not so much in 2016/17, and while these differences are interesting and should be noted, there could be many possible reasons as for why they occurred and therefore I think any additional explanation or speculation is beyond the scope of this study.

**Other specific comments:**

1) Lines 490-494: Cyclones also contributed to the low sea ice during the summer of 2016 (e.g., Finocchio et al. 2020; Lukovich et al. 2021).

2) In the DJF 2016/17 case study (Section 5.2), the results about that sea ice transport from several cyclones pushes the sea ice edge further north is a little strong with the given evidence. The PIOMAS data in Figure S7 is simply showing the transport vectors. While it looks quite plausible, there are still other factors that can not be ruled out, such as the impact of waves or upwelling. So I think this part of the discussion on sea ice can be shortened to say that there is an an apparent association with sea ice transport and the passage of cyclones.

**References**

Finocchio, P. M., J. D. Doyle, D. P. Stern, and M. G. Fearon, 2020: Short-term impacts of Arctic summer cyclones on sea ice extent in the marginal ice zone. *Geophys. Res. Lett.*, **47 (13)**, e2020GL088 338.

Lukovich, J. V., J. C. Stroeve, A. Crawford, L. Hamilton, M. Tsamados, H. Heorton, and F. Massonnet, 2021: Summer extreme cyclone impacts on arctic sea ice. *J. Climate*, **34 (12)**, 4817–4834.

[Figure]

FIG. 1. Anomalies in monthly mean (a), (c), (e) October-November 2011 and (b), (d), (f) December - February 2011/2012 (a)-(b) surface skin temperature, (c)-(d) 2-m air temperature, and (e)-(f) 850 hPa air temperature. NCEP/NCAR reanalysis data are freely available from https://psl.noaa.gov/cgi-bin/data/composites/printpage.pl.

[Figure]

Fig. 2. Anomalies in (a) November 2011 and (b) February 2012 sea ice concentration and (c) anomalies in monthly mean December - February 2011/2012 500 hPa air temperature. Data in panels (a)-(b) are from the NSIDC available from https://nsidc.org/data/seaice_index/archives/image_select and data in panel (c) are from NCEP/NCAR reanalysis data and are freely available from https://psl.noaa.gov/cgi-bin/data/composites/printpage.pl.

---

## Author Response (AR2)

**WCD-2021-18**

**Identification, characteristics, and dynamics of Arctic extreme seasons**

Response to the Reviewers' comments by Katharina Hartmuth, Maxi Boettcher, Heini Wernli, and Lukas Papritz

We thank both reviewers for their additional, helpful comments. We address each comment point by point below. The reviewers' comments are given in blue and our responses in black.

Please note, that we always refer to the lines in the **updated, revised manuscript** (document without track changes). Figures in the reply document are referred to as "Fig. R1", etc., and figures in the revised manuscript are referred to as "Fig. 1", etc. We supplement this document with a latexdiff-pdf showing changes from the first to the second revised manuscript.

**Reviewer 1**

**General comments:**

The authors evaluate the atmospheric conditions during anomalously extreme seasons in the Arctic. This is performed using a regional principal component (PC) analysis (PCA) from ERA5 data of the first two PCs of all seasons from 1979-2018. The PCA uses six key surface variables and divided spatially into 9 Arctic sub-regions subjectively chosen based on climatological sea ice conditions in either the Nordic Seas, Kara and Barents Seas, and the rest of the Arctic. Results identify 2-3 extreme seasons for each season and in each sub-region. The PCA applied here provides a quantification of how anomalous a season is relative to another season, which variables contribute most to the extreme conditions of the respective season, and how consistent those conditions are during those particular seasons. The authors then choose two extreme or anomalous seasons in the Kara-Barents sea during winter (DJF) and one extreme DJF season over the "High Arctic" to further investigate the synoptic weather conditions that were occurring. The chosen seasons are picked based on their orthogonal, yet anomalous or extreme, projections onto the PCs, as well as their diverse processes.

This research nicely demonstrates how PCs can be used to identify seasonal anomalies and extremes in certain regions of the Arctic. It furthermore demonstrates how to use that information to provide an expectation of how an extreme season was characterized with regard to of one of the six variables and how consistent those conditions were. It is certainly a notable method to identify extreme seasons that might be worth analyzing in further detail at shorter time and space scales if desired. I thank the authors for considering and addressing my major concerns from the previous version. I note that in the previous version of the manuscript, the sign convention for the surface energy budget was not stated (and I thank the authors for adding that in the latest revision), which has now allowed me to comment on the surface energy budget in this version with the new insight. Overall the arguments are much clearer and this manuscript will make a positive contribution and be published once a few remaining issues are addressed. Specifically:

1) I appreciate that the authors added correlation coefficient values and respective p-values in Tables S1 and S2. The corresponding text still needs some refinement, however. Since they are performing a statistical significance test, they can choose a p-value (say 0.05 or 0.01) to be a threshold as a "statistically significant correlation" a priori. This would clarify some of the statements, as sometimes I am not sure whether they mean that the correlation coefficient value is high (subjective) or whether they mean that the p value is low (objective since the correlation in that case is statistically significant). For example, on line 243, do they mean there is no statistically significant correlation in the summer? That would make most sense to me, given the high p-values. The magnitudes of the coefficients are also relatively low, but it is subjective to make conclusions based off that, alone. Generally, the lower p-values correspond

to the higher coefficients, so it shouldn't change most of the conclusions to refine the wording whenever the word "correlation" is used.

Thank you very much for this comment, this is indeed a good point as we mostly defined correlation rather subjectively via the correlation coefficient. However, we agree that it would be better to use p-values for a more objective assessment. Thus, we now use a p-value threshold of 0.05 to define a correlation as "statistically significant". We changed the wording about correlations throughout the manuscript to clarify our statements.

2) The story for the DJF 2011/12 case (Section 5.1) is very interesting but does not quite seem to make complete sense to me as described here. It is argued that the consistently warm temperatures are because of the repeated passage of cyclones from the Nordic Seas. Yet, the authors point out that overall, cyclone frequency was below climatology, and there were frequent blocking episodes. However, precipitation was one of the relatively lowest seasons over KBM with perhaps a few episodic precipitation events (i.e., Fig. 5h). Of course, blocking episodes favor warmer temperatures, and SLP was higher than average for much of the time when temperatures were above average and when there was blocking (Figure 9a,e, and colormaps). But then how does the consistently lower SIC come into the story? Note that the September 2011 sea ice extent was the second lowest since 1979 up to this point, and much of this was in the Kara and Barents Seas, so there could very well have been a new surface forcing in that region. Indeed it seems at a glance like this may not have an impact in DJF, because of the strongly positive and consistent E∗s (Fig. 5l) and H∗S (Figure 7b). Presumably cloud cover was anomalously low, though this could easily be checked by looking at whether R∗L has the same sign as E∗s and H∗S. Perhaps though, there was preconditioning in the autumn (October - November), which if there was anomalously more open water instead of ice, then there could be anomalously high upward sensible heat fluxes, which lead to warmer temperatures and higher tropospheric thicknesses by the time DJF started leading to forcings that favored patterns for blocking. Can this be ruled out? Figure 1 shows that in the two months before DJF 2011/2012, surface skin temperature, 2-m air temperature, and 850 hPa air temperature anomalies all had similar patterns to DJF in the lower troposphere before the blocking pattern arose, supporting a preconditioning by the surface conditions.

Thank you for these detailed remarks, we appreciate the amount of thought which has been put into this comment. It is indeed true that there has been a slight preconditioning in terms of sea ice during autumn 2011, as the sea ice formed later than usual, leading to a negative sea ice anomaly especially in October (which is evident in the anomaly patterns for skin temperature and 2m temperature, which match the shift in the sea ice edge). This is also evident from Fig. 13 in the manuscript, although it shows that the preconditioning for DJF 2011/12 was much weaker compared to DJF 2016/17. In addition, as the sea ice coverage in the Kara and Barents Seas reached values close to normal at the end of November 2011 (see Fig. R1d), surface preconditioning presumably plays a smaller role in determining the unusualness of this winter.

Regarding the surface fluxes during December 2011, $E_S$, $H_S$ and $R_L$ show a consistently positive anomaly, which speaks against the argument of low cloud cover leading to the positive $E_S$ anomaly. An analysis of the total cloud cover anomaly during this period shows no significant anomalies during this period.

Addressing the possible preconditioning of the increased blocking frequency in DJF due to the reduced sea ice coverage during October, we can show in Fig. R1c that there have not been consistently anomalous upward sensible heat fluxes in the Kara and Barents Seas in autumn. We think that such an increase in upward $H_S$ would need to occur consistently over a longer period to explain the increased number of blocks during DJF, especially considering the reduction in upward $H_S$ during December. Of course, we cannot completely rule out the role of autumn heat fluxes for winter blocks, but in our opinion this scenario is highly unlikely for DJF 2011/12.

Regarding the initial question "why has DJF 2011/12 been so warm?", it is surely a combination of different synoptic situations and dynamical processes that led to such a strong warm anomaly rather than one specific process. We slightly adapted the text such that this result is emphasized more strongly and further encourage the reader to have a look at the supplementary animation, which shows the relevance of a variety of synoptic processes throughout this winter.

[Figure]

Figure R1: Time series of daily-mean (a) $T_{2m}$ (in °C), (b) $E_S$ (in W m$^{-2}$), (c) $H_S$ (in W m$^{-2}$), (d) SIC, and (e) SST (in °C) averaged in the region of the Kara and Barents Seas in SON 2011 (black lines). The transient climatology is shown by grey lines.

3) In the DJF 2016/17 case, is it that an extreme E∗s anomaly occurs when there is reduced SIC and a marine CAO? The cyclones reduce the SIC in KBM, and the passage of the cyclone is then followed by a CAO. For example, the extreme negative anomaly in E∗s between days 30- 35 (Figure 10b). There was a cyclone around day 25 (Figure 10e, Figure S7) that reduced SIC. After the cyclone passed, there was a strong CAO at nearly the same time as the negative E∗s (Compare Figure 10b with Figure 10 CAO heatmap) and negative T∗2m (Figure 10a). Right now, the wording of the text in lines 449-462 makes it seem like the CAO is just an additional factor, but it seems like it may be the critical factor in order to lead to the overall magnitude/rank in E∗s since lower sea ice alone would not necessarily do so. Also, the cyclone paths may be important such that there are frontal passages that promote CAOs in their wake in the right regions. Furthermore, if this is the story, I might consider choosing one of these marine CAO cases to highlight in Figure 11 instead of the current warm temperature anomaly case, esp. since overall T∗2m was not ranked as highly (Table 2). I think this then flows better for the text on lines 481-486 where SIC, T∗2m, marine CAOs, and E∗s are readdressed and furthermore when preconditioning is discussed thereafter on lines 488-506.

Thank you for these very helpful remarks. In DJF 2016/17, the CAOs are indeed a critical factor, leading to strong anomalies in the surface fluxes and thus, strongly influencing this seasons' rank in $E_S$. As you point out, their occurrence in the wake of cyclones is particularly relevant, as the cyclones often cause a retreat of the sea ice edge (e.g., shown in Fig. S7), favoring strongly increased surface fluxes during the subsequent CAO. The cyclone paths do play a very important role as can be seen when comparing this winter with the winter 2011/12 in the same region, when most of the time the cyclones' cold sector does not reach the region of the Kara and Barents Seas, hampering the formation of strong upward surface fluxes. We adapted the text in section 5.2 to clarify the importance of this link between cyclones and CAOs.

We further adapted Fig. 11, reproduced here as Fig. R2, such that it now shows three different timesteps, including day 62, which features a CAO case in the Kara and Barents Seas. For

this timestep we added a fourth panel displaying the daily-mean $E_S$ anomaly, which is strongly negative in the region of the CAO. Thus, this figure does now not only highlight the variety of synoptic processes occurring throughout this season, but further emphasizes the importance of the link between cyclones and CAOs for the unusualness of this winter season.

[Figure]

Figure R2: Synoptic situation on (a) 09 January (day 40), (b) 31 January (day 62), and (c) 07 February 2017 (day 69). Daily anomaly of potential temperature at 900 hPa ($\theta_{900}$; K, color). Sea-level pressure (SLP, grey contours, in intervals of 10 hPa), sea ice edge (SIC=0.5, solid yellow line), climatological sea ice edge (SIC$_{clim}$=0.5, dashed yellow line), cyclone mask (dashed black contour) and blocking mask (dashed green contour) at 00 UTC on the considered days. Daily $E_S$ anomaly ($E_S$*; W m$^{-2}$, color) on day 62 is shown in (d). The region of the Kara and Barents Seas is marked by orange hatching.

**Other specific comments:**

1) Line 19: normal → either "average" or "typical"

Changed "normal" to "average".

2) Line 88: Recent studies emphasized the importance of polar anticyclones and blocking events in the High Arctic for what? Insert into sentence.

We mention this already in the second part of the sentence. For clarification we rephrased the line slightly such that it now reads:

"Recent studies emphasized the importance of polar anticyclones and blocking events in the High Arctic for driving subsidence-induced adiabatic warming, leading to anomalies in surface temperature and net surface radiation which cause increased sea ice melting."

3) Lines 90-93: Burt et al. (2016) also discuss this positive warming feedback, and additionally show a possible regional implication.

We added a reference to Burt et al. (2016).

4) Line 145: Since the CAOs here can only be identified over water, insert "marine" in front of "cold air outbreaks."

Line 139 now reads "Further, we define marine cold air outbreaks (CAOs) based on …".

5) Lines 299-300: How exactly can we see from Figures 5 and 6 that the driest years have non-zero precipitation amounts? What does it mean that it is less evident? Is this just a result that is not shown and therefore not evident from the plots?

Thank you for these questions. First, we can see from Figs. 5e-h and 6e-h that there is no season with a continuously negative precipitation anomaly, which would be the case if the marker of that season was located on the outer-left dashed grey line. Thus, for each season, the seasonal-mean absolute anomaly $\overline{|P^*|}$ is always larger than the seasonal-mean anomaly $\overline{P^*}$, indicating that both, positive and negative daily $P$ anomalies ($P^*$) exist. This would not be the case if precipitation was zero throughout the season, resulting in either no or a negative $P^*$.

Regarding your second and third question, we think it is remarkable that there are no seasons without at least a few days with above-average precipitation. This is not something we have calculated, but we just simply think that this is not obvious per se. (We also changed "less evident" to "less obvious", which better reflects what we intend to say.)

6) Lines 346-347: In sub-regions ARI and ARM, I do not understand why the positive correlation between R∗L and T∗2m emphasizes the importance of clouds. For example, in the summer, when it is cloudier, it is cooler due to less shortwave radiation reaching the surface, so T∗2m is more negative. Clouds increase the downwelling longwave radiation, thus making R∗L more positive according to the sign convention.

Thanks for pointing this out. As we have not looked in detail at these correlations (and the processes that possibly cause them) and only made an assumption, we removed this statement.

Here, we refer to our definition of a season with a "continuous anomaly" (see line 266ff.). However, as this result changed slightly with our correction of the calculation of seasonal-mean absolute anomalies (see corrigendum to the first revised manuscript), this statement does not appear anymore.

8) Line 371: What does "equally strong" mean when comparing two different variables?

Thank you for the remark, this is indeed not a very objective comparison, as the "equally" only refers to a subjective evaluation of the distance between outlier and other seasons in Fig. 6a and 6i. However, as this is not the case anymore for the sub-regions KBM and KBS, we changed the sentence such that it now reads "In summer, all extreme seasons are characterized by a strong $T_{2m}$ outlier (Fig. 6a,c,d), which co-occur with $E_S$ outliers in ARI and KBM (Fig. 6i,k)."

9) Line 395: Can't we see that the largest T∗2m over the 39-year period from Figure 5b,d instead of Figure 7b,d, where we can only see the principal component values, which are a combination of variables, correct?

Thanks a lot for spotting this typo! We want to refer to Fig. 5b and d here obviously, as the PCA biplots only show a combination of anomalies and no actual seasonal-mean anomalies.

10) Line 401: Insert T∗2m in front of "values" to make it less ambiguous exactly what values are being referred to, esp. since T∗2m was just referred to beforehand.

We changed the line such that it now reads "In the whole region, during December, $T_{2m}$ values are continuously around +6 K above climatology, …".

11) Line 407: What is meant by "strongly correlated" exactly? Is there a correlation coefficient being referred to or is this meant to be a visual comparison between panels a and b in Figure 9?

Yes, this is a visual comparison between both panels. We calculated the actual values of the correlation coefficient (0.66) and the p-value (1.16e-12), which indicate that the correlation between daily-mean $T_{2m}$ and $E_S$ is indeed statistically significant.

12) Line 532: How much of the warming was caused by subsidence vs. horizontal advection? The trajectories show descending air parcels, but I cannot see how much warming would be caused by that in this case.

Thank you for this question. For each warm event, we can classify the trajectories based on their thermodynamic development in terms of temperature ($T$) and potential temperature ($\theta$). If the trajectories experience an overall increase in $T$ but decrease in $\theta$, we define them as subsiding air masses (affected by radiative cooling). If, however, the changes in $T$ and $\theta$ have the same sign (negative, e.g., in the case of long-range transport from lower latitudes or positive, e.g., transport of very cold air masses over a relatively warm ocean), we assign them to horizontal transport (for more detailed information about this method see, e.g., Binder et al., 2017 and Papritz, 2020).

For the cases shown in supplementary Fig. S8 the share of subsiding/horizontally transported air parcels that reach the region of a warm anomaly is about 37%/61% in January and 58%/41% in February. Thus, we conclude that horizontal transport dominates the warm event in January, whereas in February, subsiding air contributes more to the warm surface temperature anomaly. However, these numbers do not reflect how much of the anomaly value has been caused by which process, they only reflect the percentage share of the two transport categories.

13) Lines 539-543: Up until this point, preconditioning referred to the state of the sea ice at the beginning of the season in question (i.e., Question 4 on line 117). The discussion here and Figure 13 suggests a different (yearly) time scale, i.e., that the sea ice state from previous winters could be a preconditioning for the current winter. The shorter-term preconditioning may be more important in the 2011/2012 case, given that the September 2011 sea ice extent minimum was the second lowest extent on record up to that point.

Thank you for mentioning this point. We agree that there are most probably different types and/or definitions of preconditioning. As the ERA5 dataset is not suitable to do a statistical analysis, it is one aim of our current research project to use CESM data to statistically analyze and classify different types of preconditioning leading to extreme seasons.

14) Lines 666-667: Is the CESM large ensemble capable of representing the synoptic processes described in this paper?

It is one aim of our current studies to verify, if climate models such as the CESM large ensemble can represent extreme seasons and the processes leading to them sufficiently well.

**Technical corrections:**

1) Line 372: don't → do not

Changed as suggested.

Changed as suggested.

Changed as suggested.

Changed as suggested.

Changed as suggested.

**References**

Burt, M. A., D. A. Randall, and M. D. Branson, 2016: Dark warming. J. Climate, 29 (2), 705–719

**Reviewer 2**

I am very pleased with the work the authors have done revising the paper. I was particularly concerned that not all selected seasons showed a significant anomaly in any variable, however, revised Table 2 reveals that extreme seasons were ranked either the first or second in at least one of the variables (with an exception for MAM 1990 for NOI, however, that season was extreme in ARI).

Interestingly, extreme winter seasons show strong Es anomalies, while T2m is hardly anomalous. On the other hand, in summer the leading factor is T2m, while in transition seasons the dominant variable is Es again with higher uncertainty though. I am not sure how this can be explained - perhaps, increased standard deviation of Es /T2m during colder / warmer seasons, respectively. It would be good to see a comment on this and also on the reversibility of this relationship, i.e., if extreme winter seasons are those ranked high in Es anomalies (and T2m for summer)?

Thanks a lot for this comment, this is indeed a remarkable difference between extreme Arctic summers and winters, respectively and certainly an interesting observation. However, we would not yet dare to firmly state that there is such a relationship because of the few seasons we used for this analysis. We are currently working on the analysis of Arctic extreme seasons in the CESM climate model, and it is one of our goals to generalize these results based on ERA5 with the model data.

I appreciate extra work done on DJF2016/17 and 2012/2013 seasons, that showed subsidence in the former season and inversion in the low troposphere in the latter season. That was very convincing.

New Figures 4 and 3 look much better.

I think the information on explained variance by the first two modes given in reply to R3 can be added to the manuscript, e.g., by mentioning that the first two PCs explain over XX% of variance.

Please have a look at line 298 of the manuscript, where we mention that, depending on the region and sub-region, PC1 and PC2 "usually explain about 80%-90% of the total variance".

**Minor suggestions** (lines in the manuscript with tracked changes):

l.142: is there a threshold on a displacement of atmospheric blocks? I think the term 'blocking' implies some stationarity, while the term 'track' usually suggests propagation. In this case the 'track' must mean duration rather than propagation; perhaps, the wording can be changed to be less confusing.

There is indeed a threshold for the displacement of atmospheric blocks. We define them as spatial objects with a negative PV anomaly exceeding −1.3 pvu, which show a spatial overlap of at least 70% between each 6-hourly time step. Thus, these features are quasi-stationary and can, to a certain extent, propagate with time, which is why we speak of a "track" in this case.

L. 535: I'd say 'depends on their location within the regions of interest'

Thank you for this suggestion. However, we would like to keep the line as it is ("the impact of cyclones on surface anomalies depends critically on their track relative to the region of interest"), as it seems that not only the cyclones' position *within* a region is important, but also *next to it*. For example, in the case of the winter 2011/12 it is important that the cyclones do not move *through* the region of the Kara-Barents Seas but get stationary in the Nordic Seas (and thus only partly reach the region of interest).

l.545: Maybe 'Having analysed two anomalous winters in the Kara and Barents Seas, in this section we present an anomalous winter in the High Arctic to highlight the difference in dynamical processes leading to extreme seasons in various parts of the Arctic'. I think it may be not only the sea ice conditions that are different between those regions, but also the location and, hence, accessibility for midlatitude synoptic systems.

Thank you for the suggestion. We changed the sentence such that it now reads:

"After analysing two anomalous winters in the Kara and Barents Seas, we now discuss an anomalous Arctic winter in the High Arctic to better understand the different dynamical processes leading to such seasons in Arctic regions with distinct surface conditions."

l.563: There is an interesting relationship with the stratospheric vortex. The timing suggests that SSW may have been caused by anomalous tropospheric conditions in late December and then in January the stratospheric anomaly helped create anomalous tropospheric conditions. (this is just a comment)

Thank you for this comment, this is indeed an interesting fact to think about. Coy and Pawson (2015) have shown that the strong vertical wave activity flux from the troposphere to the stratosphere, which led to the preconditioning and the actual splitting of the polar vortex in DJF 2012/13, has been driven by anomalous upper-tropospheric flow conditions. The formation of a high-latitude cutoff high from a strong tropospheric ridge (and subsequent transition into a lower-stratospheric high) as well as a strong tropospheric storm developing over the North Atlantic seem to have played an important role in preconditioning the SSW. Studies which analyzed previous SSW events over the North Pole (e.g., Coy et al., 2009; Harada et al., 2010) also emphasized the importance of significant synoptic-scale events in the upper troposphere for the occurrence of SSW events.

At the same time, it is maybe not surprising, that such strong events in the stratosphere can also strongly influence the tropospheric conditions below due to stratosphere-troposphere coupling. Several studies showed the impacts of SSWs on tropospheric conditions, causing for example extreme cold air outbreaks (Matthias and Kretschmer, 2020). Butler et al. (2017,

see Fig. R3) could show that a positive surface pressure anomaly over the Arctic concomitant with a negative surface temperature anomaly (such as we found it for DJF 2012/13) is typical following strong SSWs:

[Figure]

Figure R3: Composites of the 60 days after historical SSWs in the JRA-55 reanalysis for (a) mean sea level pressure anomalies (hPa), (b) surface temperature anomalies (K), and (c) precipitation anomalies (mm). The stippling indicates regions that are significantly different from the climatology at the 95% level. Figure from Butler et al. (2017).

l. 623: remove 'We can show that' (if you do this then I'd replace 'find' with 'show' in l.625)

Thank you, changed as suggested.

l.627: I am not sure I understand how multivariate approach helps with different regions with respect to the SIE. The regions have been defined prior to application of the multivariate approach, so even if a single variable was used, you could still do an analysis for various regions depending on the ice coverage.

The multivariate approach allows us to directly compare the unusualness of seasons in one sub-region and between different sub-regions (and thus different surface conditions) via the definition of an "anomaly magnitude", which is based on the Mahalanobis distance $d_M$ in each PCA biplot. This anomaly magnitude combines the anomalies of the six different precursor variables into one measure for unusualness. As different variables dominate the unusualness of a season for varying surface conditions, this method allows us to better compare the different sub-regions. For example, in winter, variations in $H_S$ seem to be strongly determining the anomaly magnitude over ice, whereas over the open ocean, variations in $P$ seem to be more important. Our approach allows us to compare the unusualness of seasons independently of the parameter(s) which caused this unusualness and thus also independently of the predominant surface conditions in a specific region.

A few times 'indicative for': I am not sure if this is correct, I'd say 'indicative of'. Please check.

Thank you for this comment, we changed "indicative for" to "indicative of".

l. 254-255: I think it is worth mentioning that smaller P over the KB seas is probably due to reduced moisture availability compared to the Nordic seas. In both geographics regions, |P|-bar increases over the open-water surface relative to mixed sea-ice conditions.

Thank you for this remark, we extended the concerning sentence such that it now reads "… a larger variability of $P$ can be observed in the Nordic Seas compared to the Kara and Barents Seas, probably due to reduced moisture availability in the latter region."

l.259: I suggest adding 'in **summer** seasons'. It took me a while to understand this sentence as I initially thought that 'season**s**' referred to JJA and DJF.

Changed as suggested.

l. 281: I am not sure if the word 'continuous' is a good choice. I think you meant that seasonal-mean T2m anomaly showed a wide range of values.

We are not sure we understood your remark correctly. In line 280, we refer to our definition of a "continuous season" in terms of the seasonal substructure of daily anomalies of a particular variable, which we introduce in lines 266f: "Thus, we define seasons with $0.8 \leq \left| \frac{\overline{\chi^*}}{\overline{|\chi^*|}} \right| \leq 1$ as seasons with a "continuous" anomaly.". Several summer seasons fulfill this criterion, implicating that the daily anomalies throughout these seasons are more or less continuously either above or below average.

We decided to keep the structure as it is, focusing on the separate discussion of different aspects such as the shape of the plot or the influence of surface conditions. Separating this discussion for summer and winter would lead to more text including more repetition. We also do not discuss each aspect for winter and summer separately, thus, we do not think that the text would benefit from a restructuring.

Changed as suggested.

l. 285: I think that for winter, seasonal-mean anomalies in P are so small, that I am not even sure that saying dry or wet seasons is appropriate as the difference between wet and dry may be not statistically significant. In summer, though, there is a linear relationship between P-bar and |P|-bar, which is worth mentioning instead of 'the driest seasons feature some precipitation events' (|P|-bar values for drier seasons are smaller, as would be expected, meaning that precipitation events are either weaker or less frequent or both). Also, I'd remove or rephrase 'maybe less evident' - it is well evident from the plots.

Regarding the first part of your comment, maybe the different scales for the precipitation anomalies in Figs. 5 and 6 gave the wrong impression, that the seasonal-mean anomalies in winter are much smaller than in summer. To avoid further confusion, we adapted the scales for Fig. 5. As the anomalies are similar in size, we think it is appropriate to distinguish between dry and wet seasons also in winter.

We further adapted the text in lines 284f to stronger emphasize the linear relationship between $\overline{P^*}$ and $\overline{|P^*|}$ due to the skewness of $P$.

We added references to supplementary figures in lines 281f and 289f, where they additionally support our statements.

Changed as suggested.

**References**

Binder, H., Boettcher, M., Grams, C. M., Joos, H., Pfahl, S., and Wernli, H., Exceptional air mass transport and dynamical drivers of an extreme wintertime Arctic warm event, Geophys. Res. Lett., 44, 12028-12036, https://doi.org/10.1002/2017GL075841, 2017.

Butler, A. H., Sjoberg, J. P., Seidel, D. J., and Rosenlof, K. H., A sudden stratospheric warming compendium, Earth Syst. Sci. Data, 9, 63-76, https://doi.org/10.5194/essd-9-63-2017, 2017.

Coy, L., Eckermann, S., and Hoppel, K., Planetary wave breaking and tropospheric forcing as seen in the Stratospheric Sudden Warming of 2006, J. Atmos. Sci., 66, 495-507, https://doi.org/10.1175/2008JAS2784.1, 2009.

Coy, L., and Pawson, S., The major Stratospheric Sudden Warming of January 2013: Analyses and forecasts in the GEOS-5 Data Assimilation System, Mon. Wea. Rev., 143, 491-510, https://doi.org/10.1175/MWR-D-14-00023.1, 2015.

Harada, Y., Atsushi, G., Hiroshi, H., and Fujikawa, N., A major Stratospheric Sudden Warming event in January 2009. J. Atmos. Sci., 67, 2052-2069, doi:10.1175/2009JAS3320.1, 2010.

Matthias, V., and Kretschmer, M., The influence of stratospheric wave reflection on North American cold spells, Mon. Wea. Rev., 148, 1675-1690, https://doi.org/10.1175/MWR-D-19-0339.1, 2020.

Papritz, L., Arctic lower-tropospheric warm and cold extremes: Horizontal and vertical transport, diabatic processes, and linkage to synoptic circulation features, J. Clim., 33, 993-1016, https://doi.org/10.1175/JCLI-D-19-0638.1, 2020.

---

## Author Response (AR3)

**WCD-2021-18**

**Identification, characteristics, and dynamics of Arctic extreme seasons**

Response to the Reviewers' comments by Katharina Hartmuth, Maxi Boettcher, Heini Wernli, and Lukas Papritz

We thank the editor and both reviewers for their additional, helpful comments. We address each comment point by point below. The editors' and reviewers' comments are given in blue and our responses in black.

Please note, that we always refer to the lines in the **updated, revised manuscript** (document without track changes). We supplement this document with a latexdiff-pdf showing changes since the last version of the manuscript.

**Editor**

It would be greatly appreciated if the abstract could be streamlined/condensed. Two places where I think this could easily be done: 1) the background/motivation statements (first three sentences), and the main messages from the case studies (L15-L24).

Thank you for this suggestion. We further shortened the abstract.

The reviewers commented that the use of "continuous" (such as in the paragraph starting L284) is a bit confusing. I tend to agree. It's partly because it could mean a succession of positive and negative daily anomalies in X (i.e., a highly variable time series of daily anomalies), which doesn't seem very continuous. Could you come up with an alternative wording?

Thank you for the remark, we think that there may be still some confusion about the meaning of "continuous" in the way we want to use it. As defined in L260ff, a season with a "continuous anomaly" is a season in which the sign of the daily anomaly in a certain parameter does not change during most of the season. This could be, e.g., a season with a positive $T_{2m}$ anomaly on nearly all days of the season (such as for case study DJF 2011/12, see Fig. 9a). In our understanding, the positive $T_{2m}$ anomaly then exists "continuously" throughout a season. Thus, "continuous" does not mean the succession of positive and negative daily anomalies, but rather the opposite (only positive or only negative anomalies). We adapted all relevant lines in the manuscript to clarify the wording. In some places we found it helpful to replace the word "continuous" by the word "persistent".

I'm having a bit of trouble understanding the revised explanation of the 2016/17 case.

Apologies, but we are not sure what causes the trouble. The essence of our explanation did not change since the original submission. The winter 2016/17 was strongly influenced by preconditioning and the very frequent occurrence of CAOs associated with zonally propagating cyclones across the Kara and Barents Seas.

What is meant by "resulting positive relative frequency anomaly" (L446)? Is it just an increased frequency of CAOs?

Yes. The relative frequency anomalies for weather systems as shown in our timeseries (e.g., Fig. 9), indicate the relative occurrence frequency of, e.g., CAOs compared to climatology. Thus, a positive relative frequency anomaly for DJF 2016/17 means that in that winter, more CAOs than usually occurred.

We clarified the meaning of a "relative frequency anomaly" by inserting the following sentence and equation in L393ff:

"The relative frequency anomaly of a specific weather system in a specific season is calculated as:

$$\frac{f_{seas} - f_{clim}}{f_{clim}} * 100$$

where $f_{seas}$ and $f_{clim}$ denote the spatially averaged seasonal-mean weather system frequencies for the season and the climatology, respectively."

How does the consecutive passage of warm and cold sectors (L445) fit into the explanation?

The consecutive passage of the warm and cold sectors of cyclones through the Kara and Barents Seas in winter 2016/17 is important for the consecutive occurrence of warm air advection, strong precipitation, and enhanced sea ice melt ahead of the cold front and a subsequent CAO behind the cold front (which is even enhanced by the reduced sea ice coverage), leading to intense surface heat fluxes. If, such as in DJF 2011/12, the passage of the cyclones is less zonal and primary the warm sectors of cyclones reach the region of the Kara and Barents Seas, the effects of the warm sector can still be observed, but there is not such a strong increase in CAO frequency and surface heat fluxes.

L185: stronger -> more strongly

Changed as suggested.

L208: suggest: ...of correlations becomes more precise with higher explained variance by...

Changed as suggested.

L231: extra "do" here? Also fix word order: "show either A or B"

Changed as suggested.

L286: "is much larger" - larger than what?

Added "than in summer": "In winter, the overall $T_{2m}$ variability is much larger than in summer…"

L336: ... to classify the seasonS with... as "extreme seasonS".

Changed as suggested.

L356: summer[s]

Changed as suggested.

L404: "In contrast" isn't right here (CAO frequency is reduced just like cyclone frequency). Maybe "At the same time"?

Changed "in contrast" to "at the same time".

Fig. 5 caption: The ratio of THE TWO measures...

Changed as suggested.

**Reviewer 1**

**General comments:**

The authors evaluate the atmospheric conditions during anomalously extreme seasons in the Arctic. This is performed using a regional principal component (PC) analysis (PCA) from ERA5 data of the first two PCs of all seasons from 1979-2018. The PCA uses six key surface variables and divided spatially into 9 Arctic sub-regions subjectively chosen based on climatological sea ice conditions in either the Nordic Seas, Kara and Barents Seas, and the rest of the Arctic. Results identify 2-3 extreme seasons for each season and in each sub-region. The PCA applied here provides a quantification of how anomalous a season is relative to another season, which variables contribute most to the extreme conditions of the respective season, and how consistent those conditions are during those particular seasons. The authors then choose two extreme or anomalous seasons in the Kara-Barents sea during winter (DJF) and one extreme DJF season over the "High Arctic" to further investigate the synoptic weather conditions that were occurring. The chosen seasons are picked based on their orthogonal, yet anomalous or extreme, projections onto the PCs, as well as their diverse processes.

This research nicely demonstrates how PCs can be used to identify seasonal anomalies and extremes in certain regions of the Arctic. It furthermore demonstrates how to use that information to provide an expectation of how an extreme season was characterized with regard to of one of the six variables and how consistent those conditions were. It is certainly a notable method to identify extreme seasons that might be worth analyzing in further detail at shorter time and space scales if desired. I again thank the authors for their consideration of my comments from the previous version, and now there is a more clear description of the two winter cases (DJF 2011/12 and 2016/17). However, given this extra clarity, I am not sure I agree that the two seasons are "fundamentally" different as stated on line 502.

We changed "fundamentally different" to "differ strongly" (L472).

Otherwise, I don't see any issues and think this should be published once this remaining issue is addressed as described below:

1) Thank you for elaborating on the preconditioning of DJF 2011/12 in the review response. I have re-included my figure from the last revision (Fig. 1), which I think clearly shows the atmospheric response directly over the region of anomalously low SIC would be confined to mainly the KBI region and to some degree KBM during SON. I also included an additional figure showing the SIC anomaly and that the temperature anomalies persist and extend deep in the troposphere almost directly over the region of anomalous SIC through February (Fig. 2). Perhaps due to combining the 3 regions, figure R1 does not capture the negative E∗s due to the more limited area of the surface fluxes while the temperature anomalies more broadly surrounded the region because of a lack of other dynamics moving the air masses elsewhere. Thus, combined with all of the other information, I do not necessarily think the two winters are fundamentally different and share many similarities.

Thank you for the additional explanations. We now avoid the formulation "fundamentally different" because there are indeed also similarities between the two seasons. However, based on our analyses (PCA biplots (Fig. 7), time series of anomalies and weather systems (Figs. 9 and 10), and the preconditioning diagram (Fig. 13)) we conclude that there are important differences that are worth being discussed in some detail.

I do think these extreme cases are an interesting story and should be in this paper. My suggestion would be that since the stories do not end up being very different in my view, that sections 5.1-5.3 could be condensed.

We agree that these sections are detailed, also because of the specific questions and recommendations from the other reviewer during the review process. We think that strongly condensing these sections would not do justice to the curiosity of the other reviewer (and hopefully other future readers). But we carefully checked these sections and shortened several paragraphs where appropriate.

I think this is interesting and is a great demonstration of how sensitive seasonal extremes are to blocking (and how there is still a lot to learn about the onset of blocks). It seems that there was similar preconditioning (positive SST anomalies and negative SIC anomalies) present at the beginning of the season in both cases. In 2011/12, this pattern set up in early autumn following the second lowest September sea ice extent (up to that time) and in 2016/17 it started in the previous late winter or spring. The primary difference appears to be in the atmospheric response.

The amplitude of the negative SIC anomalies at the beginning of the season differs strongly between the two seasons (compare Figs. 9c and 10c, day 1). However, and in our opinion even more important, the amplitude of the positive SST and negative SIC anomalies in the *preceding* seasons do also show large differences, especially in the sea surface temperature, which is about +2.5 K above climatology in autumn 2016 compared to an anomaly of +1 K in autumn 2011. The negative SIC anomaly is similar for both autumn seasons and even a bit smaller in JJA 2016 compared to JJA 2011. However, it existed continuously in the Kara and Barents Seas since autumn 2015, thus most probably (partially) enabled the significantly larger SST anomaly at the beginning of DJF 2016/17. We agree that the atmospheric response to such a preconditioning is not necessarily similar in both seasons. But we think that for example the strong positive SST anomaly at the beginning and during the winter of 2016/17 largely contributed to the increase in CAOs due to increasing the air-sea temperature gradient (additionally to the transport of cold air into the region). For that reason, we think that the (longer-term) surface preconditioning was of higher importance for the unusualness of DJF 2016/17 compared to DJF 2011/12.

For whatever reason, the synoptic patterns were such that they did not favor CAOs in DJF 2011/12 while they did in DJF 2016/17, resulting in different surface fluxes and strong but non-consistently signed temperature anomalies in 2016/17 (i.e., surface cyclone tracks were different because the larger-scale flow pattern was different). Persistent blocking in 2011/12 did not provide a way for heat flux introduced into the atmosphere to be advected elsewhere, while 2016/17 was much more of a transient pattern with less frequent blocks. It is interesting that the larger-scale pattern and block more resembled the SIC anomalies in 2011/12 while not so much in 2016/17, and while these differences are interesting and should be noted, there could be many possible reasons as for why they occurred and therefore I think any additional explanation or speculation is beyond the scope of this study.

We agree and we did not speculate about the reasons why the blocking frequencies differed between the two seasons.

**Other specific comments:**

1) Lines 490-494: Cyclones also contributed to the low sea ice during the summer of 2016 (e.g., Finocchio et al. 2020; Lukovich et al. 2021).

Thank you for this remark. Indeed, there was some significant sea-ice reduction in the High Arctic following several extreme storms in August 2016 as stated by Lukovich et al. (2021). In this section we do however focus on the Kara and Barents Seas, which were both already ice-free in July, enhanced by already existing anomalies in SIC and SST (Barents Sea) in the previous winter as shown by Petty et al. (2018).

2) In the DJF 2016/17 case study (Section 5.2), the results about that sea ice transport from several cyclones pushes the sea ice edge further north is a little strong with the given evidence. The PIOMAS data in Figure S7 is simply showing the transport vectors. While it looks quite plausible, there are still other factors that can-not be ruled out, such as the impact of waves or upwelling. So I think this part of the discussion on sea ice can be shortened to say that there is an apparent association with sea ice transport and the passage of cyclones.

Thank you for pointing this out. We shortened this part of the discussion to not over-emphasize this seemingly strong relationship between the sea ice reduction and the passage of the cyclones, as other factors could affect the sea ice transport as well.

**Reviewer 2**

I appreciate the work the authors have done revising the paper. Overall, I am impressed with the amount of work done in this study. The goal to objectively identify extreme Arctic seasons across subregions was ambitious and the authors have successfully achieved it. The case studies are insightful and provide a lot of additional information.

The paper is well written, the analysis is solid and comprehensive. I think what is potentially missing from the analysis is extreme winds (the wind is only mentioned once throughout the paper!); perhaps, this can be covered in future studies. While wind is imbedded into other variables, a strong wind may have a significant direct impact on ecosystems. However, I am not asking for adding this variable to the submitted manuscript.

My very minor suggestion is to show heatmaps Fig.9,10, 14 as a background for time series (frequency anomalies may be indicated in the Captions). In that case, the heat maps will be repeated five times, but it will make easier to related anomalies in variables to a specific circulation regime. I don't know how good this is going to look though. It may also be good to show shading around the running climatological mean as it is hard to say how significant the anomalies are. Lastly, the vertical axis in Fig. 14d can be changed to better represent the range of values (also in Fig. 9d).

Thank you for your comment and the additional suggestions. We added shading around the climatological mean to the timeseries figures (Fig. 9, 10 and 14), which shows the standard deviation of the daily anomalies of the whole study period relative to the transient climatology and is thus a measure of how significant the anomalies are for the winter shown. We further adapted the vertical axis in Figs. 9d and 14d but did not add additional heat map shading, as the plot is already very dense and with the additional shading it would be too confusing.

Fig. 11, caption: to be consistent with (a-c), add '31 January 2017 (day 62)' for (d). SLP is also shown in d, but not mentioned.

Changed as suggested. We further rearranged the figure caption to clarify that SLP contours are shown in (d) as well.

**References:**

Lukovich, J. V., Stroeve, J. C., Crawford, A., Hamilton, L., Tsamados, M., Heorton, H., Massonnet, F., Summer extreme cyclone impacts on Arctic sea ice, J. Clim., 34, 4817-4834, https://doi.org/10.1175/JCLI-D-19-0925.1, 2021.

Petty, A. A., Stroeve, J. C., Holland, P. R., Boisvert, L. N., Bliss, A. C., Kimura, N., Meier, W. N., The Arctic sea ice cover in 2016: a year of record-low highs and higher-than-expected lows, The Cryosphere, 12, 433-452, https://doi.org/10.5194/tc-12-433-2018, 2018.